# A pathogen effector HaRxL10 hijacks the circadian clock component CHE to perturb both plant development and immunity

Mengyao Fu [1,2], Yaoyu Zhou[1,2], Xin Zhang[1,2], Keyi Yang [1,2], Yufeng Xu[1,2], Xingwei Wang [3,4], Zhaodan Chen[5], Yu Wang[1,2], Yabo Shi [3,4], Lin Ma[1,2], Hanguang Liu[1,2], Yuhua Deng[3,4], Shujing Cheng[6], Jinfang Chu[6,7], Jingyi Song[8], Tongjun Sun [8], Yuanchao Wang [5], Wei Wang [3,4] & Mian Zhou [1,2] ✉

The intertwining between the life cycle of plants and their pathogens made the plant circadian clock an integral constituent of the plant immune system. Reciprocally, pathogens were also found to perturb the expression pattern of certain clock genes. However, how pathogens influence clock components remains largely unknown. Here we show that an oomycete effector HaRxL10 directly targets *Arabidopsis* central clock component CCA1 HIKING EXPEDITION (CHE) to manipulate its function. HaRxL10 stabilises CHE by disrupting E3 ligase ZEITLUPE-mediated CHE protein degradation. Surprisingly, the accumulation of CHE does not enhance but rather suppresses CHE function, inhibiting its binding to the downstream gene promoter. HaRxL10 triggers reprogramming of the transcriptome including expression of genes related to circadian oscillations. Moreover, HaRxL10 hijacks CHE to repress plant immunity and manipulate physiological processes, including hypocotyl growth and flowering. Taken together, our study discovers the first plant pathogen effector that directly targets a plant circadian clock component and elucidates the underlying molecular mechanism.

Pathogens and their hosts employ various strategies to gain the upper hand in the tug-of-war[1–3]. Due to the miniature size of most pathogens, their life cycle is strongly affected by environmental factors including temperature and humidity, whose levels oscillate on a daily basis[4]. Consequently, increased success of pathogen colonization happens at a specific time of a day when the environmental conditions are more favourable[5,6]. Ingeniously, plants couple their immune system with the circadian clock, an endogenous timing system that enables plants to anticipate and prepare for such

time-of-day infections[7]. Given the prevalence and importance of this circadian-mediated defence, it is conceivable that pathogens may interfere with plant circadian clock. Indeed, previous studies have shown that pathogen infection can alter the expression pattern of plant circadian clock genes[8–12]. However, how phytopathogens trigger these perturbations and the specific clock components targeted by pathogens remain elusive.

The downy mildew pathogen *Hyaloperonospora arabidopsidis* (*Hpa*) belongs to oomycetes which also include notorious plant

[1]College of Life Sciences, Capital Normal University, Beijing, China. [2]Beijing Key Laboratory of Plant Gene Resources and Biotechnology for Carbon Reduction and Environmental Improvement, Beijing, China. [3]State Key Laboratory for Gene Function and Modulation Research, School of Life Sciences, Peking University, Beijing, China. [4]Center for Life Sciences, Beijing, China. [5]Department of Plant Pathology, Nanjing Agricultural University, Nanjing, China. [6]National Centre for Plant Gene Research (Beijing), State Key Laboratory of Seed Innovation, Institute of Genetics and Developmental Biology, Chinese Academy of Sciences, Beijing, China. [7]University of Chinese Academy of Sciences, Beijing, China. [8]Shenzhen Branch, Guangdong Laboratory of Lingnan Modern Agriculture, Genome Analysis Laboratory of the Ministry of Agriculture and Rural Affairs, Agricultural Genomics Institute at Shenzhen, Chinese Academy of Agricultural Sciences, Shenzhen Guangdong, China. ✉e-mail: mianzhou@cnu.edu.cn

pathogens such as the soybean pathogen *Phytophthora sojae* and the potato late blight pathogen *Phytophthora infestans* which caused the Irish Great Famine. As an obligate biotroph, *Hpa* can only grow on living plants, including *Arabidopsis thaliana*[13]. The availability of genome sequences of both *Hpa* and *Arabidopsis* has led to the widespread use of the *Hpa-Arabidopsis* pathosystem for studying the co-evolution of the host and parasite[14]. And the circadian-mediated plant defence response was first reported in the studies of the interaction between *Hpa* and *Arabidopsis*[8].

To counteract plant defences, *Hpa* secretes apoplastic effectors to the plant extracellular space or cytoplastic effectors into the plant cells. A subset of the cytoplastic effectors is characterized by a highly conserved region known as the RXLR motif (which stands for arginine, any amino acid, leucine, arginine). These RXLR-type effectors typically contain an N-terminal signal peptide for secretion from the pathogen, a RXLR motif flanked by a high frequency of acidic (D/E) residues for targeting the host cell, and a C-terminal effector domain that determines their activity[15]. Bioinformatic analysis has predicted more than 130 RXLR-type effectors from *Hpa*[16], but the functions of most of these effectors have not been characterized.

In this study, we first conducted a yeast-two hybrid (Y2H) screen testing potential interaction between 50 predicted *Hpa* effectors which have been previously reported to be highly expressed during *Hpa* infection[17] and 11 *Arabidopsis* central clock components. We found that the effector named HaRxL10 interacted with several clock components including CCA1 HIKING EXPEDITION (CHE, also known as TCP21, AT5G08330) in yeast (Supplementary Fig. 1), implying that HaRxL10 may affect plant circadian clock in a CHE-dependent manner. CHE belongs to TCP (TEOSINTE BRANCHED 1/CYCLOIDEA/PCF) family transcription factors and recognises the TCP-binding site (TBS) cis-element in the gene promoters. CHE is a well-known central clock component, directly repressing the expression of the morning gene *CIRCADIAN CLOCK ASSOCIATED 1* (*CCA1*, AT2G46830)[18]. CHE also interacts with other clock components, such as the evening-phased transcription factor TIMING OF CAB EXPRESSION 1 (TOC1, AT5G61380) and the E3 ubiquitin ligase ZETLUPE (ZTL, AT5G57360)[18,19]. CHE plays a crucial role in salicylic acid (SA)-mediated plant immunity[20,21]. Our subsequent molecular and genetic data showed that HaRxL10 interacted with CHE to interfere with ZTL-mediated CHE degradation. While HaRxL10 stabilised CHE, it repressed CHE's binding to its downstream regulatory genes. HaRxL10 triggered a significant reprogramming of the transcriptome, particularly affecting the expression of several central clock genes, such as *CHE*, *CCA1*, *TCP22* and *TOC1*. Additionally, HaRxL10 acted as a virulent effector to repress SA-mediated plant immunity and had a notable impact on leaf movement, maltose levels, hypocotyl growth and flowering time in a CHE-dependent manner. Overall, HaRxL10 perturbed the key clock component CHE's transcriptional regulation function as well as several physiological processes.

## Results

### HaRxL10 is a virulent effector which promotes pathogenesis
HaRxL10 is a predicted RXLR-type effector, which contains a potential N-terminal signal peptide for secretion from the pathogen cells (Supplementary Fig. 2). To evaluate the secretion function of the predicted signal peptide (SP), we conducted an invertase secretion assay using the invertase-negative yeast strain YTK12 and the pSUC2 vector which contains the invertase gene but lacks the signal peptide sequence. The predicted signal peptide of HaRxL10 (HaRxL10$^{sp}$) fused with the pSUC2 vector was transformed into YTK12 strain. The transformed yeast (pSUC2-HaRxL10$^{sp}$) could grow on YPRAA media and the reduction of 2,3,5-triphenyltetrazolium chloride (TTC) to insoluble red-coloured 1,3,5-triphenylformazan (TPF) could be observed, implying that the signal peptide of HaRxL10 enables secretion of the invertase (Fig. 1a, b).

*Hpa* is an obligate biotrophic pathogen, which extracts nutrients only from living plant tissue and cannot grow apart from its hosts[13]. Conventional genetic tools such as plasmid transformation and transposon insertion are usually not applicable to this type of pathogen. To surmount this technical difficulty, we employed a natural effector delivery system, the bacterial type-three secretion system (TTSS) of *Pseudomonas syringae* pv *tomato* DC3000 (*Pst* DC3000), to deliver HaRxL10 into the plant cells. To study the function of HaRxL10, its signal peptide was removed, and we referred to ΔSP-HaRxL10 as HaRxL10 in short for convenience. This TTSS of *Pst* DC3000-mediated effector delivery method has been routinely adopted by previous studies[16,22–24]. Introduction of HaRxL10 into *Pst* DC3000 enabled us to study the virulence of HaRxL10 by using *Pst* DC3000 as a control pathogen. We infiltrated *Arabidopsis* plants with either *Pst* DC3000 or *Pst* DC3000-HaRxL10 for subsequent observation of disease phenotype and quantification of bacterial growth. HaRxL10 significantly enhanced the susceptibility of both *Arabidopsis* (Fig. 1c, d) and *N. benthamiana* (Fig. 1e, f) to *Pst* DC3000 infection, suggesting that HaRxL10 may suppress plant immunity to promote pathogenesis. To further test the function of HaRxL10, we generated *Arabidopsis* stable transgenic lines overexpressing HaRxL10 (Supplementary Fig. 3). This ectopic expression of an exogenous protein in *Arabidopsis* transgenic plants was a widely used strategy for studying the function of a *Hpa* effector[25]. We infected two independent HaRxL10 overexpression lines with *Phytophthora sojae*, a nonhost oomycete pathogen of *Arabidopsis*. HaRxL10 dramatically promoted the virulence of *Phytophthora sojae*, which does not typically induce disease symptoms in *Arabidopsis* (Fig. 1g, h). Therefore, HaRxL10 is a powerful virulent effector which could enhance the pathogenesis of both bacterial and oomycete pathogens.

### HaRxL10 interacts with CHE to inhibit CHE protein degradation
The HaRxL10-CHE interaction revealed by our Y2H screen suggests that HaRxL10 may suppress plant defence by interfering with the function of CHE. We first confirmed the interaction between HaRxL10 and full-length CHE *in planta* by Co-IP assays using tobacco transient expression system (Fig. 2a). We further constructed different truncated CHE proteins to dissect specific interacting domains through Y2H (Fig. 2b). The TCP domain is the basic helix-loop-helix (bHLH) motif conserved in the TCP gene family, which mediates DNA binding and protein-protein interaction[26]. The TCP domain of CHE is located in the middle part of the protein from 32 to 86 amino acids. Based on this TCP domain, the other two parts were named CHE$^N$ (N-terminal of CHE) and CHE$^C$ (C-terminal of CHE), respectively. Our Y2H results showed that the TCP domain alone could not interact with HaRxL10 (Fig. 2c). While CHE$^C$ alone was necessary for the interaction with HaRxL10, the addition of TCP domain enhanced this interaction, indicating that TCP and C-terminal domains of CHE determine the interaction with HaRxL10 in yeast (Fig. 2c). Through bimolecular fluorescence complementation (BiFC) assays, we found that TCP and C-terminal domains of CHE also determined the interaction with HaRxL10 in plants (Fig. 2d). Interestingly, unlike the interaction between full-length CHE and HaRxL10 which only occurred in the nucleus, the interaction between CHE$^C$ and HaRxL10 could be observed in both nucleus and cytosol, suggesting that the N-terminal and TCP domains may influence the subcellular localisation of CHE (Fig. 2d). Therefore, the *Hpa* effector HaRxL10 interacts with *Arabidopsis* clock component CHE *in planta*.

The interaction between HaRxL10 and CHE implies that HaRxL10 may influence the function of CHE by altering its subcellular localisation, protein abundance, or transcriptional regulatory function. To investigate the subcellular localisation of these proteins, we expressed HaRxL10-CFP and RFP-CHE alone or together in *N. benthamiana*. We found that HaRxL10-CFP was localised in both the nucleus and cytoplasm, while RFP-CHE was only localised in the nucleus

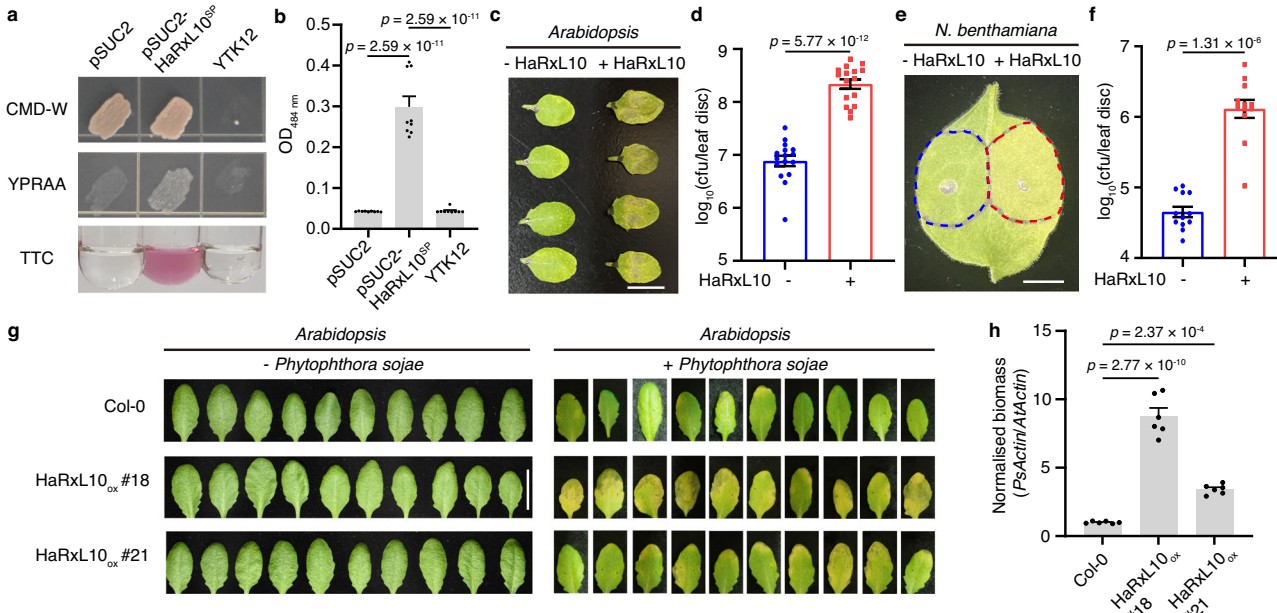

**Fig. 1 | HaRxL10 is a virulent effector which could be secreted from the pathogen. a** Functional validation of the signal peptide of HaRxL10. The yeast YTK12 strain carrying the HaRxL10$^{SP}$ (HaRxL10 signal peptide) fragments fused in-frame to the invertase gene in the pSUC2 vector was able to grow in both CMD-W media and YPRAA media, as well as reduce 2,3,5-triphenyltetrazolium chloride (TTC) to an insoluble red-coloured compound (1,3,5-triphenylformazan, TPF), indicating secretion of invertase. The untransformed YTK12 strain and YTK12 carrying the empty pSUC2 vector served as controls. Experiments were repeated three times with similar results. **b** Quantification of TPF was performed by recording the absorbance at 484 nm of yeast strains in the presence of 0.1% TTC. Data represent the mean ± SEM ($n = 9$, from 3 independent experiments). The $p$ values were calculated by one-way ANOVA followed by Holm-Šídák's multiple comparisons test. **c, d** Disease symptoms (**c**) and bacterial growth (**d**) in wild-type (Col-0) *Arabidopsis* leaves infil-trated with *Pst* DC3000 (-HaRxL10, OD$_{600\ nm}$ = 0.002) or *Pst* DC3000-HaRxL10 (+HaRxL10, OD$_{600\ nm}$ = 0.002) at 3 days post-infiltration (dpi). The 4$^{th}$ and 5$^{th}$ infil-trated leaves from two representative plants were shown. Scale bar, 1 cm. Data represent the mean ± SEM ($n = 16$ plants). The $p$ value was calculated by two-sided unpaired Student's $t$-test. cfu, colony forming unit. Experiments were repeated three times with similar results. **e, f** Disease symptoms (**e**) and bacterial growth (**f**) in *N. benthamiana* leaves infiltrated with *Pst* DC3000 (-HaRxL10, OD$_{600\ nm}$ = 0.0002) and *Pst* DC3000-HaRxL10 (+HaRxL10, OD$_{600\ nm}$ = 0.0002) at 5 dpi. Scale bar, 1 cm. Data represent the mean ± SEM ($n = 12$ leaves). The $p$ value was calculated by two-sided paired Student's $t$-test. cfu, colony forming unit. Experiments were repeated three times with similar results. **g, h** Disease symptoms (**g**) and pathogen growth (**h**) in wild-type and two HaRxL10 overexpression (HaRxL10$_{ox}$) *Arabidopsis* stable lines infected with *Phytophthora sojae* (1000 zoospores) at 3 dpi. The 5th and 6th infil-trated leaves of five plants from one batch of experiment were shown. Biomass of *Phytophthora sojae* (*Ps*) was represented by the amount of *PsActin* using *AtActin* as an internal control analysed by qPCR and normalised to Col-0. Data represent the mean ± SEM ($n = 6$, from 2 independent experiments). The $p$ values were calculated by one-way ANOVA followed by Holm-Šídák's multiple comparisons test.

(Supplementary Fig. 4a). Co-expression with HaRxL10-CFP did not perturb the nuclear localisation of RFP-CHE (Supplementary Fig. 4b). To test whether HaRxL10 affects the protein abundance of CHE in *Arabidopsis*, we generated a CHE complementation line with Myc-tagged CHE (CHE$_{CE}$), which showed statistically indistinguishable transcription level of *CHE* as that of the wild-type (Supplementary Fig. 5). We then infiltrated CHE$_{CE}$ with *Pst* DC3000 or *Pst* DC3000-HaRxL10, and found that HaRxL10 significantly inhibited *Pst* DC3000-induced decrease in the protein abundance of CHE-Myc (Supplementary Fig. 6). Moreover, two independent stable *Arabidopsis* transgenic lines overexpressing YFP-HaRxL10 in CHE-HA (*CHEp:CHE-3×HA* in *che-2* background) exhibited higher CHE protein levels compared to the CHE-HA plants, further demonstrating that HaRxL10 promotes CHE protein accumulation (Fig. 2e). To determine the mechanism under-lying this HaRxL10-mediated increase in CHE protein level, we con-ducted a cell-free degradation assay. We found that similar to the effect of MG132, a chemical inhibitor of the 26S proteasome, HaRxL10 significantly inhibited CHE protein degradation, suggesting that HaRxL10 helps to stabilise CHE protein (Fig. 2f, g).

Since the domain arrangement of HaRxL10 (Fig. 2h) follows the typical structure of RXLR-type effectors whose C-terminal domains usually determine their effector activity[15], we reasoned that the C-terminal domain of HaRxL10, which follows the D/E residues, may manifest the function of this RXLR-type effector. Based on the location of two D/E residues, we constructed HaRxL10$^{37-221}$ and HaRxL10$^{68-221}$,

two truncated versions of HaRxL10. We speculated that the interaction between the C-terminal of HaRxL10 and CHE is responsible for pro-moting the protein stability of CHE. To test this, we performed a Y2H assay and found that HaRxL10$^{37-221}$, but not HaRxL10$^{68-221}$, interacts with CHE (Fig. 2h, i and Supplementary Fig. 7a). We then investigated the effect of HaRxL10$^{37-221}$ on CHE protein stability using a cell-free degradation assay and found that it stabilises CHE protein (Fig. 2j, k). Moreover, HaRxL10$^{37-221}$ displays a similar predicted three-dimensional structure of the full-length HaRxL10 (Supplementary Fig. 7b, c), further implying that the C-terminal of HaRxL10 is the functional domain.

## HaRxL10 inhibits the interaction between CHE and ZTL
Next, we explored how HaRxL10 stabilises CHE. An earlier study revealed that the stability of CHE is regulated by another clock com-ponent ZTL, a member of the E3 ubiquitin ligase Skp-Cullin-F-box (SCF) component[19]. Through pull-down assays, we observed a direct inter-action between purified MBP-His-CHE and GST-His-ZTL in vitro (Fig. 3a). We also observed an interaction between CHE and ZTL in yeast and plant systems, where the C-terminal domain of CHE deter-mined the interaction with both ZTL and HaRxL10 (Fig. 3b, c, Fig. 2c, d). HaRxL10 did not interact with ZTL in either yeast or plant cells (Sup-plementary Fig. 8). Therefore, we hypothesised that HaRxL10 might interfere with the interaction between CHE and ZTL, thereby stabilis-ing the CHE protein. To test this hypothesis, we performed a yeast-three hybrid (Y3H) assay and found that the interaction between CHE

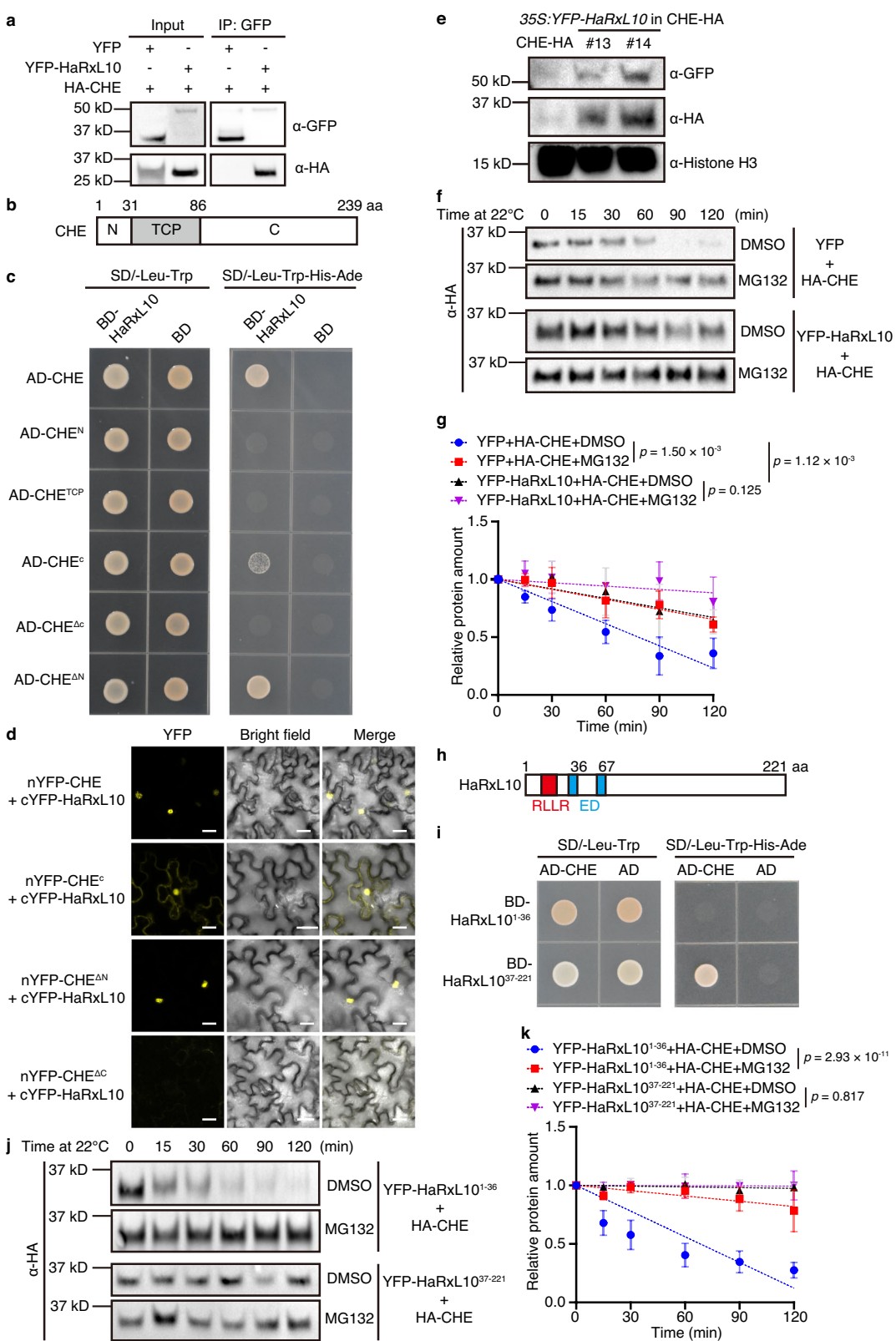

and ZTL was significantly weaker in the presence of HaRxL10 compared to the control YFP protein. The CHE-ZTL interaction was repressed by HaRxL10 in the presence of 3-AT, which inhibited autoactivation in the yeast system (Fig. 3d). Moreover, we studied the effect of His-HaRxL10 on the interaction between MBP-His-CHE and GST-His-ZTL by a pull-down assay. Our results showed that the interaction between MBP-His-CHE and GST-His-ZTL was inhibited by His-HaRxL10

in a dose-dependent manner, suggesting that HaRxL10 disrupts the CHE-ZTL interaction (Fig. 3e). Furthermore, through microscale thermophoresis (MST) assay, we demonstrated that HaRxL10 disrupted high-affinity binding between CHE and ZTL (Fig. 3f). Collectively, our results suggest that HaRxL10 inhibits the interaction between CHE and ZTL. The evening clock component TOC1[27] is a known CHE-interacting protein[18]. To investigate whether HaRxL10 may interfere with the

**Fig. 2 | The interaction of HaRxL10 and CHE stabilises CHE protein. a** Co-IP of YFP-HaRxL10 and HA-CHE. HA-CHE and YFP-HaRxL10 or YFP (as negative control) were transiently expressed in *N. benthamiana* for 36 h followed by 100 µM MG132 treatment for 12 h and samples were then collected for protein extraction. Proteins were immuno-precipitated by GFP beads followed by Western blot with α-GFP and α-HA antibodies respectively. The molecular weight of YFP is 33.2 kD. The molecular weight of YFP-HaRxL10 is 52.3 kD. The molecular weight of HA-CHE is 28.3 kD. Experiments were repeated three times with similar results. **b** Schematic illustration of different domains of CHE protein. aa, amino acid. **c** Interactions of HaRxL10 with full-length and truncated CHE in Y2H assays. Synthetic dropout medium without leucine and tryptophan (SD/-Leu-Trp) was used for positive yeast transformant selection. Synthetic dropout medium without leucine, tryptophan, histidine and adenine (SD/-Leu-Trp-His-Ade) was used for the selection of protein interaction by the reporter gene *HIS3*. Photographs were taken 2 days after plating of yeast cells with OD₆₀₀ ₙₘ = 1. This experiment was repeated three times with similar results. **d** BiFC assays demonstrating the interactions of full-length and truncated CHE with HaRxL10 in *N. benthamiana*. cYFP-HaRxL10 were transiently co-expressed with nYFP-CHE, nYFP-CHE^C, nYFP-CHE^ΔN, or nYFP-CHE^ΔC in tobacco leaves for two days and then observed under a confocal microscope. Scale bars, 20 µm. This experiment was repeated three times with similar results. **e** Protein levels were detected in stable transgenic *Arabidopsis* seedlings expressing *CHEp:CHE-3×HA* in *che-2* (CHE-HA) and *35S:YFP-HaRxL10* in CHE-HA (two independent homozygous T3 lines). Histone H3 was used as an internal control. The molecular weight of YFP-HaRxL10 is

52.3 kD. The molecular weight of CHE-3×HA is 30.7 kD. The molecular weight of Histone H3 is 15 kD. This experiment was repeated three times with similar results. **f**, **g** Representative Western blot images (**f**) and quantification of three independent experiments (**g**) of cell-free degradation assays. Protein extracts were prepared from *Arabidopsis* protoplasts expressing HA-CHE and YFP (negative control) or YFP-HaRxL10 and treated with DMSO or MG132 (50 µM). The molecular weight of HA-CHE is 28.3 kD. Data represent the mean ± SEM (*n* = 3 independent experiments). The *p* values were calculated by two-way ANOVA followed by Holm-Šídák's multiple comparisons test of the slopes derived from linear regression. **h** Schematic illustration of the domain structure of HaRxL10. aa, amino acid. **i** Interaction between HaRxL10 truncations and CHE in Y2H assays. SD/-Leu-Trp medium was used for positive yeast transformant selection. SD/-Leu-Trp-His-Ade medium was used for the selection of protein interaction by the reporter gene *HIS3*. Photographs were taken 2 days after plating yeast cells with OD₆₀₀ ₙₘ = 1. These experiments were repeated three times with similar results. **j**, **k** Representative Western blot images (**j**) and quantification of three independent experiments (**k**) of cell-free degradation assays. Protein extracts were prepared from *Arabidopsis* protoplasts expressing HA-CHE and YFP-tagged truncated HaRxL10 (YFP-HaRxL10^1-36 or YFP-HaRxL10^37-221) and treated with DMSO or MG132 (50 µM). The molecular weight of HA-CHE is 28.3 kD. Data represent the mean ± SEM (*n* = 3 independent experiments). The *p* values were calculated by two-way ANOVA followed by Holm-Šídák's multiple comparisons test of the slopes derived from linear regression.

---

interaction of CHE with other proteins, we performed Y3H assay and found that HaRxL10 could not inhibit the interaction of CHE with TOC1 (Supplementary Fig. 9a), probably due to distinct interaction regions of CHE with HaRxL10 and TOC1, as validated by Y2H assays (Supplementary Fig. 9b and Fig. 2c).

### HaRxL10 sequesters CHE to repress its transcriptional regulatory effects on the downstream gene

In addition to stabilising CHE, HaRxL10 may also affect CHE's transcriptional regulatory function on its target genes. While the transcription of the clock gene is typically regulated by other clock components, how the transcription of *CHE* is regulated remains largely unknown. A bioinformatic analysis of the *CHE* promoter sequence revealed the presence of a TBS, suggesting that CHE may regulate its gene expression. Y1H experiment showed that CHE could indeed bind to its own promoter (Fig. 4a). Further ChIP-qPCR assays using *35S:YFP-CHE* transgenic plants demonstrated the recruitment of YFP-CHE to the TBS and the nearby region, providing evidence of CHE binding to its promoter *in planta* (Fig. 4b, c). Additionally, dual-luciferase assays revealed that the CHE protein activated the transcription of the *CHE* gene (Fig. 4d). Collectively, these results indicate that CHE positively regulates its gene expression by binding to the *CHE* promoter.

To investigate the effect of HaRxL10 on CHE's transcriptional regulation function, we selected the *CHE* gene as a representative of downstream genes regulated by CHE. Surprisingly, two independent *Arabidopsis* stable transgenic lines overexpressing *HaRxL10* exhibited lower *CHE* transcripts abundance compared to the wild-type (Fig. 4e). Furthermore, dual-luciferase assays showed that both HaRxL10 and its truncated protein, HaRxL10^37-221, significantly inhibited the transcription of *CHE* in the presence of CHE protein, confirming that HaRxL10^37-221 is the functional domain and suggesting that HaRxL10 may also interfere with the CHE's transcriptional regulation on its downstream target genes (Fig. 4f). We hypothesised that HaRxL10 may hinder CHE's binding to the promoter of its target gene. To test this hypothesis, we performed EMSA assays and found that HaRxL10 and HaRxL10^37-221, but not HaRxL10^1-36, inhibited the binding of CHE to TBS (Fig. 4g, h). To further study the effect of HaRxL10 on the binding of CHE to its target gene promoter *in planta*, we performed a ChIP-qPCR assay using transgenic *Arabidopsis* seedlings expressing *CHEp:CHE-3×HA* in *che-2* (CHE-HA) or *35S:YFP-HaRxL10* in CHE-HA. Our results showed that HaRxL10 inhibited the binding of CHE to the *CHE* promoter (Fig. 4i). Our Y2H results showed that HaRxL10 could not bind to

the TCP domain of CHE, which usually mediates the DNA binding and probably contains a TBS-binding site (Fig. 2c). Therefore, HaRxL10 probably could not bind to the TBS-binding site on CHE's surface. The binding of HaRxL10 to the C-terminal of CHE (Fig. 2c) may trigger a conformational change of CHE, inhibiting its DNA-binding ability. Collectively, these results suggest that HaRxL10 sequesters CHE to repress its transcriptional regulation of the downstream target genes.

### HaRxL10 affects SA-related defence gene expression to enhance plant disease susceptibility in a CHE-dependent manner

Next, we investigated the role of CHE in the regulation of the virulence of HaRxL10. We utilised two strategies to express HaRxL10 in *Arabidopsis* plants. One was the TTSS of *Pst* DC3000-mediated effector delivery. The other is the generation of an effector overexpression line of *Arabidopsis*. For the *Pst* DC3000-mediated way, the phenotype of HaRxL10-induced elevated susceptibility to the bacterial infection in the wild-type was blocked in the *che* mutant and rescued in the *CHE* complementation line (Fig. 5a, b). For the ectopic expression in transgenic plants, we generated *HaRxL10* overexpressing lines in the *che* background and found that constitutive high expression of *HaRxL10* in *Arabidopsis* also enhanced susceptibility to *Pst* DC3000 in the wild-type but not in the *che* mutant, suggesting that the virulent function of HaRxL10 is dependent on CHE (Supplementary Fig. 10). Consistent with our finding that HaRxL10^37-221 determined the interaction with CHE and the interference with CHE's binding to TBS, HaRxL10^37-221 is sufficient to enhance pathogenesis, further confirming that HaRxL10^37-221 is the functional domain of this effector (Fig. 5c and Supplementary Fig. 11).

To investigate whether HaRxL10 triggers transcriptome reprogramming and its dependence on CHE, we performed a time-course RNA-seq experiment using wild-type and *che-2* plants upon pathogen infection with *Pst* DC3000 or *Pst* DC3000-HaRxL10 respectively (Fig. 5d). HaRxL10-repressed genes significantly overlapped with CHE-induced genes (Fig. 5e–g). Expression patterns of these 814 HaRxL10-repressed and CHE-induced genes exhibited high similarity between *Pst* DC3000-HaRxL10-infiltrated wild-type sample and *Pst* DC3000-infiltrated *che-2* sample, indicating that HaRxL10 mimics loss-of-function of *CHE* (Fig. 5g). GO analysis revealed that the most enriched functions of these 814 overlapping genes were related to response to SA and regulation of defence response, such as *ACCELERATED CELL DEATH 6* (*ACD6*, AT4G14400) and *CYSTEINE-RICH RECEPTOR-LIKE PROTEIN KINASE 4* (*CRK4*, AT3G45860) (Fig. 5h).

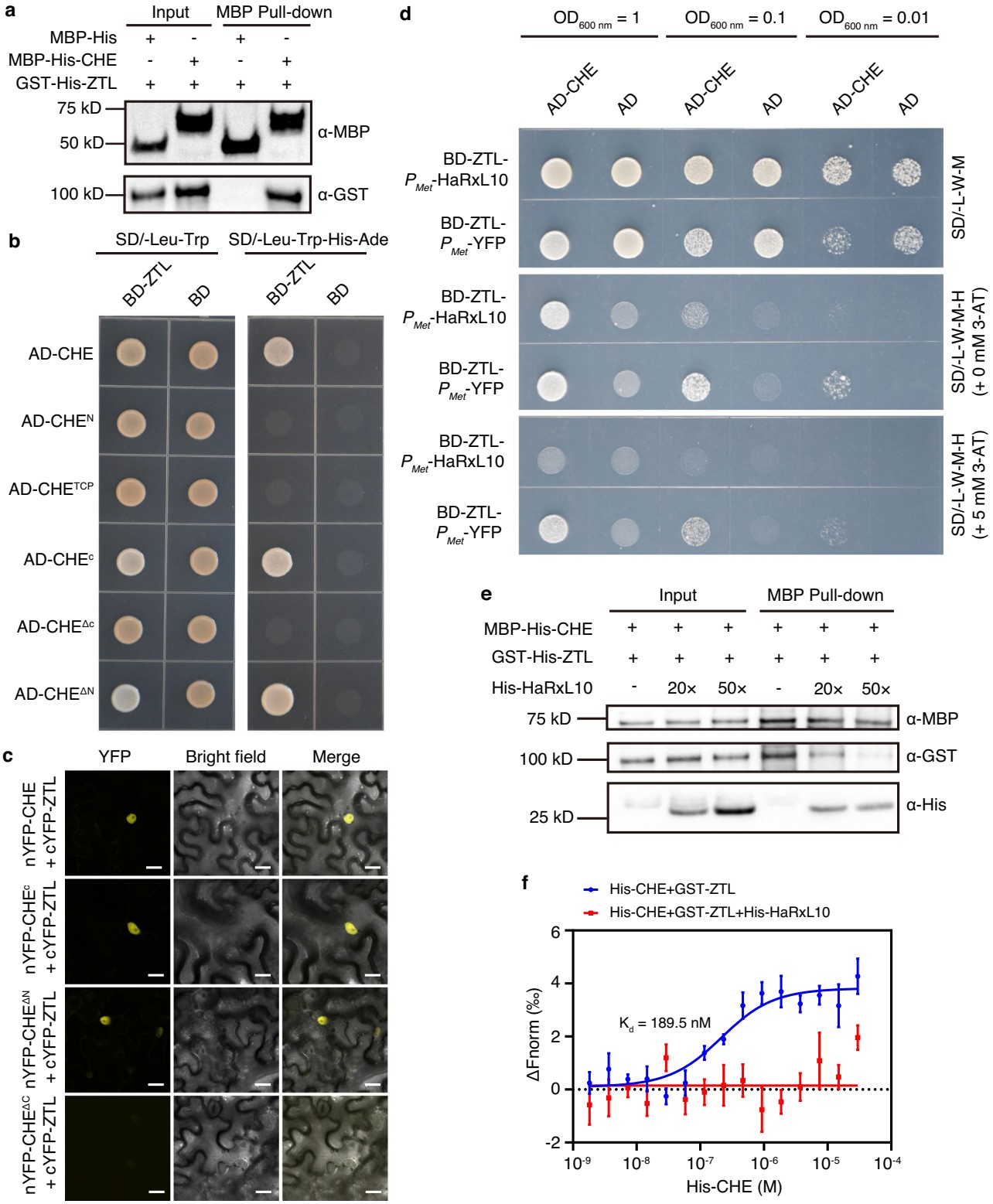

*ACD6* encodes a multipass membrane protein with an ankyrin domain and has been reported to function as an ion channel, regulate the SA accumulation, cell death and defence responses to bacteria[28,29]. *CRK4* encodes cysteine-rich receptor-like protein kinases and is involved in programmed cell death as well as defence responses to pathogens and SA[30–32]. To further confirm this finding, we conducted RT-qPCR assays and found that the expression of *ACD6* and *CRK4* were significantly repressed by *Pst* DC3000-HaRxL10 compared to *Pst* DC3000 in the wild-type plants (Fig. 5i, j). However, this HaRxL10-mediated

suppression of *ACD6* and *CRK4* was compromised in *che-2* plants (Fig. 5i, j). We also analysed the expression of SA-responsive marker gene *PR2* after pathogen infection with *Pst* DC3000 or *Pst* DC3000-HaRxL10 respectively. Our results showed that HaRxL10 suppressed *PR2* expression in a CHE-dependent way (Fig. 5k). We did not find the TBS within the promoter sequences of these SA-responsive genes, indicating that CHE may indirectly affect the expression of these genes.

A previous study revealed that CHE positively regulates the expression of the SA synthesis gene *ISOCHORISMATE SYNTHASE 1*

**Fig. 3 | The interaction of CHE with ZTL is suppressed by HaRxL10. a** GST-His-ZTL could be pulled down by MBP-His-CHE in vitro. Protein combinations were pulled down by amylose resin followed by Western blot with α-MBP and α-GST antibodies respectively. MBP-His alone served as a negative control. The molecular weight of MBP-His is 48.5 kD. The molecular weight of MBP-His-CHE is 70.1 kD. The molecular weight of GST-His-ZTL is 99 kD. **b** Interactions of ZTL with full-length and truncated CHE in Y2H assays. Synthetic dropout medium without leucine and tryptophan (SD/-Leu-Trp) was used for positive yeast transformant selection. Synthetic dropout medium without leucine, tryptophan, histidine and adenine (SD/-Leu-Trp-His-Ade) was used for the selection of protein interaction by the reporter gene *HIS3*. Photographs were taken 2 days after plating of yeast cells with $OD_{600\,nm} = 1$. **c** BiFC experiments illustrating the interactions of full-length and truncated CHE with ZTL in *N. benthamiana*. cYFP-ZTL were transiently co-expressed with nYFP-CHE, nYFP-CHE$^C$, nYFP-CHE$^{ΔN}$, or nYFP-CHE$^{ΔC}$ in tobacco leaves for two days and then observed under a confocal microscope. Scale bars, 20 μm. **d** Y3H assays illustrating that HaRxL10 significantly inhibits the interaction between CHE and ZTL. Synthetic dropout medium without leucine, tryptophan and methionine (SD/-L-W-M) was used for positive yeast transformant selection.

Synthetic dropout medium without leucine, tryptophan, methionine and histidine (SD/-L-W-M-H) was used for the selection of protein interaction by the reporter gene *HIS3*. 3-AT was used to inhibit the self-activation in yeast. Photographs were taken 3 days after plating of the yeast cells with $OD_{600\,nm} = 1$, 0.1 or 0.01 respectively. **e** His-HaRxL10 inhibited the interaction between MBP-His-CHE and GST-His-ZTL. Protein combinations were pulled down by amylose resin followed by Western blot with α-MBP, α-GST and α-His antibodies respectively. Equal amounts of MBP-His-CHE and GST-His-ZTL were used. For the competition of protein-protein interaction, 20-fold or 50-fold amount of His-HaRxL10 proteins were added. The molecular weight of MBP-His-CHE is 70.1 kD. The molecular weight of GST-His-ZTL is 99 kD. The molecular weight of His-HaRxL10 is 28.5 kD. **f** MST binding curves of His-CHE to GST-ZTL with or without His-HaRxL10. ΔFnorm represents the normalised fluorescence of NHS-labelled GST-ZTL protein. Data represent the mean ± SEM ($n = 5$ independent titrations). $K_d$ was estimated through nonlinear regression, and the best-fit values are shown. The 95% confidence interval of the dissociation constant ($K_d$): 90.91 - 385.6 nM. The experiments (**a**–**f**) were repeated three times with similar results.

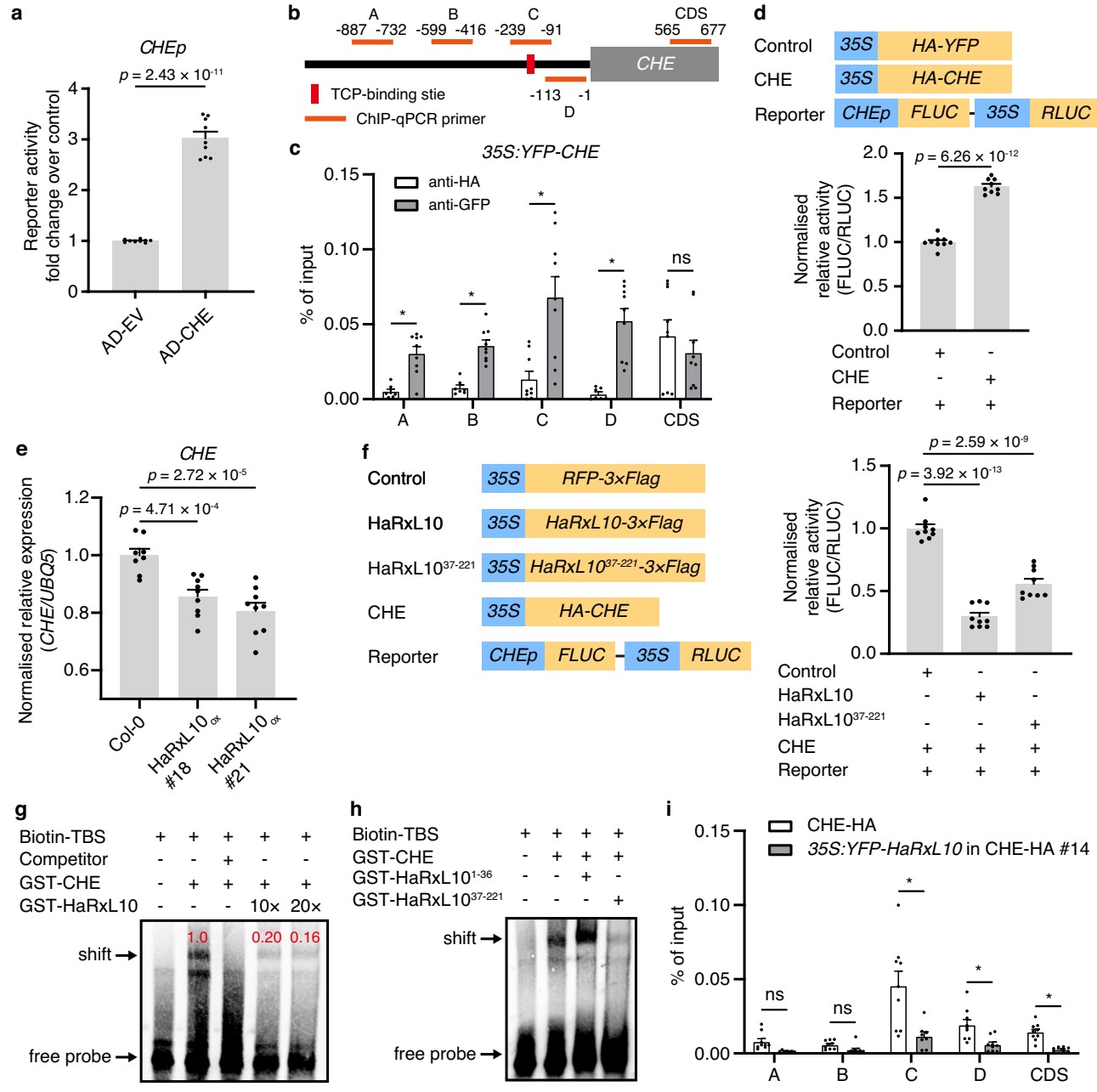

**Fig. 4 | HaRxL10 inhibits the binding of CHE to the cis-element, TCP-binding site. a** CHE binds to its promoter in Y1H assays. $OD_{420\,nm}$ was measured and β-galactosidase reporter activities are shown as fold change of CHE (AD-CHE) over AD empty vector (AD-EV as control) in yeast strains with *CHE* promoter. The data are shown as mean ± SEM (*n* = 9, 3 independent experiments with 3 technical replicates). The *p* value was calculated using two-sided unpaired Student's *t*-test. **b** Illustration of primers used for the ChIP-qPCR experiment. **c** Binding of CHE to different regions of *CHE* promoter revealed by ChIP-qPCR. Seven-day-old *35S:YFP-CHE* seedlings were used. Anti-HA antibody was used as a negative control. Data represent the mean ± SEM (*n* = 9, 3 independent experiments with 3 technical replicates). *\*q* < 0.01; ns, not significant (multiple *t* test with multiple comparison correction by false discovery rate using the two-stage step-up method of Benjamini, Krieger, and Yekutieli). **d** Dual-luciferase assays performed using *N. benthamiana* leaves transiently co-expressing HA-CHE or HA-YFP (control) and the reporter driven by *CHE* promoter. The ratio of firefly luciferase (FLUC) and *Renilla* luciferase (RLUC) activities was calculated and normalised to the control. The data are shown as mean ± SEM (*n* = 9, 3 independent experiments with 3 technical replicates). The *p* value was calculated using two-sided unpaired Student's *t*-test. **e** The expression of *CHE* in the wild-type (Col-0) and two HaRxL10 overexpression (HaRxL10$_{ox}$) *Arabidopsis* transgenic lines was analysed by RT-qPCR with *UBQ5* as an internal control and normalised by the expression in Col-0. The data are shown as mean ± SEM (*n* = 9, 3 independent experiments with 3 technical replicates). The *p* values were calculated by one-way ANOVA followed by Holm-Šídák's multiple comparisons test.

**f** Dual-luciferase assays performed using *Arabidopsis* protoplasts transiently co-expressing different protein combinations and the reporter driven by *CHE* promoter. The ratio of firefly luciferase (FLUC) and *Renilla* luciferase (RLUC) activities was calculated and normalised to the control. The data are shown as mean ± SEM (*n* = 9, 3 independent experiments with 3 technical replicates). The *p* values were calculated by one-way ANOVA followed by Holm-Šídák's multiple comparisons test. **g** The binding between GST-CHE protein and biotin-labelled TBS probes was attenuated by GST-HaRxL10 revealed by EMSA assays. 200-fold un-labelled TBS was used as the competitor. 5 pmol of GST-CHE protein was used. 10-fold or 20-fold GST-HaRxL10 protein was added. This experiment was repeated three times with similar results. **h** Effects of truncated HaRxL10 on the binding between CHE and TBS revealed by EMSA assays. 200-fold un-labelled TBS was used as the competitor. 5 pmol of GST-CHE protein was used. 20-fold GST-tagged truncated HaRxL10 protein (GST-HaRxL10$^{1-36}$ and GST-HaRxL10$^{37-221}$) were added. This experiment was repeated three times with similar results. **i** HaRxL10 inhibited the binding of CHE to the *CHE* promoter revealed by ChIP-qPCR. Seven-day-old transgenic *Arabidopsis* seedlings expressing *CHEp:CHE-3×HA* in *che-2* (CHE-HA) or *35S:YFP-HaRxL10* in CHE-HA (line #14) were used. Anti-HA antibody was used. Data represent the mean ± SEM (*n* = 9, 3 independent experiments with 3 technical replicates). *\*q* < 0.01; ns, not significant (multiple *t* test with multiple comparison correction by false discovery rate using the two-stage step-up method of Benjamini, Krieger, and Yekutieli).

(*ICS1*, AT1G74710*)* by direct binding to the TBS within its gene promoter[20]. To test the effect of HaRxL10 on *ICS1* transcription, we performed dual-luciferase assays in tobacco leaves and observed significant repression of *ICS1* transcription in the presence of CHE and HaRxL10 (Supplementary Fig. 12a). However, HaRxL10 induced *ICS1* expression in wild-type *Arabidopsis* plants and this HaRxL10-mediated *ICS1* induction may require functional CHE depending on different experiment settings (Supplementary Fig. 12b, c). These data indicate that although HaRxL10 represses CHE-mediated *ICS1* induction, HaRxL10 may promote expression of other *ICS1* positive regulators, such as *SAR DEFICIENT 1* (*SARD1*, AT1G73805) and *CAM-BINDING PROTEIN 60-LIKE G* (*CBP60g*, AT5G26920). To test this hypothesis, we used wild-type, *che-2* and stable HaRxL10 overexpression transgenic plants infiltrated with *Pst* DC3000. Our results showed that the expression of *SARD1* and *CBP60g* were induced by HaRxL10 (Supplementary Fig. 12d, e). Consistent with these results, overexpression of HaRxL10 promoted free SA contents (Supplementary Fig. 12f). HaRxL10-triggered increase of SA seems contradictory to the function of HaRxL10 as the virulence effector to repress plant immunity. To address this puzzle, we further analysed the expression of *PR2*, which is a downstream defence gene identified by our RNA-seq experiment and also a more downstream proxy of SA-mediated defence level. HaRxL10 overexpression and *che-2* plants showed decreased *PR2* expression (Supplementary Fig. 12g), which were consistent with enhanced disease susceptibility of these plants (Supplementary Fig. 10c). In summary, HaRxL10 affects SA-mediated defence regulation in a complicated way through multiple plants targets (Supplementary Fig. 12h). HaRxL10 represses CHE-mediated *ICS1* induction but promotes SARD1/CBP60g-mediated *ICS1* induction, resulting in higher *ICS1* expression and SA level. HaRxL10 represses *PR2* expression to dampen defence response and cause disease susceptibility.

In addition to *ICS1*, CHE may regulate other defence genes directly. To identify the potential direct target genes of CHE, we constructed linear models followed by empirical Bayesian analysis on the two-way interaction between the presence of CHE and HaRxL10 on the time-course RNA-seq data and validated these conclusions through RT-qPCR experiments. As a result, we found that the HaRxL10-induced expression of *CYTOKININ OXIDASE 5* (*CKX5*, AT1G75450), *ANAC055* (At3G15500), and *STRESS INDUCED FACTOR 1* (*SIF1*, AT1G51830) required CHE (Supplementary Fig. 13a–f). Furthermore, the promoters of *CKX5*, *ANAC055*, and *SIF1* contain TBS which could be bound by CHE in yeast, indicating that *CKX5*, *ANAC055*, and *SIF1* are potential direct

target genes of CHE (Supplementary Fig. 13g). *CKX5* encodes a protein whose sequence is similar to cytokinin oxidase/dehydrogenase, which catalyses the degradation of cytokinin. The expression of *CKX5* displays a circadian rhythm at basal conditions (Supplementary Fig. 13h), suggesting that *CKX5* may be regulated by the circadian clock. A previous study has shown that together with SA, cytokinin stimulates defence responses against virulent oomycete pathogen *Hpa* Noco2, suggesting the role of cytokinin-SA crosstalk in defence responses to biotrophs[33]. Therefore, HaRxL10 may induce *CKX5* expression to downregulate the level of cytokinin, enhancing the virulence of *Hpa*. Transcription factor *ANAC055* is the direct target of MYC2 to suppress SA accumulation and mediate coronatine-induced stomatal reopening[34]. HaRxL10-induced *ANAC055* may result in suppression of SA signalling, favouring pathogenesis. *SIF1* encodes a leucine-rich repeat receptor-like protein kinase and is rapidly induced by biotic stress, indicating its potential role in plant immunity. Moreover, the basal expression of *SIF1* shows a circadian oscillation (Supplementary Fig. 13i), consistent with our finding that CHE may directly regulate the transcription of *SIF1*. Additionally, the *che* mutant plants displayed higher susceptibility to *Pst* DC3000 and CHE-induced genes are enriched with defence response genes, indicating that CHE positively regulates innate immunity (Fig. 5b, c and Supplementary Fig. 14). In summary, our results suggest that CHE regulates transcription of defence genes and HaRxL10 interferes with CHE-mediated immunity to trigger pathogenesis.

## HaRxL10 affects the expression of central clock genes and circadian-controlled rhythmic genes

Since HaRxL10 could trigger transcriptome reprogramming and repress the expression of clock gene *CHE*, it is possible that HaRxL10 could also influence the expression of other clock genes due to the complicated transcriptional regulation network of the circadian clock. Our time-course RNA-seq results suggested that the average expression of *CHE*, *REVEILLE 4* (*RVE4*, AT5G02840), *TCP22* (AT1G72010), and *LIGHT-REGULATED WD 1* (*LWD1*, AT1G12910) was significantly repressed by HaRxL10, while the expression profiles of the other 16 clock genes were not significantly altered by HaRxL10 (Supplementary Fig. 15). The RNA-seq data indicated that HaRxL10-triggered repression of *TCP22* but not *RVE4* and *LWD1* required CHE (Supplementary Fig. 16a–c). The subsequent single time-point RT-qPCR results supported that HaRxL10-triggered repression of *TCP22* is dependent on CHE (Supplementary Fig. 16d). RT-qPCR results showed that although

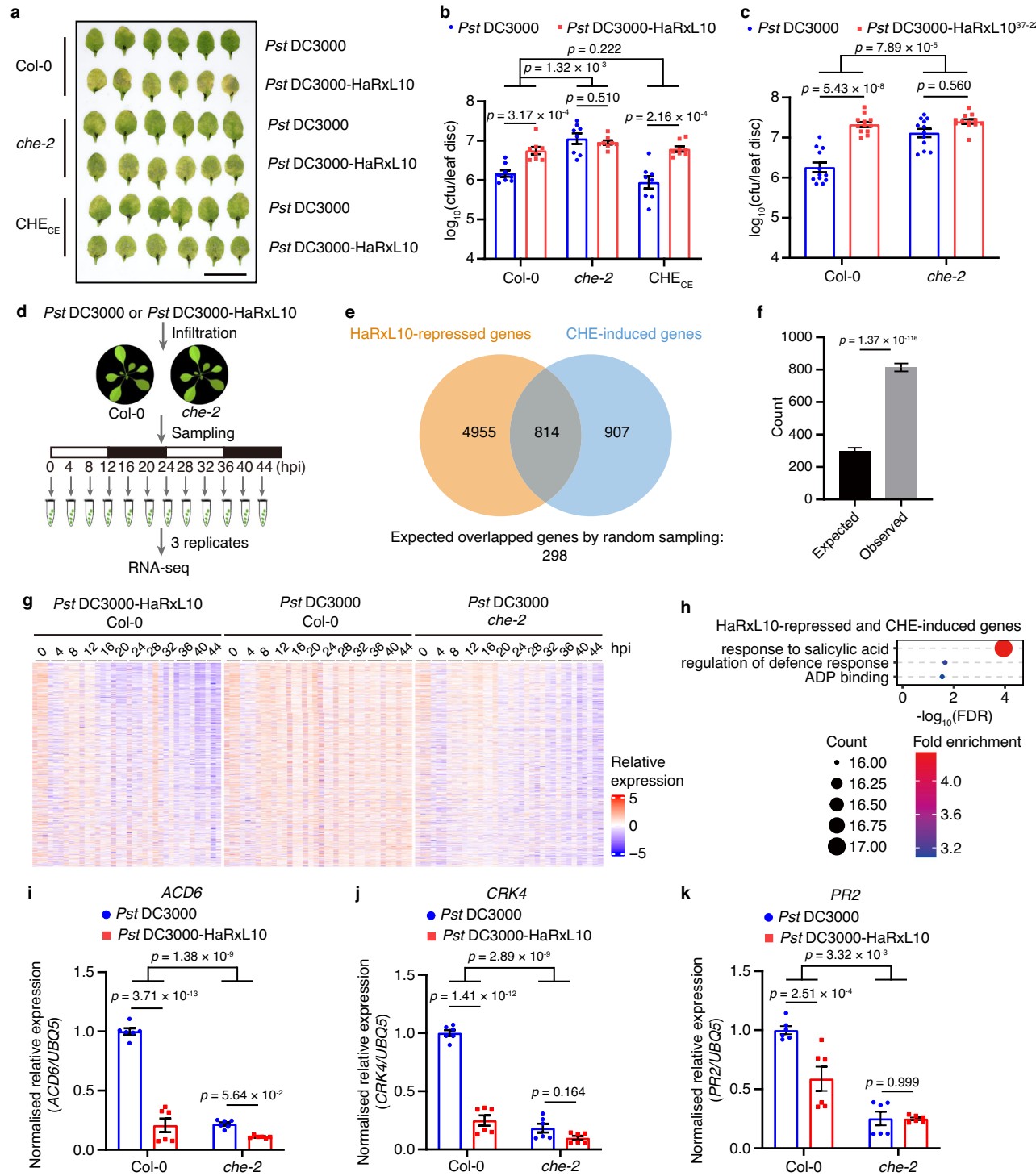

the reduction in gene expression of *RVE4* and *LWD1* was less pronounced in *che-2* mutants compared to wild-type plants, *Pst* DC3000-HaRxL10 could still significantly repress the expression of *RVE4* and *LWD1* (Supplementary Fig. 16e, f). Therefore, it is probable that the effects of CHE on HaRxL10-mediated expression changes of *RVE4* and *LWD1* are marginal.

Light is the strongest input signal of the plant circadian clock, which synchronises the internal clock with the external environmental cues[35]. Due to this reason, some known clock mutants display no or very mild clock dysfunction under diurnal conditions, but obvious clock period change phenotype under constant light conditions[36,37]. Light significantly elevates plant defence against pathogen infection

and constant light conditions reduce the effect of pathogen infection[38]. Therefore, the light and dark cycle was used in our experiment to provide a favourable condition for bacterial infection. This diurnal condition combines the effect of the internal clock and environmental light on gene expression. Therefore, the effect of HaRxL10 on the clock gene expression may be diminished under diurnal conditions. To further study the effect of HaRxL10 on the clock gene expression, we analysed the circadian expression pattern of two key central clock genes, *CCA1* and *TOC1*, in the wild-type and HaRxL10 overexpression plants under the constant light condition. *CCA1* is a well-known target gene of CHE[18]. Our results showed that HaRxL10 overexpression plants displayed lower amplitude and average

**Fig. 5 | HaRxL10 affects SA-related defence gene expression to enhance plant disease susceptibility in a CHE-dependent manner. a, b** Disease symptoms (**a**) and bacterial growth (**b**) in wild-type (Col-0), *che-2* and *CHE* complementation line (CHE$_{CE}$) *Arabidopsis* leaves infiltrated with *Pst* DC3000 or *Pst* DC3000-HaRxL10 (OD$_{600\,nm}$ = 0.002) at 3 days post-infiltration. The 4$^{th}$ and 5$^{th}$ infiltrated leaves from three representative plants were shown. Scale bar, 2 cm. Data represent the mean ± SEM (*n* = 8 plants). The *p* values were calculated by two-way ANOVA followed by Holm-Šídák's multiple comparisons test. cfu, colony forming unit. This experiment was repeated three times with similar results. **c** Bacterial growth in wild-type (Col-0) and *che-2 Arabidopsis* leaves infiltrated with *Pst* DC3000 or *Pst* DC3000-HaRxL10$^{37-221}$ (OD$_{600\,nm}$ = 0.002) at 3 days post-infiltration. Data represent the mean ± SEM (*n* = 12 plants). The *p* values were calculated by two-way ANOVA followed by Holm-Šídák's multiple comparisons test. cfu, colony forming unit. This experiment was repeated three times with similar results. **d** Illustration of time-course RNA-seq experiment setting. Three-week-old wild-type (Col-0) and *che-2* plants growing under 12 h light/12 h dark conditions were infiltrated with *Pst* DC3000 or *Pst* DC3000-HaRxL10 (OD$_{600\,nm}$ = 0.002) respectively. Three

independent batches of samples were collected for RNA-seq. White bar, 12 h light. Black bar, 12 h dark. **e, f** Venn diagram (**e**) and statistical analysis (**f**) showing the significant overlap between HaRxL10-repressed genes and CHE-induced genes. The expected number of overlapped genes by random sampling is compared with the observed number of overlapped genes. The data are shown as mean ± SEM. The error bars were derived through random sampling based on hypergeometric distribution. The *p* value was calculated using Fisher's exact test. **g** Heatmap showing the relative gene expression profiles of 814 HaRxL10-repressed and CHE-induced genes in 3 biological replicates. hpi, hours post-infiltration. **h** Enriched GO terms among the 814 HaRxL10-repressed and CHE-repressed genes. FDR, false discovery rate. **i–k** Relative gene expression levels of *ACD6* (**i**), *CRK4* (**j**), and *PR2* (**k**) in the wild-type (Col-0) and *che-2 Arabidopsis* plants at 24 h after *Pst* DC3000 or *Pst* DC3000-HaRxL10 (OD$_{600\,nm}$ = 0.002) infection analysed by RT-qPCR with *UBQ5* as an internal control. The data are shown as mean ± SEM (*n* = 6, 2 independent experiments with 3 technical replicates). The *p* values were calculated by two-way ANOVA followed by Šídák's multiple comparisons test.

expression of *CCA1* compared to the wild-type plants (Fig. 6a). Period and phase of *CCA1* did not show a significant difference in the wild-type and HaRxL10 overexpression plants (Fig. 6a). The single time-point RT-qPCR of two independent HaRxL10 overexpression lines confirmed that overexpressing HaRxL10 suppressed the transcription of *CCA1* (Fig. 6b). However, Our RNA-seq data using *Pst* DC3000 TTSS to deliver HaRxL10 did not find the significant effect of HaRxL10 on the transcription of *CCA1* (Supplementary Fig. 15s). This discrepancy may be due to the higher HaRxL10 protein levels in HaRxL10 overexpression plants (Fig. 6a, b) than the wild-type plants upon *Pst* DC3000 TTSS-mediated HaRxL10 delivery (Supplementary Fig. 15s). The higher amount of HaRxL10 may trigger more pronounced *CCA1* suppression. Co-expression of HaRxL10 and CHE significantly suppressed the expression of *CCA1* in the wild-type *Arabidopsis* protoplasts, suggesting that CHE could mediate HaRxL10-triggered *CCA1* inhibition (Fig. 6c). To confirm the requirement of CHE in HaRxL10-mediated *CCA1* suppression, we performed RT-qPCR to analyse *CCA1* expression in the wild-type and *che-2 Arabidopsis* plants infiltrated with *Pst* DC3000 or *Pst* DC3000-HaRxL10 (Fig. 6d). HaRxL10 significantly repressed the expression of *CCA1* in the wild-type plants. The *che-2* plants displayed higher *CCA1* expression level than wild-type plants in the absence of HaRxL10, which is consistent with the previous finding that CHE negatively regulates *CCA1* expression[18]. HaRxL10 could still repress *CCA1* expression in the *che-2* plants, suggesting that other components may regulate HaRxL10-mediated *CCA1* suppression in this scenario. These results could not be explained by HaRxL10 interfering with CHE's binding to *CCA1* promoter, since CHE negatively regulates *CCA1* expression. Therefore, HaRxL10 may affect other components which regulate *CCA1* expression.

One possibility is that HaRxL10 represses the expression of positive regulators of *CCA1*[39], *TCP22*, in the *che-2* background (Supplementary Fig. 16a, d), leading to the HaRxL10-mediated suppression of *CCA1*. HaRxL10 overexpression plants displayed higher amplitude and average expression of *TOC1* compared to the wild-type plants, indicating that HaRxL10 could promote the expression of *TOC1*, which is a *CCA1* repressor (Supplementary Fig. 17). Period and phase of *TOC1* did not show a significant difference in the wild-type and HaRxL10 overexpression plants (Supplementary Fig. 17). This HaRxL10-mediated *TOC1* induction may also account for HaRxL10-mediated suppression of *CCA1*. In addition to transcriptional regulation, HaRxL10 may indirectly affect the expression of *CCA1* through protein interaction. We found that HaRxL10 interacted with CCA1 in yeast (Supplementary Fig. 1). A previous study showed that overexpression of CCA1 suppresses endogenous *CCA1* expression, suggesting that CCA1 protein could regulate its gene expression[40]. Therefore, this gene regulatory function of CCA1 protein may be altered upon interacting with HaRxL10, resulting in altered gene expression of *CCA1*.

Moreover, we analysed the expression of a widely used clock output gene, *CAB2* (*CHLOROPHYLL A/B-BINDING PROTEIN 2*, AT1G29920). Our results showed that the period of *CAB2* was shorter in HaRxL10 overexpression plants compared to the wild-type plants (Fig. 6e). HaRxL10 overexpression plants also displayed lower amplitude and average expression of *CAB2* (Fig. 6e). Phase of *CAB2* did not show a significant difference in the wild-type and HaRxL10 overexpression plants (Fig. 6e).

To assess the effects of HaRxL10 on the circadian-controlled genes (circadian oscillation correlation >0.7), we analysed the period, phase and amplitude of the circadian-controlled genes with robust rhythmic expression (circadian oscillation correlation >0.7) under at least one condition of our RNA-seq experiment illustrated in Fig. 5d. Our findings indicate that HaRxL10 alters the distributions of periods, phases and amplitudes of these circadian genes (Fig. 6f-h). Moreover, these effects were partially compromised in the *che-2* mutant compared to the wild-type, implying that HaRxL10 may partially relies on CHE to regulate the expression of rhythmic genes (Fig. 6f–h).

## HaRxL10 affects physiological processes in a CHE-dependent manner

The alterations in circadian clock gene expression may cause changes in some physiological processes. Leaf movement is a circadian-regulated physiological process[41]. Our results showed that over-expression of HaRxL10 in the wild-type plant delayed the phase of leaf movement while overexpression of HaRxL10 in *che-2* displayed a similar phase as *che-2* (Fig. 7a–c), suggesting that HaRxL10-triggered leaf movement phase delay is dependent on CHE. However, HaRxL10 shortened the period and reduced the amplitude of leaf movement in both wild-type and *che-2* backgrounds (Supplementary Fig. 18), indicating that HaRxL10 requires other unknown components to affect the period and amplitude of leaf movement.

The plant circadian clock regulates the amount of some metabolites, such as maltose[42]. We found that maltose levels showed a strong circadian rhythm in the wild-type plants, but this rhythm was disrupted in HaRxL10 overexpression plants (Fig. 7d). To study whether this HaRxL10-induced maltose oscillation perturbation was dependent on CHE, we measured maltose contents in the *che-2* and HaRxL10 overexpression in *che-2* plants. Both plants showed a strong circadian rhythm of maltose (Fig. 7e), suggesting that HaRxL10 requires CHE to perturb maltose oscillation. Hypocotyl growth is another well-characterised circadian-regulated physiological process. We found that overexpressing HaRxL10 in the wild-type plants lengthened the hypocotyl (Fig. 7f, g). However, HaRxL10 overexpression plants in the *che-2* did not show a hypocotyl length difference with the *che-2* plants (Fig. 7f, g), suggesting that HaRxL10-regulated hypocotyl growth was dependent on CHE.

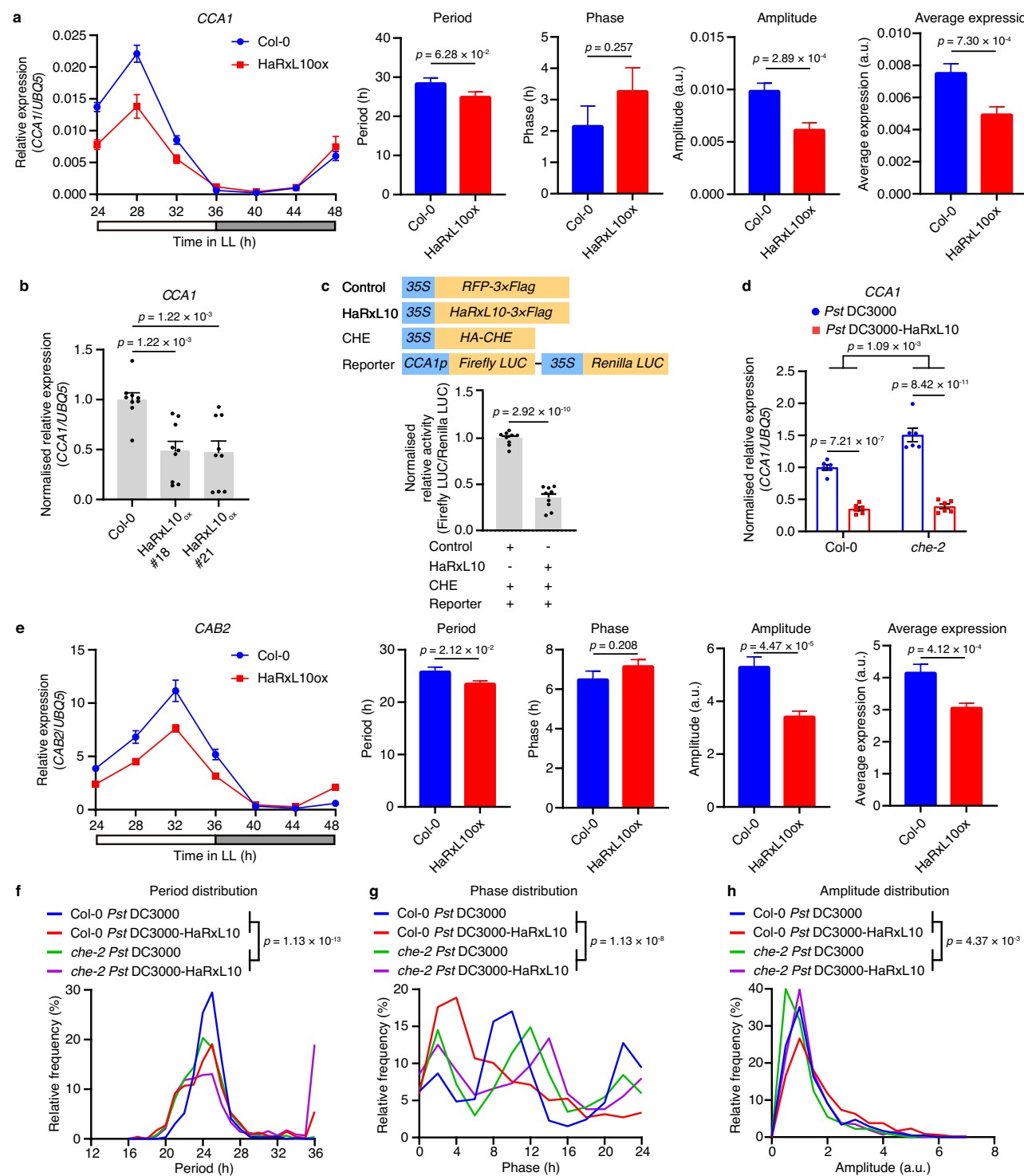

We found that overexpressing *HaRxL10* in the wild-type *Arabidopsis* significantly promoted flowering time under both long-day and short-day conditions (Fig. 7h–k). While the *che-2* mutant displayed an early flowering phenotype, overexpressing *HaRxL10* failed to further promote flowering time, implying that CHE is crucial for HaRxL10-mediated regulation of flowering (Fig. 7h–k). Furthermore, overexpression of HaRxL10 significantly stimulated the expression of *FLOWERING LOCUS T* (*FT*, AT1G65480) in wild-type plants but not in the *che-2* mutant under long-day conditions (Supplementary Fig. 19a). However, the *FT* transcripts could not be detected under short-day conditions and we therefore analysed the expression of the main flowering repressor *FLOWERING LOCUS C* (*FLC*,

AT5G10140) under short-day condition. *FLC* was significantly suppressed in *che-2*, consistent with the early flowering phenotype in *che-2* (Supplementary Fig. 19b). HaRxL10 could inhibit the expression of *FLC* in the wild-type background but not in the *che-2* background, providing additional evidence that HaRxL10 modulates flowering time in a CHE-dependent manner (Supplementary Fig. 19b). We noticed that HaRxL10 overexpression line #18 showed significantly lower expression of *FLC* than the line #21. It is probably due to the significantly higher level of HaRxL10 in line #18 than in line #21 (Supplementary Fig. 3). In summary, our results showed HaRxL10 could affect several physiological processes in a CHE-dependent manner.

**Fig. 6 | HaRxL10 affects the expression of central clock genes and circadian-controlled rhythmic genes. a** Relative expression levels of *CCA1* in 3-week-old wild-type (Col-0) and HaRxL10 overexpression (HaRxL10$_{ox}$ #18) *Arabidopsis* plants. Plants were grown under the 12 h light/12 h dark condition for 3 weeks and transferred to the constant light (LL) condition for 24 h. Samples were collected every 4 h under the LL condition and analysed by RT-qPCR with *UBQ5* as an internal control. White bar, subjective day. Grey bar, subjective night. The data are shown as mean ± SEM ($n = 6$, 2 independent experiments with 3 technical replicates). Period, phase, amplitude and average expression were calculated by nonlinear regression using a cosine wave. Data of period, phase, amplitude and average expression represent mean ± SEM (degree of freedom = 38). The *p* values were calculated by unpaired two-sided *t*-test with Welch's correction. a.u., arbitrary unit. **b** Expression levels of *CCA1* in the wild-type (Col-0) and two HaRxL10 overexpression (HaRxL10$_{ox}$) *Arabidopsis* plants. Samples were collected at ZT0 and analysed by RT-qPCR with *UBQ5* as an internal control and normalised by the expression in Col-0. The data are shown as mean ± SEM ($n = 9$, 3 independent experiments with 3 technical replicates). The *p* values were calculated by one-way ANOVA followed by Holm-Šídák's multiple comparisons test. **c** Dual-luciferase assay performed using *Arabidopsis* protoplasts transiently co-expressing different protein combinations and the reporter driven by *CCA1* promoter. The ratio of firefly luciferase and *Renilla* luciferase activities was calculated and normalised to the control. The data are shown as mean ± SEM ($n = 9$, 3 independent experiments with 3 technical replicates). The *p* value was calculated by two-sided unpaired Student's *t*-test. **d** Relative gene expression levels of *CCA1* in the wild-type (Col-0) and *che-2 Arabidopsis* plants at 24 h after *Pst* DC3000 or *Pst* DC3000-HaRxL10 (OD$_{600\ nm}$ = 0.002) infection analysed by RT-qPCR with *UBQ5* as an internal control. The data are shown as mean ± SEM ($n = 6$, 2 independent experiments with 3 technical replicates). The *p* values were calculated by two-way ANOVA followed by Šídák's multiple comparisons test. **e** Relative expression levels of *CAB2* in 3-week-old wild-type (Col-0) and HaRxL10 overexpression (HaRxL10$_{ox}$ #18) *Arabidopsis* plants. Plants were grown under the 12 h light/12 h dark condition for 3 weeks and transferred to the constant light (LL) condition for 24 h. Samples were collected every 4 h under the LL condition and analysed by RT-qPCR with *UBQ5* as the internal control. White bar, subjective day. Grey bar, subjective night. The data are shown as mean ± SEM ($n = 6$, 2 independent experiments with 3 technical replicates). Period, phase, amplitude and average expression were calculated by nonlinear regression using a cosine wave. Data of period, phase, amplitude and average expression represent mean ± SEM (degree of freedom = 38). The *p* values were calculated by unpaired two-sided *t*-test with Welch's correction. a.u., arbitrary unit. This experiment was repeated twice with similar results. **f–h** Distributions of period (**f**), phase (**g**), and amplitude (**h**) of rhythmic genes in the wild-type (Col-0) and *che-2 Arabidopsis* leaves after infiltrated with *Pst* DC3000 or *Pst* DC3000-HaRxL10. To derive the oscillation parameters including period, phase and amplitude, weighted non-linear regression analysis was performed to fit the time-course expression levels to a cosine wave with an intercept using RStudio. The estimated period, phase and amplitude were retrieved from the fitted cosine wave. The *p* value was calculated by two-way ANOVA.

GO analysis of HaRxL10-repressed genes revealed that HaRxL10 may also interfere with photosynthesis (Supplementary Fig. 20), suggesting that HaRxL10 may affect other plant developmental processes in addition to flowering time. We found that the plants overexpressing HaRxL10 in the wild-type displayed larger true leaves in the 2-week-old seedlings (Supplementary Fig. 21a, b). However, overexpressing HaRxL10 in *che-2* did not result in the leaf size difference compared to the *che-2* plants (Supplementary Fig. 21a, b), suggesting that this HaRxL10-mediated leaf size phenotype is CHE-dependent. Different from young seedlings, the 3-week-old HaRxL10 overexpression plants in both the wild-type and the *che-2* showed no obvious developmental phenotypes in leaf size and shape (Supplementary Fig. 21c, d).

### HaRxL10 interacts with multiple TCP proteins and affects TCP gene expression

The TCP family proteins shared a common TCP domain for DNA-binding function. Based on these TCP domain sequence variations, TCP family members were further grouped into two major classes, named class I and class II TCPs, respectively. Class I is formed by a group of relatively closely related proteins, whereas class II can be further subdivided into two clades also based on differences within the TCP domain. The CIN clade exemplified by *CINCINNATA* (*CIN*) of *Antirrhinum*, contains genes involved in lateral organ development and the CYC/TB1 clade includes genes mainly involved in the development of axillary meristems giving rise to either flowers or lateral shoots[26]. Since HaRxL10 interacted with CHE (also known as TCP21) and regulated its transcription, it is possible that HaRxL10 also interacts and affects the expression of other TCP genes.

Our Y2H assays showed that 10 of 24 TCPs could interact with HaRxL10 in yeast, including 7 Class I TCPs and 3 Class II TCPs (Supplementary Fig. 22). Different TCP genes show diverse daily expression patterns. Therefore, we performed a 24-hour time-course RT-qPCR experiment to analyse the effects of overexpressing HaRxL10 on the expression of several representative TCP genes. TCP7 belongs to Class I TCPs and shares the highest sequence similarity with CHE[26]. Moreover, TCP7 interacted with HaRxL10 in yeast (Supplementary Fig. 22). Our results showed that overexpression of HaRxL10 significantly affected the expression pattern of *TCP7* (Supplementary Fig. 23a). During the subjective day, HaRxL10 repressed *TCP7*, while during the subjective night, HaRxL10 promoted *TCP7*. TCP20, a Class I TCP, is a clock component[39]. Overexpression of HaRxL10 did not significantly affect the expression of *TCP20* (Supplementary Fig. 23b). TCP3 and TCP12 are both classified as Class II TCPs. Specifically, TCP3 is a member of CIN clade and TCP12 is a member of CYC/TB1 clade. HaRxL10 did not significantly affect the expression of *TCP3* but significantly repressed the expression of *TCP12* (Supplementary Fig. 23c, d).

Collectively, these results indicated that HaRxL10 could target multiple TCP proteins in addition to CHE and affect gene expression of several *TCP*s, rendering its unknown functions which require further investigation. Considering that TCP proteins play important roles in regulating plant growth and development, HaRxL10 may affect other physiological processes of plants not assessed in our study in a CHE-independent manner. Supporting this hypothesis, we found that HaRxL10 triggers a large-scale transcriptome reprogramming and the majority of HaRxL10 responsive genes are not regulated by CHE (Supplementary Fig. 24).

## Discussion

Our studies suggest that the oomycete pathogen *Hpa* employs the effector HaRxL10 to interfere with the function of the central clock component CHE through both transcriptional and post-transcriptional regulations. Direct interaction between HaRxL10 and CHE disrupts the CHE-ZTL interaction to stabilise the CHE protein but inhibits the binding of CHE to its target gene, such as *CHE*. HaRxL10 affects the expression of multiple central clock genes, including *CCA1*, *TOC1* and *CHE*. HaRxL10 inhibits SA-related positive defence genes as well as promotes negative defence regulator genes to suppress plant immunity in a CHE-dependent manner. HaRxL10 also regulates plant rhythmic processes, such as expression of genes related to circadian oscillations, leaf movement and maltose levels and promotes plant flowering and hypocotyl growth in a CHE-dependent manner (Fig. 7l). Our study unveiled the function of a previously uncharacterized pathogen effector and provided the evidence that a pathogen effector could directly target a central clock component.

In our study, oomycete effector HaRxL10 interferes with plant immunity in a CHE-dependent manner. Interestingly, a recent study showed that the effector of virus, NSs also interacts with CHE, suppressing the hormone-induced immunity[43]. These two independent studies suggest that CHE is vulnerable to attack by effectors from different species. However, the underlying molecular mechanisms of these two studies are different. The virus effector NSs maintains CHE-COI1 interaction, suppressing JA signalling[43]. Our results indicated that the oomycete effector HaRxL10 hijacks CHE to influence the

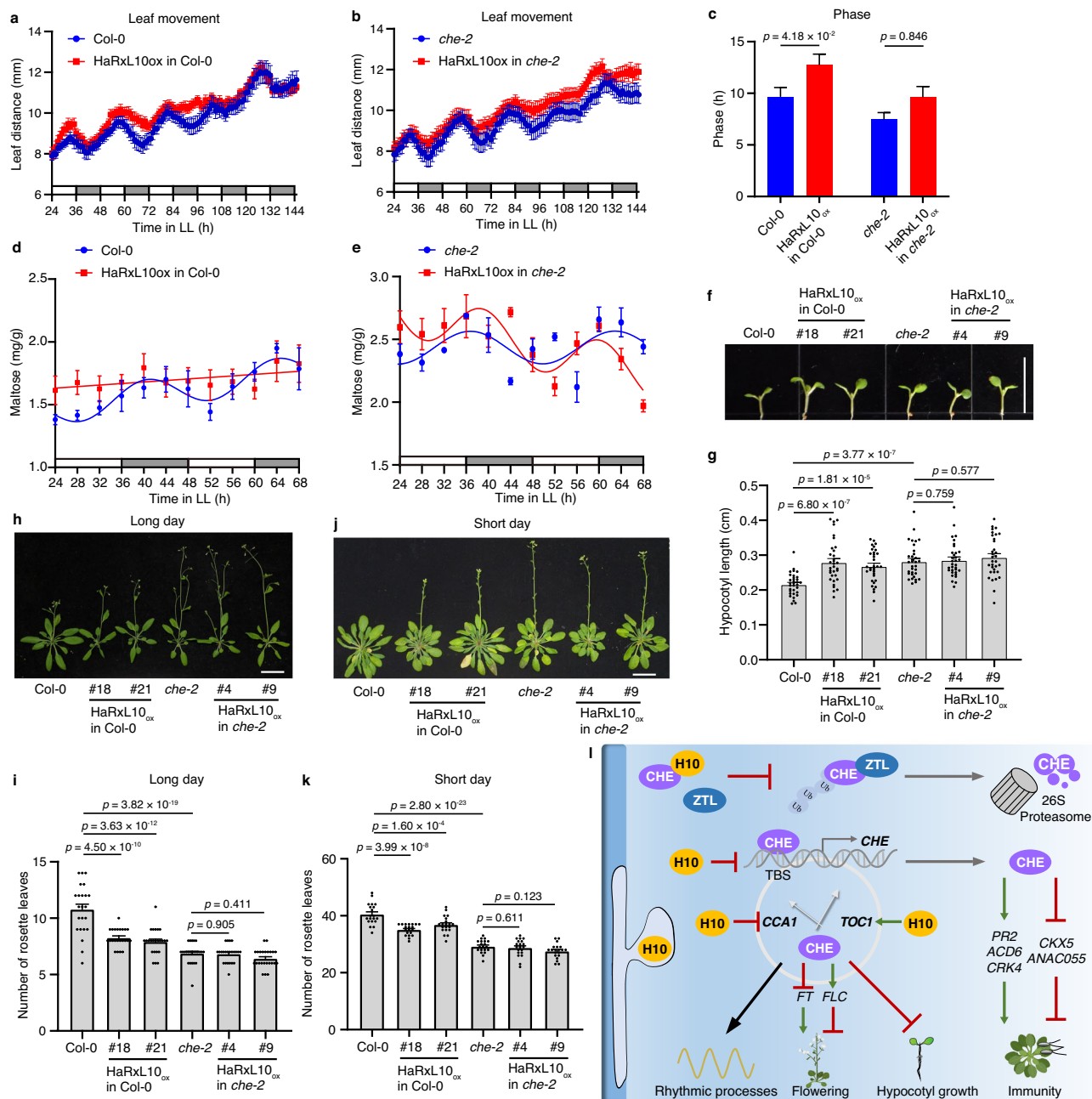

expression of multiple defence genes, including defence-promoting genes like *PR2*, *ACD6* and *CRK4* as well as defence-inhibiting genes like *CKX5* and *ANAC055*, leading to the suppression of SA-mediated immunity.

Moreover, our study revealed the impact of HaRxL10 on the expression of central clock genes and rhythmic genes. We found that HaRxL10 affects the expression of multiple central clock genes, such as *CHE*, *CCA1*, *TCP22* and *TOC1*. Our results suggested that HaRxL10-triggered suppression of *TCP22* requires CHE while HaRxL10 may affect other components to regulate the expression of *CCA1*. HaRxL10 also affects the expression of rhythmic gene *CAB2*. Roles of CHE in HaRxL10-mediated gene expression changes of *TOC1* and CAB2 were not analysed. Our time-course RNA-seq data indicated that CHE contributes to HaRxL10-triggered alteration of distributions of periods, phases and amplitudes of clock-regulated gene expression. Furthermore, we discovered that HaRxL10 depends on CHE to promote *Arabidopsis* hypocotyl growth and flowering as well as affect the oscillation of leaf movement and maltose abundance. Our findings

suggest that although HaRxL10 may have other plant targets, CHE is one of the linkers between pathogens and the plant circadian clock.

We employed two methods to investigate the effect of HaRxL10 on the host plant in this study. One method is *Pst* DC3000 TTSS-mediated natural delivery of HaRxL10. Introduction of HaRxL10 into *Pst* DC3000 enabled us to study the effect of HaRxL10 by using *Pst* DC3000 as a control pathogen. Several other studies have also utilised *Pst* DC3000 system to study the function of nonbacterial pathogen effectors[16,22–24]. However, this bacterial pathogen-mediated effector delivery requires light-dark cycles for successful pathogen infection. This diurnal condition may diminish the effects of HaRxL10 on the clock gene expression and circadian-regulated output pathways. The other method is the generation of transgenic plants overexpressing the interested pathogen effector. This method is more widely used and is appropriate for studying the effects of HaRxL10 on circadian-regulated physiological processes under constant light conditions as well as on host development. However, the ectopic overexpression of a pathogen effector in plants may not accurately reflect the effector's

**Fig. 7 | HaRxL10 affects physiological processes in a CHE-dependent manner.**
**a–c** Rhythm of leaf movement was measured in wild-type (Col-0) and HaRxL10 overexpression (HaRxL10$_{ox}$ #21 in Col-0) (**a**) as well as *che-2* and HaRxL10 overexpression (HaRxL10$_{ox}$ #4 in *che-2*) (**b**) *Arabidopsis* seedlings. Plants were grown under 12 h light/12 h dark conditions for 7 days and transferred to the constant light (LL) condition for 1 day. Pictures were then taken once an hour under the LL condition. The distance between two cotyledons was measured to represent the leaf movement. Grey bar, subjective night. The data are shown as mean ± SEM (*n* = 12 or 15 seedlings). Phase (**c**) was calculated by nonlinear regression using a cosine wave. Data represent the mean ± SEM. The *p* values were calculated by two-sided unpaired *t*-test with Welch's correction. **d, e** Maltose contents in 3-week-old wild-type (Col-0) and HaRxL10 overexpression (HaRxL10$_{ox}$ #21) *Arabidopsis* plants (**d**) or *che-2* and HaRxL10 overexpression (HaRxL10$_{ox}$ #4 in *che-2*) *Arabidopsis* plants (**e**). Plants were grown under the 12 h light/12 h dark condition for 3 weeks and transferred to the constant light (LL) condition for 24 h. Samples were collected every 4 h under the LL condition. White bar, subjective day. Grey bar, subjective night. The data are shown as mean ± SEM (*n* = 6 biological replicates from 2 independent experiments). Statistically significant oscillation of the data was determined by *F*-test comparing the nonlinear regression using a cosine wave or a straight line (*p* < 0.05). **f, g** Representative images (**f**) and statistical analysis of hypocotyl length (**g**) in the 7-day-old seedlings of wild-type (Col-0), *che-2* and HaRxL10 overexpression (HaRxL10$_{ox}$) in different backgrounds (Col-0 or *che-2*). Plants were grown under the 12 h light/12 h dark condition for 4 days and transferred to the constant light (LL) condition for 3 days. Photos were taken and hypocotyl length was measured. The *p* values were calculated by one-way ANOVA followed by Holm-Šídák's multiple comparisons test. The data are shown as mean ± SEM (*n* = 33 seedlings). **h** Flowering phenotypes under long-day conditions (16 h light/8 h dark). Scale bar, 4 cm. **i** The number of rosette leaves at flowering under long-day conditions (16 h light/8 h dark). The *p* values were calculated by one-way ANOVA followed by Holm-Šídák's multiple comparisons test. The data are shown as mean ± SEM (*n* = 23 plants). **j** Flowering phenotypes under short-day conditions (8 h light/16 h dark). Scale bar, 4 cm. **k** The number of rosette leaves at flowering under short-day conditions (8 h light/ 16 h dark). The *p* values were calculated by one-way ANOVA followed by Holm-Šídák's multiple comparisons test. The data are shown as mean ± SEM (*n* = 20 plants). **l** Working model of *Hpa* effector HaRxL10 (H10) manipulating the plant circadian clock to interfere with immunity and growth. HaRxL10 directly interacts with the central clock component CHE to inhibit ZTL-mediated CHE protein degradation. The interaction between HaRxL10 and CHE suppresses the binding of CHE to the cis-element TCP-binding site (TBS) of target gene promoters, such as *CHE*. Meanwhile, HaRxL10 affects the expression of multiple other central clock genes, including *CCA1* and *TOC1*. CHE inhibits flowering by promoting the expression of *FLC* and suppressing the expression of *FT*. HaRxL10 inhibits SA-related positive defence genes (*PR2*, *ACD6* and *CRK4*) as well as promotes defence negative regulator genes (*CKX5* and *ANAC055*) to suppress plant immunity in a CHE-dependent manner. HaRxL10 also regulates plant rhythmic processes, such as expression of genes related to circadian oscillations, leaf movement and maltose levels, and promotes plant flowering and hypocotyl growth in a CHE-dependent manner. The red lines represent inhibition. The green arrows represent induction. The black arrow (pointing from CHE protein to rhythmic processes) refers to the regulatory role.

natural function probably due to its higher abundance in the over-expression plants. In summary, these two methods have both advantages and disadvantages. Therefore, the findings revealed by these two methods may apply under certain circumstances.

CHE regulates phytohormones-mediated immunity and flowering time, while CHE is vulnerable to attack by pathogen effectors. Our Y2H and BiFC results showed that CHE$^C$ interacted with HaRxL10 while CHE$^{\Delta C}$ did not interact with HaRxL10, suggesting that the C-terminal of CHE mediates protein interaction with HaRxL10 (Fig. 2c, d). Previous studies showed that the TCP domain functions in DNA-binding[44]. Therefore, it is possible that CHE$^{\Delta C}$, which contains the TCP domain, maintains its DNA-binding ability and gene regulatory function but cannot be recognised by HaRxL10. Future functional and phenotypic studies of this CHE mutant may provide insights into whether it can evade effector attacks while maintaining its regulatory function in plant growth and defence.

## Methods
### Plasmid constructs
Coding sequences of *CHE* (AT5G08330), *ZTL* (AT5G57360), *TOC1* (AT5G61380), *PRR3* (AT5G60100), *PRR5* (AT5G24470), *PRR7* (AT5G02810), *PRR9* (AT2G46790), *CCA1* (AT2G46830), *LHY* (AT1G01060), *ELF3* (AT2G25930), *ELF4* (AT2G40080), *LUX* (AT3G46640), *TCP1* (AT1G67260), *TCP2* (AT4G18390), *TCP3* (AT1G53230), *TCP4* (AT3G15030), *TCP5* (AT5G60970), *TCP6* (AT5G41030), *TCP7* (AT5G23280), *TCP8* (AT1G58100), *TCP9* (AT2G45680), *TCP10* (AT2G31070), *TCP11* (AT2G37000), *TCP12* (AT1G68800), *TCP13* (AT3G02150), *TCP14* (AT3G47620), *TCP15* (AT1G69690), *TCP16* (AT3G45150), *TCP17* (AT5G08070), *TCP18* (AT3G18550), *TCP19* (AT5G51910), *TCP20* (AT3G27010), *TCP22* (AT1G72010), *TCP23* (AT1G35560) and *TCP24* (AT1G30210) were amplified from *Arabidopsis* Col-0 cDNA. The sequence of *HaRxL10* was synthesised by GENEWIZ. Coding sequence of HaRxL10$^{SP}$ (signal peptide of HaRxL10) was directly amplified from synthesised full-length *HaRxL10* for signal peptide secretion assay. The signal peptide was deleted for all the functional studies of HaRxL10, which is ΔSP-HaRxL10. We referred to ΔSP-HaRxL10 as HaRxL10 in short for convenience. Truncated *CHE* and truncated *HaRxL10* were directly amplified from their full-length templates. The target genes were cloned into destination vectors by Gateway cloning (Invitrogen, 11789 & 11791) or seamless cloning (ZOMANBIO, ZC231) according to the manufacturer's protocols. See detailed primer and vector information in Supplementary Data 1.

### Plant materials and growth conditions
Col-0 plants were used as wild-type plants in this study. The *che-2*[18] mutant was previously described. Transgenic plants of *3SS:YFP-CHE*, *3SS:YFP-HaRxL10* (HaRxL10$_{ox}$) were generated by *Agrobacterium*-mediated floral dip method[45]. *Arabidopsis* seeds were surface sterilised and treated at 4 °C in darkness for 3 days. Seeds were then placed on 1/2 Murashige and Skoog (MS) plates with 2% (w/v) sucrose and 0.8% (w/v) agar and grown in a chamber at 22 °C under 12 h light/12 h dark for 7 days. Seedlings were then transferred to soil and grown at 22 °C under 12 h light/12 h dark for 2 weeks. Differences in growth conditions in some phenotypic experiments were described in the related methods.

### Signal peptide secretion assay in yeast
The signal peptide (SP) sequence of HaRxL10 was predicted by SignalP-5.0 (https://services.healthtech.dtu.dk/services/SignalP-5.0/). The coding sequences of the signal peptide and the flanking two amino acids of HaRxL10 were constructed into the pSUC2 vector and then transformed into yeast strain YTK12 through the lithium acetate method. After 2 days of growth on the CMD-W medium (0.67% YNB (yeast nitrogen base without amino acids), 0.074% Trp DO supplement, 2% sucrose, 0.1% glucose, 2% agar), yeast cells carrying fused pSUC2 vectors were screened on the YPRAA medium (1% yeast extract, 2% peptone, 2% raffinose, 2% agar, 2 μg/μL antimycin A). Quantification of the activities of sucrose invertase was performed using 0.1% TTC (2,3,5-Triphenyltetrazolium Chloride) as substrate according to the previous report[46,47]. Specifically, yeast colonies successfully grew on CMD-W medium plates were cultured in CMD-W liquid medium at 30 °C for 36 h. 1.5 mL of yeast culture was centrifuged at 12,000 *g* for 1 min and washed twice with ddH$_2$O. The yeast cells were resuspended in 750 μL of ddH$_2$O, 250 μL of 10 mM acetic acid-sodium acetate buffer (pH = 4.7) and 500 μL of 10 % sucrose solution. The suspension was incubated at 37 °C for 10 min and then centrifuged at 12,000 *g* for 1 min. 100 μL of the supernatant was taken and co-incubated with 900 μL of 0.1% TTC at room temperature for 5 min or longer until the red colour could be observed. The OD value of each sample was measured at 484 nm using a spectrophotometer.

## Secretion of nonbacterial effector by *Pst* DC3000 type-three secretion system

The first 15 amino acids of AvrPto (AvrPto[1-15 aa]) are sufficient to direct the secretion of the foreign effector protein[48]. Hence, the synthesised AvrPto promoter and the DNA fragment encoding AvrPto[1-45 bp] were first cloned into the pDSK519 vector. The DNA fragment of ΔSP-HaRxL10-3×Flag or ΔSP-HaRxL10[37-221]-3×Flag were then cloned and inserted into the downstream of AvrPto[1-45 bp] to generate constructs of *ProAvrPto:AvrPto[1-45 bp]-ΔSP-HaRxL10-3×Flag* (pDSK519-HaRxL10) or *ProAvrPto:AvrPto[1-45 bp]-ΔSP-HaRxL10[37-221]-3×Flag* (pDSK519-HaRxL10[37-221]). Constructs of pDSK519-HaRxL10 or pDSK519-HaRxL10[37-221] were respectively transformed into *Pst* DC3000 strain by electroporation to generate strains of *Pst* DC3000-HaRxL10 and *Pst* DC3000-HaRxL10[37-221] for bacterial infection assays.

### Bacterial infection assay

Three-week-old *Arabidopsis* or 4-week-old *N. benthamiana* leaves were used for syringe-infiltration with bacteria strains ($OD_{600\,nm} = 0.002$ for *Arabidopsis*; $OD_{600\,nm} = 0.0002$ for *N. benthamiana*). Leaf discs were sampled using a hole punch with 7.0 mm in diameter at 3 days post-infiltration (dpi). To quantify bacterial growth, the samples with 2 leaf discs each were placed in a 96-well plate containing 500 μL 10 mM $MgCl_2$ per well and ground in a grinder. Samples were then diluted 10, $10^2$, $10^3$, $10^4$, $10^5$, and $10^6$ times and 10 μL of each dilution were taken and placed on the KB plates with appropriate antibiotics. The number of bacterial colonies was counted after 2 days (for *Arabidopsis*) or 5 days (for *N. benthamiana*) of growth at 30 °C. The bacterial growth was quantified using the following formula:

$$\log_{10}\left(\frac{cfu}{leaf\ disc}\right) = \log_{10}\left(colonies \times 10^{dilution} \div 10 \times 500 \div 2\right)$$

### Phytophthora sojae infection assay

*Phytophthora sojae* was cultured for 4–5 days on V8 juice agar medium to prepare for infection. Zoospore suspensions were obtained by washing the mycelium of *P. sojae* with sterile water 3–4 times. Leaves of 4-week-old *Arabidopsis* plants were inoculated with *P. sojae* zoospore suspensions (1000 zoospores) in a plastic tray and maintained with high humidity in a dark climate chamber at 25 °C. Infected leaves were photographed and collected for genomic DNA extraction followed by qPCR to analyse the amount of *AtActin* and *PsActin* at 3 days after inoculation.

### Protein purification

The vectors used for protein expression were pET-42a (GST-tag), pQLink (MBP-His-tag) and pET28a (His-tag). All proteins were expressed in *E. coli* (BL21 DE3 strain) cells with induction by 0.4 mM isopropyl *β*-D-1-thiogalactopyranoside (IPTG, Biopped, I6070) at 16 °C for 24 h.

For GST-tag protein purification, bacteria cells were collected and lysed by sonication in GST lysis buffer (50 mM Tirs-HCl pH 7.4, 100 mM NaCl, 1 mM EDTA, 0.5 mg/mL Lysozyme, 1 mM PMSF, 1 mM DTT, 0.5% Triton X-100). After centrifugation at 16,800 *g* for 30 min, the supernatant was further filtered through a 0.22 μm filter, incubated with GST-beads (Thermo Fisher, 16100) for 2 h and then washed using GST wash buffer (50 mM Tirs-HCl pH 7.4, 100 mM NaCl, 1 mM EDTA) for 5 times. The proteins were finally eluted using GST elution buffer with glutathione (50 mM Tirs-HCl pH 7.4, 100 mM NaCl, 10 mM glutathione).

For MBP-tag protein purification, bacteria cells were collected and lysed by sonication in MBP column buffer (20 mM Tirs-HCl pH 7.4, 0.2 M NaCl, 1 mM EDTA). After centrifugation at 16,800 *g* for 30 min, the supernatant was further filtered through a 0.22 μm filter, incubated with amylose resin (Sangon, C500096) for 1 h and then washed using MBP column buffer for 5 times. The proteins were finally eluted using MBP elution buffer (20 mM Tirs-HCl pH 7.4, 0.2 M NaCl, 1 mM EDTA, 10 mM maltose monohydrate).

For His-tag protein purification, bacteria cells were collected and lysed by sonication in His buffer A (50 mM Tris, 1 M NaCl, pH 7.4). After centrifugation at 55,814 *g* for 30 min, the supernatant was further filtered through a 0.22 μm filter loaded onto a Ni-NTA column (GE Healthcare). The proteins were washed and eluted using His buffer B (50 mM Tris, 1 M NaCl, 500 mM imidazole, pH 7.4). The eluted proteins were concentrated using an ultrafiltration column and finally eluted into 1× PBS buffer (Beyotime, C0221A).

### Pull-down assay

MBP-His and MBP-His-recombinant proteins were mixed with GST-recombinant protein in equal amounts (about 5 μg), respectively. Each sample was made up to 100 μL using MBP column buffer (20 mM Tirs-HCl pH 7.4, 0.2 M NaCl, 1 mM EDTA). Samples of 20 μL were saved as input. The remaining samples were incubated with 1 mL MBP column buffer and 20 μL amylose resin (Sangon, C500096) for 2-3 h at 4 °C. The input and MBP pull-down samples were added to SDS loading buffer and boiled at 100 °C for 5 min, collected by centrifugation and subjected to Western blot assays using α-MBP (NEB, E8032S) and α-GST (TRANSGEN, HT601) antibodies.

For the pull-down competition assay, equal amounts of recombinant proteins MBP-His-CHE and GST-His-ZTL (1.2 μmol each) were mixed and divided into three parts. The first part was made up to 120 μL using MBP column buffer. The second and the third parts were made up to 120 μL by adding 8 or 20 μmol of His-HaRxL10 recombinant proteins, respectively. Samples of 30 μL were saved as input. The remaining samples were incubated with 1 mL MBP column buffer and 20 μL amylose resin for 8-12 h at 4 °C. The samples were then washed using MBP column buffer 5 times. The input and MBP pull-down samples were added to SDS loading buffer and boiled at 100 °C for 5 min, collected by centrifugation and subjected to Western blot assay by α-MBP, α-GST and α-His (TRANSGEN, HT501) antibodies.

### Co-immunoprecipitation (Co-IP) assay

Constructs of *35S:HA-CHE*, *35S:YFP*, and *35S:YFP-HaRxL10* were transformed into *Agrobacterium tumefaciens* GV3101, respectively. Four-week-old *N. benthamiana* leaves were injected with different combinations of bacterial suspension for transient protein expression. After 36 h, tobacco buffer (10 mM MES, pH 5.6, 10 mM $MgCl_2$, 200 μM acetosyringone) containing 100 μM MG132 was injected into tobacco leaves. Samples were taken after 12 h of MG132 treatment, with YFP + HA-CHE as the control group and YFP-HaRxL10 + HA-CHE as the experimental group. For the control group, GFP lysis buffer (10 mM Tris-HCl pH 7.5, 150 mM NaCl, 0.5 mM EDTA, 0.5% Nonidet P40 Substitute, 40 μM MG132) was added to the samples after liquid nitrogen grinding. The samples were lysed for 2 h, and total protein was extracted. The supernatant was collected after centrifugation at 14,000 *g* for 2 min, and 30 μL was taken as input. The remaining samples were added to GFP dilution buffer (10 mM Tris-HCl pH 7.5, 150 mM NaCl, 0.5 mM EDTA) to a final volume of 1 ml for immunoprecipitation (IP). For the experimental group, nuclear extraction buffer (20 mM Tris-HCl pH 7.4, 25% glycerol, 20 mM KCl, 2 mM EDTA, 2.5 mM $MgCl_2$, 250 mM sucrose, 1 mM DTT, 1 mM PMSF) was added to enrich the nuclei after centrifugation at 1,500 *g* for 10 min. The nuclei were washed three times with NBRT (20 mM Tris-HCl pH 7.4, 25% glycerol, 2.5 mM $MgCl_2$, 0.2% Triton X-100) and then lysed with 100 μL nuclear lysis buffer (50 mM Tris-HCl, pH 8.0, 10 mM EDTA, 1% SDS, 0.1 mM PMSF, 40 μM MG132) for 2 min. The supernatant was collected after centrifugation at 14,000 *g* for 2 min, and 30 μL was taken as input. The remaining samples were added to GFP dilution buffer to a final volume of 1 mL for IP. 20 μL of GFP beads (ChromoTek, gta-20) were added to 1 mL of GFP dilution buffer. After centrifugation at 2,500 *g* for 3 min, the supernatant was discarded. This step was repeated twice,

and the samples were divided into two equal parts. Each part was added to its respective sample. After overnight incubation at 4 °C, the samples were centrifuged at 2500 $g$ for 3 min, and the supernatant was discarded. The samples were washed five times with GFP washing buffer (10 mM Tris-HCl pH 7.5, 150 mM NaCl, 0.5 mM EDTA, 0.05% Nonidet P40 Substitute). Both the input and GFP-IP samples were mixed with 4×SDS loading buffer and boiled at 100 °C for 10 min for Western blot analysis. α-GFP antibody (Abmart, M20004M) was used to detect YFP and YFP-HaRxL10, and α-HA antibody (CWBIO, CW0092) was used to detect HA-CHE.

## Western blot

Three-week-old *Arabidopsis* plants were used for protein extraction. After being ground in liquid nitrogen, 0.12 g of the resulting powder was added to Lysis Buffer (20 mM Tris-HCl pH 7.4, 25% glycerol, 20 mM KCl, 2 mM EDTA, 2.5 mM MgCl$_2$, 250 mM sucrose, 1 mM DTT, 1 mM PMSF) for lysis. The solution was filtered twice through gauze, and the supernatant was collected by centrifugation at 4 °C 1500 $g$ for 10 min. The nuclear pellet was washed three times with NBRT (20 mM Tris-HCl pH 7.4, 25% glycerol, 2.5 mM MgCl$_2$, 0.2% Triton X-100) and resuspended in 60 µL of NBRT. Then, 20 µL of 4×SDS loading buffer was added, mixed, and boiled at 100 °C for 10 min. Western blotting was performed using a 10% resolving gel (10% acrylamide solution, 375 mM Tris-HCl pH 8.8, 0.1% SDS, 0.1% APS, 0.033% TEMED) and a 5% stacking gel (5% acrylamide solution, 125 mM Tris-HCl pH 6.8, 0.1% SDS, 0.1% APS, 0.033% TEMED) with MOPS buffer (50 mM MOPS, 50 mM Tris, 0.1% SDS, 1 mM EDTA) as the electrophoresis buffer. The protein was transferred onto a PVDF membrane using a transfer apparatus (Pyxis, SPJ-T20S). After transfer, the membrane was blocked with 5% skim milk in TBST solution (50 mM NaCl, 20 mM Tris-HCl, pH 8.0, 0.1% Tween 20) for 30 min. Primary antibody was then added and incubated overnight at 4 °C with shaking. After removing the primary antibody, the membrane was washed three times with TBST solution and then incubated with the secondary antibody for 1.5 h at 4 °C. After removing the secondary antibody, the membrane was washed three times with TBST solution and visualized using the chemiluminescent substrate (Thermo Scientific, 34580).

## Yeast two-hybrid (Y2H) assay

Y2H assays were performed following the manufacturer's protocol (Clontech). For the initial Y2H screen, 50 *Hpa* effector genes were synthesised by GENEWIZ, constructed into the pGBKT7-GW vector as bait and then transformed to the yeast strain Y187. Eleven *Arabidopsis* central clock genes (*CCA1, LHY, CHE, TOC1, PRR3, PRR5, PRR7, PRR9, ELF3, ELF4, LUX*) were constructed into the pGADT7-GW vector and then transformed to the yeast strain AH109. Yeast mating was performed between Y187 and AH109 strains. Colonies were cultured on synthetic dropout medium without leucine and tryptophan (SD/-Leu-Trp) for positive yeast transformant selection and synthetic dropout medium without leucine, tryptophan, histidine and adenine (SD/-Leu-Trp-His-Ade) for protein interaction selection by the reporter gene *HIS3*.

To test individual protein-protein interaction, the two genes encoding interested proteins were constructed into the pGADT7-GW and pGBKT7-GW vectors respectively and then transformed into the yeast strain AH109. Specifically, full-length and truncated CHE as well as other 23 *Arabidopsis* TCPs were constructed into the pGADT7-GW vector. Full-length and truncated HaRxL10 as well as ZTL were constructed into the pGBKT7-GW vector. SD/-Leu-Trp medium was used for positive yeast transformant selection. SD/-Leu-Trp-His-Ade medium was used for the selection of protein interaction by the reporter gene *HIS3*.

## Bimolecular fluorescence complementation (BiFC) assay

The two genes of interest were constructed into the XY105 (for cYFP-tagged fusion protein) and XY106 (for nYFP-tagged fusion protein) vectors respectively and then transformed into *Agrobacterium tumefaciens* (GV3101). The successfully transformed *A. tumefaciens* cells and P19 were incubated overnight at 30 °C. Bacteria were then collected by centrifugation, washed twice with tobacco buffer (10 mM MES, pH 5.6, 10 mM MgCl$_2$, 200 µM acetosyringone), and finally resuspended in tobacco buffer to the concentration of OD$_{600\,nm}$ = 1. Four-week-old *N. benthamiana* leaves were injected with different combinations of bacterial suspension for transient protein expression. After 2 days, fluorescence was observed under a Zeiss 780 laser scanning confocal microscope and the images were analysed using ImageJ.

## Subcellular localisation by tobacco transient expression system

*Agrobacterium tumefaciens* strain GV3101 containing the *35S:HaRxL10-CFP* or *35S:RFP-CHE* construct was cultured overnight at 30 °C. Agrobacteria were pelleted after centrifugation at 4000 $g$ for 15 min. The bacterial cells were then diluted to OD$_{600\,nm}$ of 1.0 in 10 mM MES (pH 5.6), 10 mM MgCl$_2$ and 200 µM acetosyringone. After incubation at room temperature for 2–3 h, the Agrobacteria suspension was infiltrated into 4-week-old *N. benthamiana* leaves using a 1 mL needless syringe. After 2 days, the infiltration areas were imaged under a Zeiss 780 laser scanning confocal microscope.

## Cell-free protein degradation assay

Three-week-old Col-0 plants were used for protoplast isolation. Leaf peels were obtained using the tape-*Arabidopsis* sandwich method[49]. The leaf peels were digested in an enzyme solution containing 0.4 M mannitol, 20 mM KCl, 20 mM MES (pH 5.7), 1.5% cellulase R10 (RPI Corp.), 0.4% macerozyme R10 (Fisher Scientific), 10 mM CaCl$_2$, and 0.1% BSA for 2 h at room temperature. After digestion, an equal volume of W5 solution containing 154 mM NaCl, 125 mM CaCl$_2$, 5 mM KCl and 2 mM MES pH 5.7 was added. The resulting protoplasts were filtered through a 70 µm cell strainer and pelleted by centrifugation at 100 $g$ for 5 min. The protoplast concentration was adjusted to approximately $3 \times 10^5$ cells/mL using W5 solution. The protoplasts were incubated on ice for at least 1 h. The protoplast concentration was adjusted to approximately $3 \times 10^5$ cells/mL using MMG solution containing 0.4 M mannitol, 15 mM MgCl$_2$, and 4 mM MES at pH 5.7. Plasmid DNA (50 µg), 1 mL protoplasts, and 1 mL PEG solution (40% PEG 4000, 100 mM CaCl$_2$ and 0.2 M mannitol) were mixed. This transfection mixture was incubated at room temperature in the dark for 20 min. The transfection was stopped by dilution of the mixture with 5 mL W5 solution. After removing the supernatant by centrifugation at 100 $g$ for 2 min, the protoplasts were resuspended in 5 mL W5 solution and incubated in the dark at 22 °C overnight. The protoplasts were collected the next day by centrifugation and incubated in 250 µL IP buffer (50 mM HEPES pH 7.5, 150 mM NaCl, 1% Triton X-100, 10% glycerol, EDTA-free protease inhibitor) on ice for 2-4 h. Protein concentrations were determined by BSA method and adjusted to the same. Each group of proteins was divided into two parts for subsequent treatment with DMSO (control) or MG132 (50 µM). Samples were mixed well, divided into 6 tubes, and then incubated at 22 °C. The samples were finally collected at 0, 0.25, 0.5, 1, 1.5, and 2 h post-treatment for Western blot analysis with an α-HA (CWBIO, CW0092) antibody.

## Yeast three-hybrid (Y3H) assay

HaRxL10 or YFP (used as negative control) was constructed driven by the methionine deficiency-induced promoter ($P_{Met}$) and ZTL or TOC1 were fused with BD domain of the pBridge vector to obtain BD-ZTL-$P_{Met}$-HaRxL10, BD-ZTL-$P_{Met}$-YFP, BD-TOC1-$P_{Met}$-HaRxL10, and BD-TOC1-$P_{Met}$-YFP. Each of these constructs was then co-transformed with pGADT7-CHE into the *Saccharomyces cerevisiae* Gold strain through the lithium acetate method. Synthetic dropout medium without leucine, tryptophan and methionine (SD/-Leu-Trp-Met) was used for positive yeast transformant selection. A synthetic dropout medium without leucine, tryptophan, methionine and histidine (SD/-Leu-Trp-

Met-His) was used for the selection of protein interaction by the reporter gene *HIS3*. 3-AT was used to inhibit the self-activation in yeast.

## Microscale thermophoresis (MST) assay

MST assays were performed using a Monolith NT.115 instrument (NanoTemper Technologies). The purified GST-ZTL proteins were fluorescently labelled according to the manufacturer's procedure (NanoTemper Technologies, RED-NHS 2nd Generation) and kept in MST buffer (50 mM Tris-HCl pH 7.4, 150 mM NaCl, 10 mM MgCl$_2$, 0.1% Pluronic F-127, 0.01‰ Tween-20). For ZTL-CHE binding assay, 50 nM labelled GST-ZTL proteins were mixed with serially diluted His-CHE proteins (29750 nM, 14875 nM, 7437.5 nM, and 1.82 nM) and incubated at room temperature for 30 min. The samples were then loaded into standard capillaries (NanoTemper Technologies, MO-K022) and measured at 25 °C using 40% LED power and low MST power. The MST data was analysed using Affinity Analysis v.2.3 and GraphPad Prism 8.0. For the competition assay, 5 µM purified His-HaRxL10 proteins were mixed with the labelled GST-ZTL and His-CHE proteins and subjected to the MST assay.

## RNA extraction and RT-qPCR

Ten-day-old *Arabidopsis* seedlings (for *FT* gene analysis), 5-week-old *Arabidopsis* leaves (for *FLC* gene analysis) or 3-week-old *Arabidopsis* leaves (for other experiments) were collected for RNA extraction using TRNzol Universal reagent (TIANGEN, DP424) according to the manufacturer's protocols. Complementary DNA synthesis (HiScript III 1st Strand cDNA Synthesis Kit, Vazyme, R312) and qPCR (SYBR qPCR Master Mix, Vazyme, Q711) were performed according to the manufacturer's protocols. All primer sequences used for RT-qPCR are listed in Supplementary Data 1.

## Yeast one-hybrid (Y1H) assay

The promoters of *CHE*, *CKX5*, *ANAC055*, and *CCA1* were constructed into pLacZi expression vector respectively. The linearized pLacZi-promoter construct (referred to pLacZi-CHEp, pLacZi-CKX5p, pLacZi-ANAC055p, or pLacZi-CCA1p) was transferred into YM4271 yeast strain using the lithium acetate method. CHE was constructed into the pGADT7-GW vector. The pGADT7-GW-CHE or pGADT7-GW (negative control) construct was transferred into Y187 yeast strain. The mating of the yeast containing pLacZi-promoter and the yeast containing pGADT7-GW-CHE or pGADT7-GW would generate yeast progenies that contained both promoter and protein of interest and could grow in Synthetic dropout liquid medium without leucine and uracil (SD/-Leu-Ura). The β-galactosidase activity was quantified by the ONPG assay[18]. The successfully mating yeast cells were cultured in 5 mL YPDA liquid medium to OD$_{600 nm}$ around 0.8-1.0. The exact value of OD$_{600 nm}$ was recorded. Yeast cells were collected from 1 mL of yeast cell cultures by centrifugation at 12,000 $g$ for 2 min and resuspended in 150 µL Z-buffer (60 mM Na$_2$HPO$_4$, 40 mM NaH$_2$PO$_4$, 10 mM KCl, 1 mM MgSO$_4$). After being frozen in liquid nitrogen and thawed at room temperature twice, the yeast suspension was added 850 µL Z-buffer containing 600 µg ONPG (Beyotime, ST429) and incubated at 30 °C until the yeast suspension turned yellow. This colour change usually could be observed within 0.5 to 5 h. After centrifugation at 12,000 $g$ for 2 min, OD$_{420 nm}$ of the supernatant was measured. OD$_{420 nm}$/OD$_{600 nm}$ represents the β-galactosidase activity.

## ChIP-qPCR

Two grams of 7-day-old *35S:YFP-CHE Arabidopsis* seedlings were collected in 1% formaldehyde for vacuum treatment for 15-20 min and then subjected to vacuum treatment for 5 min in the presence of 125 mM glycine. Samples were washed twice with ddH$_2$O. The dried samples were then ground in liquid nitrogen. To obtain nuclei, samples were lysed in extraction buffer (20 mM Tris-HCl pH 7.4, 25% glycerol, 20 mM KCl, 2 mM EDTA, 2.5 mM MgCl$_2$, 250 mM sucrose, 1 mM DTT,

1 mM PMSF) and then centrifuged at 1000 $g$ at 4 °C. The pellet was washed 3–5 times with NBRT (20 mM Tris-HCl pH 7.4, 25% glycerol, 2.5 mM MgCl$_2$, 0.2% Triton X-100). The obtained nuclei were lysed in 300 µL of lysis buffer (50 mM Tris-HCl pH 8.0, 10 mM EDTA, 1% SDS, 0.1 mM PMSF, Protease inhibitors) by sonication. After centrifugation at 13,800 $g$ for 5 min at 4 °C, 10% of supernatant was saved for input and the remaining part was equally divided into two parts (control and experimental group) and then diluted at 10 folds in dilution buffer (1.1% Triton X-100, 1.2 mM EDTA, 16.7 mM Tris-HCl pH 8.0, 167 mM NaCl). GFP antibody (Abcam, ab290) was added to the experimental group and incubated overnight. Both control (HA antibody added; Roche, 11867423001) and experimental groups were then mixed with Protein A/G beads (Thermo Fisher, 20421) and washed sequentially with low salt wash buffer (150 mM NaCl, 0.1% SDS, 1% Triton X-100, 2 mM EDTA, 20 mM Tris-HCl pH 8.0), high salt wash buffer (500 mM NaCl, 0.1% SDS, 1% Triton X-100, 2 mM EDTA, 20 mM Tris-HCl pH 8.0), LiCl wash buffer (0.25 M LiCl, 1% NP-40, 1% sodium deoxycholate, 1 mM EDTA) and TE buffer (10 mM Tris-HCl pH 8.0, 1 mM EDTA) for two times. Input, experimental and control groups were incubated with a 50:1 volume ratio of 5% Chexlex-100 solution and DNase-free RNase A (10 mg/mL) for 10 min at 37 °C, followed by the addition of 10 µL 0.5 M EDTA, 20 µL 1 M Tris-HCl pH 6.5 and 2 µL 10 mg/mL Protease K for an additional 10-min incubation at 37 °C and 10-min incubation at 95 °C. For each group, ddH$_2$O was first added to make the volume to 400 µL and 400 µL of phenol: chloroform: isoamyl alcohol (25:24:1 v/v) were then added, mixed well and centrifuged to obtain the top layer. The top layer solution was mixed with 700 µL of ethanol, 2 mM glycogen, and 10 mM acetic acid. After centrifugation, the precipitate was washed twice with 70% ethanol and finally dissolved in 100 µL of ddH$_2$O. qPCR experiments were performed using SYBR qPCR Master Mix (Vazyme, Q711).

## Dual-luciferase assay

The promoters of *CHE* and *CCA1* were cloned into the vector pGreen0800II. The promoter of *ICS1* was cloned into the vector LZ004. The CDS of CHE was constructed into the pEG201 vector. The CDS of RFP, HaRxL10 and HaRxL10$^{37-221}$ were cloned into the vector LZ0102. Constructs were expressed in tobacco leaves or *Arabidopsis* protoplasts.

Four-week-old tobacco (*Nicotiana benthamiana*) leaves were co-infiltrated with *Agrobacteria* GV3101 carrying pEG201-gene/LZ0102-gene and pGreen0800II/LZ004-gene promoter, respectively. Tobacco leaves were sampled 2 days post-infiltration and the enzyme activities of firefly luciferase and *Renilla* luciferase were detected using the Dual-Luciferase Reporter Assay System Kit (Promega, E1910) following the manufacturer's protocol.

*Arabidopsis* protoplasts were prepared as described in the cell-free protein degradation assay in this paper. For protoplast transfection, 10 µg of each plasmid were mixed well with 200 µL of protoplasts and 200 µL of PEG solution (40% PEG 4000, 100 mM CaCl$_2$ and 0.2 M mannitol) and then incubated at room temperature in dark for 20 min. The transfection was stopped by the addition of 1 mL W5 solution. After removing the supernatant by centrifugation at 100 $g$ for 2 min, the protoplasts were washed in 1 mL W5 solution and finally resuspended in 1 mL W5 solution and incubated in dark at 22 °C overnight. The protoplasts were collected the next day by centrifugation and enzyme activities were measured by Dual-Luciferase Reporter Assay System Kit (Promega, E1910) following the manufacturer's protocol.

## Electrophoretic mobility shift assay (EMSA)

Synthesised 5′-terminal biotin-labelled forward and reverse single-stranded oligonucleotides were annealed in the annealing buffer (10 mM Tris, 1 mM Na$_2$EDTA, 50 mM NaCl) to generate double-stranded probes. The EMSA experiments were performed with 5 pmol protein and 50 fmol double-stranded probes using LightShift Chemiluminescent EMSA Kit (Thermo Fisher Scientific) according to

the manufacturer's instructions. The concentration of the competitive probes without the biotin label was 200-fold of that of the biotin-labelled probes. Concentrations of competitive probes and mutated competitive probes were 10 μM.

## RNA-seq and data analysis

Three-week-old wild-type (Col-0) and *che-2 Arabidopsis* leaves were infiltrated with *Pst* DC3000 or *Pst* DC3000-HaRxL10 (OD$_{600 \text{ nm}}$ = 0.002). Leave samples were taken every 4 h for 2 days. Three batches of samples were used for the following total RNA extraction by TRNzol Universal reagent (TIANGEN, DP424). Total RNA was then used for library construction and RNA sequencing (Novogene). The obtained raw reads underwent cleaning with fastp[50] (v0.21.0) to eliminate sequencing adapters and low-quality reads. Subsequently, clean reads were aligned to the *Arabidopsis thaliana* genome, employing TopHat[51] (v2.1.1). Mapped reads were then counted using StringTie[52] (v2.1.5) to generate a raw count matrix representing gene expression.

To identify differentially expressed genes (DEGs), the R packages limma[53] and edgeR[54] were used for analysing the raw count matrix, following the described protocols[55]. Genes with q-values < 0.05 were considered as DEGs. To identify rhythmic genes, weighted non-linear regression analysis was performed to fit the time-course expression levels to a cosine wave with an intercept using RStudio. Genes with circadian oscillation correlation >0.7 were considered rhythmic. For gene enrichment analysis, Fisher's Exact tests were performed to obtain *p* values and random sampling was performed to obtain expected overlapped genes. For functional annotation enrichment analysis of genes, the DAVID[56] tool was employed, and the results were visualized using RStudio. See RNA-seq data in the Supplementary Data 2.

## Measurement of endogenous SA

The analysis of endogenous SA was performed based on the method reported previously with some modification[57]. About 150 mg of ground powder of fresh plant tissues was extracted overnight in methanol containing $^2H_4$-SA as internal standards. The crude extracts were further purified by loading onto the Oasis MAX SPE cartridge, which was sequentially preconditioned with methanol, water and ammonia solution (0.02%). After the samples loading into the cartridge, the cartridge was sequentially washed with ammonia solution and methanol. Finally, SA was eluted with methanol containing 10% formic acid and analysed on a liquid chromatography-tandem mass spectrometry system comprising an ACQUITY UPLC (Waters) and Qtrap 5500 system (AB SCIEX) equipped with electrospray ionization (ESI) source. The separation of analytes was achieved on a Waters ACQUITY UPLC BEH C18 column (100 × 2.1 mm i.d., 1.7 μm). The multiple reaction monitoring (MRM) transitions were as follows: SA 137.1 > 93.0, $^2H_4$-SA 141.1 > 97.0. Three biological replicates were analysed for each treatment.

## Leaf movement analysis

Leaf movement assay was modified from previously described method[41]. *Arabidopsis* seeds were surface sterilised and treated at 4 °C in dark for 3 days and then grown on 1/2 MS plates with 2% (w/v) sucrose and 0.8% (w/v) agar under 12 h light/12 h dark conditions (110–120 μmol photons m$^{-2}$ sec$^{-1}$) for 7 days and transferred to constant light conditions (25 μmol photons m$^{-2}$ sec$^{-1}$) for 1 day. Pictures were then taken once an hour under the constant light condition. Distance between two cotyledons were measured using Image J. Period, phase and amplitude of leaf movement were derived through nonlinear regression using a cosine wave.

## Maltose contents measurement

*Arabidopsis* seeds were surface sterilised and treated at 4 °C in dark for 3 days and then grown under the 12 h light/12 h dark condition (110-120 μmol photons m$^{-2}$ sec$^{-1}$) for 21 days and transferred to the constant

light condition (110–120 μmol photons m$^{-2}$ sec$^{-1}$) for 1 day. About 0.2 g of samples from the 3rd, 4th, 5th and 6th leaves were collected every 4 h under constant light condition and frozen in liquid nitrogen. Samples were extracted in 1 mL extraction solution and the maltose content was determined using the Maltose Assay Kit (Grace Bio-technology, G0566W) following the manufacturer's instructions. Maltose contents were calculated by the following formula:

$$Maltose(mg/g) = \frac{V \times [A_{test} - (A_{control} - A_{blank})] \times D}{(A_{standard} - A_{blank}) \times W}$$

V (volume of extraction solution, mL), A (Absorbance at OD$_{520 \text{ nm}}$, Abs), W (Weight of sample, g), D (Dilution factor).

## Flowering time analysis

*Arabidopsis* seeds were surface sterilised and treated at 4 °C in dark for 3 days and then grown under either long-day (16 h light/8 h dark) or short-day (8 h light/16 h dark) conditions, with a light intensity of 110–120 μmol photons m$^{-2}$ sec$^{-1}$. The time of flowering was determined by the number of rosette leaves at the time of the first flower opening[58].

## Reporting summary

Further information on research design is available in the Nature Portfolio Reporting Summary linked to this article.

## Data availability

Raw RNA sequencing data have been deposited in the Sequence Read Archive (SRA) of the National Centre for Biotechnology Information (NCBI) under project PRJNA1015292. All other data generated in this study are provided in the Source Data file or from the corresponding author upon request. Source data are provided with this paper.

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

## Acknowledgements

We thank Xinnian Dong of Duke University for providing *che-2*, CHE_CE (*CHEp:CHE-4×Myc* in *che-2*), and CHE-HA (*CHEp:CHE-3×HA* in *che-2*) seeds; Caoxi Luo of Huazhong Agricultural University for providing YTK12 yeast strain and pSUC2 vector; Ligeng Ma of Capital Normal University for providing Gold yeast strain and pBridge vector; Xiu-Fang Xin of CAS Centre for Excellence in Molecular Plant Sciences for providing pDSK519 vector; Susheng Song of Capital Normal University for providing pGreen0800II vector; Yi Li of Peking University for providing the Monolith NT.115 instrument. We thank the Core Facilities of College of Life Sciences at Capital Normal University for assistance with confocal imaging and Core Facilities of the School of Life Sciences and the National Centre for Protein Sciences at Peking University in Beijing, China, for assistance with large-scale protein purification. Funding was as follows. National Natural Science Foundation of China 31970283 (M. Z.)**;** National Natural Science Foundation of China 32370288 (M. Z.); Beijing Nova Programme of Science and Technology Z191100001119027 (M. Z.); Capital Normal University (M. Z.); Support Project of High-level Teachers in Beijing Municipal Universities in the Period of 14th Five-year Plan BPHR20220114 (M. Z.); State Key Laboratory for Gene Function and Modulation Research, School of Life Sciences, Peking University (W. W.); National Natural Science Foundation of China 31970641 (W. W.); Centre for Life Sciences (W. W.); National Natural Science Foundation of China 31721004 (Yuanchao W.); National Key Research and Development Program of China National Key Research and Development Program of China 2023YFA0914800 (J. C.); Fundamental Research Funds for Central Non-profit Scientific Institution 110243160002002 (T. S.).

## Author contributions

M.Z. conceived the study. M.F. performed EMSA and dual luciferase assay and measured hypocotyl growth and flowering time. Y.Z. performed cell-free protein degradation assay and signal peptide secretion assay. M.F., Y.Z., X.Z., Y.X., J.S. and T.S. performed the bacterial infection assays. M.F. and Y.Z. performed pull-down assay and BiFC assay. M.F., Y.Z., X. Z., K.Y. and L.M. performed Y2H assay. M.F. and Y.Z. performed Y3H assay. M.F. and Y.X. performed Y1H assay. M.F. and Y.D. performed MST assay. M.F., Y.Z. and Yu W. performed subcellular localisation confocal imaging, Co-IP and ChIP-qPCR. M.F and X.Z. performed RNA extraction and qPCR. M.F. and Y.S. measured maltose levels. Z.C. and Yuanchao W. performed *Phytophthora sojae* infection assay. M.F., Y.Z., and H. L. produced the constructs and generated transgenic plants. M.F. and Y.D. purified the recombinant proteins. W. W. and X. W. performed RNA-seq data analysis. Y.S. performed protein 3D structure predictions. S.C. and J.C. performed endogenous SA measurement assay. Yu W. and M.F. performed leaf movement assays. M.Z. wrote the manuscript with inputs from all co-authors.

## Competing interests

The authors declare no competing interests.
