## [Transparent Peer Review file · Nature Communications]

A novel pathogen effector HaRxL10 hijacks the circadian clock component CHE to perturb both plant development and immunity

Corresponding Author: Dr Mian Zhou

Version 0:

Reviewer comments:

Reviewer #1

(Remarks to the Author)

The manuscript by Mengyao Fu et al. is a nice contribution to our understanding of the interplay between immunity and the core clock machinery in plants. Several studies have previously revealed a complex interplay between the circadian network and immune responses, by showing that clock mutant plants display alterations in immunity as well as by revealing that pathogen infections alters expression of core clock genes. However, the precise mechanisms of these interactions were not known, particularly how pathogen effectors might modulate the circadian network to enhance pathogenicity.

By screening for potential interactions between *Hyaloperonospora arabidopsidis* (Hpa) effectors and Arabidopsis core clock proteins, the authors found that the HaRxL10 effector directly interacts the CCA1 Hiking Expedition (CHE) protein. CHE is a TCP transcription factor that represses the expression of CIRCADIAN CLOCK ASSOCIATED 1 (CCA1) 1 gene. CCA1 encodes a MYB transcription factor that acts at the core of the plant circadian network, repressing the expression of other core clock genes and hundreds of clock outputs. Both CHE and CCA1 have previously been implicated in mediating/modulating immune responses in Arabidopsis.

After the initial Y2H screen, the authors confirmed that HaRxL10 acts as a secreted effector that enhances susceptibility to *Pseudomonas* infection. They also found that HaRxL10 overexpression enhances growth of biotrophic fungi such as *Phytophthora sojae*. They also showed that HaRxL10 inhibits CHE degradation by inhibiting its interaction with ZTL, an E3 Ubiquitin ligase, while at the same time binding of HaRxL10 to CHE reduces CHE ability to bind DNA and regulate gene expression. Therefore, exposure of plant cells to HaRxL10 appears to inhibit CHE activity. In agreement with this, the authors found that exposure of Arabidopsis plants to HaRxL10 through its expression as a secreted protein by *Pseudomonas syringae* pv tomato DC3000 (Pst), enhances bacterial virulence and this effects requires CHE, as it is not observed in che mutants.

A genome wide transcriptome analysis further confirmed that HaRxL10 triggers transcriptomic changes in Pst infected plants that are similar to those observed in che mutants. Indeed, they found that expression of some defence genes was altered upon HaRxL10 exposure in a CHE dependent way. The authors also found that an acute exposure to the HaRxL10 effector triggered alterations in expression of a few clock genes such as RVE4, TCP22 and LWD1, while constant exposure to HaRxL10 through overexpression from a constitutive promoter caused a reduction in CCA1 expression. These effects were associated with an apparent global change in periodicity and phase of clock regulated genes, as results from an evaluation of these parameters in a circadian RNA-seq expression time-course. Finally, the authors also found CHE dependent effects of HaRxL10 on flowering time regulation, supporting the idea that HaRxL10 affects immunity and development in Arabidopsis plants through its inhibitory effects on CHE activity.

The manuscript is well written. The results are very interesting, the experiments have been well conducted, and most of the conclusions are supported by their data. I have only a few concerns:

While it is clear that HaRxL10 can act as an effector that alters immunity through its interactions with CHE when secreted by Pst DC3000, whether the same effect occurs in its native system, i.e. Hpa, remains to be determined. I understand the reason for using this heterologous system is likely the result in difficulties in altering the function of HaRxL10 in Hpa. The

authors should explicitly comment on this and should briefly explain the rationale of using Pst to test the effects of secreted HaRxL10 as an effector. I understand there have been reports describing the use of this approach before, but this needs to be mentioned and explained.

My major concern though is related to the effect of HaRxL10 on the plant circadian clock. While the authors indicate that period and phase of clock regulated gene expression are altered, I could not find (maybe a missed it) an explanation on how these parameters were evaluated. Furthermore, looking at the individual clock gene oscillatory expression patterns shown in supplementary figure 13, I do not see that the data supports a significant effect on rhythms of core clock gene expression. Most of the circadian patterns of expression of core clock genes are apparently unaffected in period and/or phase in that supplementary figure. The presumed effects of HaRxL10 on expression of clock genes is associated with genes that have only moderate (RVE4) or non (LWD1, TCP22) circadian regulated expression, and only overall mRNA levels of these genes are affected. The authors should clearly comment on this in the text. They should also adjust their conclusions regarding effects of HaRxL10 on the clock to their results. A simple experiment to evaluate whether HaRxL10 affects circadian function is to analyze the rhythms of leaves movements or gene expression of a few clock genes in HaRxL10 overexpressing plants compared to those of WT plants. If period and phase of leaf movement rhythms or clock gene expression are not altered in those plants, the authors should adjust their conclusions. My impression is that all their findings are interesting and well supported, but that the effect of HaRxL10, although dependent on CHE, are not mediated by changes in clock function but rather through more direct effects of CHE on immune and developmental responses as clock outputs.

Reviewer #2

(Remarks to the Author)

This was a very lengthy and comprehensive manuscript describing the interaction of a putative oomycete effector protein, HaRxL10 with the CCA1 HIKING EXPEDITION (CHE) protein of Arabidopsis. The authors suggest that HaRxL10 targets CHE which leads to a suppression of plant immunity and early flowering. Unfortunately, the authors do not provide evidence that HaRxL10 is indeed expressed by *Hyaloperonospora arabidopsidis* (Hpa) during infection, nor that it plays a role in Hpa virulence as an effector. It is not clear why HaRxL10 was chosen to study, nor why CHE was chosen to focus on rather than any of the other 8 TCP transcription factors that were demonstrated to interact with HaRxL10. The interactions of HaRxL10 with the other TCP factors is insufficiently discussed. The work will be of interest to plant pathologists and those interested in plant pathogens. The results are generally interpreted in a very simplistic, direct way without due consideration of wider effects relating to consequences of overexpression, redundancy, or pleiotropy.

Aims and introduction

The manuscript is lacking in a clearly articulated hypothesis or research question. It is not clear why the authors chose the HaRxL10 to investigate or why they thought it may interact with clock components. Although the authors state on p3 lines 7-10 that they “conducted a yeast-two hybrid (Y2H) screening between predicted Hpa RXLR effectors and Arabidopsis central clock components” these data are not shown. How many Hpa RXLR effectors were screened and where are the data for the other effectors and Arabidopsis central clock components? There are Y2H data for interactions between HaRxL10 and TCP family members in shown in Supplementary Figure 15.

Results

Fig 1c. It is not clear what the leaves represent here – are they replicates from multiple plants, or leaves from a single representative plant? Either way, why have these been selected to be shown – they do not match the n=16, and do not appear to represent the same developmental leaf from different plants.

Similarly, for Fig 1g where 10 leaves are shown per line – does this match experimental n (not stated) or representative leaves, or leaves from one plant?

It would be useful to provide some information about the phenotype of the Arabidopsis stable transgenic lines overexpressing HaRxL10 prior to infections, and include pictures of leaves that have not been challenged with *Phytophthora sojae* in Fig 1g for comparison.

The interaction of AD-CHEc with HaRxL10 in bimolecular interaction assay indicates that the interaction is not only in the nucleus (Fig2d), but this is not discussed. Please provide some comments on this finding and its implications. At the top of page 9, HaRx21 appears – presumably a typo. Please correct.

The authors state that they wish to investigate the effect of HaRxL10 on the transcriptional regulation function of CHE, and chose to investigate CHE as a representative downstream gene. Why? What about the other genes that are transcriptionally regulated by CHE, such as CCA1? This would be of interest when later discussing effects of HaRxL10 on the clock.

I find the data presented in Fig 4d difficult to extrapolate to the infection of a plant by Hpa. The level of HaRxL10 when overexpressed in plants does not necessarily relate to biologically relevant levels during an infection. This limitation should be acknowledged. Furthermore, what are the effects of overexpression of HaRxL10 on the expression of other clock genes and TCP genes?

The same comments about leaves shown in Fig 1 apply to those in Fig 5 a.

What was the time of day of inoculation of leaves with in Pst DC3000 with and without HaRxL10: CHE is expressed with a peak at mid-day – is that important in the assay? Can the authors comment on the fold-difference in cfu per leaf disc in che-2 vs Col-0 in Pst DC3000 without HaRxL10 (Fig 5.b and c)?

The authors state that HaRxL10 affects the expression of central clock genes. I am not convinced of the 'centrality' of the genes are stated to be affected. More importantly, the effect on TCP22 is not seen until 36 hpi, while the effects are detectable for LWD1 earlier, at 16-20 hpi, but appear to be similar at all other time points, but these subtleties are not commented on. What are the consequences for the clock and temporal regulation of defences and metabolism, if any?

I am not totally convinced by the data that the statement that "HaRxL10-triggered repression of RVE4 and LWD1 is partially dependent on CHE" is correct, partly because of the measurement at 24 hpi without a reference to the ZT of infection.

There are a number of typos on p17: line15 found instead of fine, line 25 regulating instead of regulate, line 30 expressions instead of expression

It is not clear what time of day samples were collected for the data shown in Fig 6 g.

While overexpression of HaRxL10 affects the levels of CCA1 expression, it seems overly speculative to extrapolate this finding to the role of HaRxL10 to a natural infection. This limitation must be acknowledged.

I do not think that the authors can conclude that HaRxL10 globally affects circadian-controlled genes as there is no Col-0 mock infection expression analyses for comparison.

Where all experiments carried out in LD, and if so how were samples for RNA analyses collected during the dark periods? Why were the times ZT20 in long days and ZT0 in SD chosen for measuring transcript levels? Was this why FT was not detected in Short days? Why were expression levels measured only in 10 day old plants?

Discussion

Some of the data have been interpreted without proper consideration of the limitations of the experimental design. Many of the data were collected from experiments with heterologously overexpressed genes in bacteria or plants. This may not reflect the activity of the putative effector in an Hpa infection.

The effects of the HaRxL10 on the other TCP proteins it was demonstrated to interact with have not been considered. The roles of other TCPs may underpin some of the effects seen.

The conclusion that CHE may be a candidate for genetic modification to improve disease resistance may be premature.

Methods

Bacterial assay and Phytophthora sojae assays should specify which leaf of the plants were chosen for inoculation.

Matching developmental stage for infections can minimise variation.

Bacteria are not in solution, but in suspension – this needs correction throughout.

Y2H assay is missing information for all the TCP genes used.

For the BiFC assay, were Agrobacterium tumefaciens cells really washed in ddH₂O prior to resuspension in tobacco buffer?

Type on p27 line 40 Agrobacteria should be Agrobacterium or simply A. tumefaciens

Reviewer #3

(Remarks to the Author)

Recent studies have established the crosstalk between the circadian clock and plant innate immunity. This manuscript reports a new mechanism underlying such a crosstalk that an oomycete effector HaRxL10 directly targets Arabidopsis central clock component CCA1 HIKING EXPEDITION (CHE) in suppressing CHE's function in circadian clock, defense, as well as flowering. Interestingly, HaRxL10 stabilizes CHE at the protein level but inhibit its transcription and transcriptional function. How to resolve such a paradox is unclear. A strong link of HaRxL10 to clock and defense regulation via CHE remains to be fully established. A large body of data were presented in this ms. The authors also made many useful tools to study the interaction between HaRxL10 and CHE. One set of such tools is stable transgenic Arabidopsis seedlings expressing CHEp:CHE-3×HA in che-2 (CHE-HA) with or without 35S:YFP-HaRxL10 in CHE-HA (two independent homozygous T3 lines) as presented in Figure 2e. These lines were used to show that HaRxL10 stabilizes CHE at the protein level. In addition to using different expression system, this same system could be further used to show whether HaRxL10 indeed suppresses CHE transcriptional function, including suppressing CHE target genes for their RNA expression (including CHE own transcription) and binding of CHE to target gene promoters.

additional specific comments:

1. Fig. 3: Fig. 3e showed an almost complete abolishment of CHE and ZTL interaction in the presence of HaRxL10. However, Fig. 3d did not clearly show this point. In absence of 3AT, HaRxL10 did not affect the interaction between CHE and ZTL, compared with the control YFP. In the presence of 3AT, the interaction of CHE and ZTL appeared to be reduced in the presence of HaRxL10 or YFP. In light of the result shown in Fig 3e, perhaps His-HaRxL10 should be included in Fig. 3a

to clearly demonstrate HaRxL10 disruption of CHE and ZTL interaction, a likely cause for CHE stabilization in the presence of HaRxL 10.

2. YFP-HaRxL10 was overexpressed in Col-0 and che-2 mutant. Two YFP-HaRxL10/Col-0 lines, #18 and #21 were used in this report. While #18 showed much higher expression of YFP-HaRxL10 than #21, both lines were reported to have reduced CHE transcripts and early flowering time (Fig 4d and Fig 7), compared with Col-0. However, line #21 showed wt-like bacterial growth (Fig. s9c) and even higher than Col-0 expression of the flowering repressor gene, FLC (Fig. 7). Such discrepancies might be due to other reasons than just expression difference of HaRxL10 between the two lines. Perhaps some other lines could be examined to clarify the point.

3. ICS1 was previously shown as a direct transcriptional target of CHE and its expression was suppressed in the che-2 mutant in the absence of pathogen challenge. Such a basal level expression pattern was unclear in Fig 5m. Fig 5m showed that ICS1 expression was induced by Pst DC3000 and such an induction was suppressed by HaRxL10. Interestingly, in the presence of HaRxL10 and Pst DC3000, che-2 expressed much higher ICS1 than just Pst DC3000-infected che-2. Does this mean that such HaRxL10-controlled ICS1 expression is actually CHE independent?

4. Fig 6h and i: in 6h, are there endogenous CHE proteins in the protoplasts used? If this is the case, the data shown in these two panels should be relatively similar for HaRxL10 suppression of CCA1 expression, which is not the case when the two panels are compared. In addition, the RNAseq data did not support this point that HaRxL10 suppresses CCA1 expression (See comment 5).

5. The data from RNAseq and qRT-PCR appear to be inconsistent in several places. For example, CCA1 is a known CHE target. RNAseq data showed almost no effect of HaRxL10 on CCA1 expression (Fig.S13s) yet Fig 6g showed a clear suppression by HaRxL10. Such a discrepancy can also be found for TCP22, LWD1, CKX5, ANAC055, and SIF (Fig 6 and S11) when the 24 hpi time point is compared for data from RNAseq and qRT-PCR. Instead of hpi, perhaps time of day should be used in RNAseq and qRT-PCR to allow a better comparison. Related to the RNAseq data, please clarify how to calculate the average expression of genes in a time course (Page 17 line 3). In addition, it would be great if data from che-2 could be included in Fig. S13. This is particularly important for the analysis of CCA1 expression influenced by CHE.

6. Page 17 line 34: It is unclear what these circadian-controlled genes with robust rhythmic expression (circadian oscillation correlation > 0.7) are. Expression of most main clock genes did not change after infection with Pst DC3000 or Pst DC3000-HaRxL10 (Fig S13). Please use a table to show such genes. With only a 44 hr-time course in the RNAseq experiment, it might be difficult to accurately assess clock parameters, such as period, amplitude, and phase.

7. Page 23 paragraph 3: the data should be presented in the result section.

Reviewer #4

(Remarks to the Author)

This article demonstrates that the Hpa effector HaRxL10 stabilizes the Arabidopsis central circadian protein CHE and inhibits ZTL binding to CHE. The accumulation of CHE inhibits its binding to downstream gene promoters, leading to reduced expression of circadian rhythmic genes, SA-responsive genes, and flowering-related genes, and enhanced pathogenesis. The paper presents new findings on the relationship between the circadian cycle and plant immunity. The quality of experiments and logic are reliable, but further demonstration is needed for some points. Additionally, the writing needs improvement in terms of English language usage.

Major

1. The sentences should be more concise, and the Material and Method section should be explained more thoroughly if necessary. There are several incomplete sentences in the text, such as "YFP, 33.2 kD. YFP-HaRxL10, 52.3 kD. HA-CHE, 28.3 kD".

2. The last paragraph in the introduction seems incomplete. It would be better to add a couple of sentences for closing remarks.

3. More description is needed to understand why RxL10 was chosen among the 130 effectors of Hpa. Providing detailed procedures for Y2H screening is helpful in the Material and Method section.

4. The paragraph and experimental results focused on the CHE and CHE-mediated circadian cycle regulation. However, there is no explanation about CHE or its related mechanism. To make the importance of RxL10 function clear, it is recommended to provide a brief description of CHE-mediated regulation in any section.

5. The authors mention the TCP domain of CHE, but do not provide an explanation about it. Additionally, an explanation of the CHE gene or protein structure may be helpful in understanding its interaction with the effector.

6. Please provide a description of TCP. Additionally, provide the importance of RxL10 function in terms of inhibiting CHE binding and interfering with the interaction between CHE and ZTL. The discussion section did not mention the novel aspect of RxL10 functioning as a virulence effector by manipulating the circadian clock.

7. It is unclear whether the SP sequences of effectors used in this paper were present or removed. It appears that RxLR was used exclusively, but this should be accurately described and indicated in the figures.

8. HaRxL10 stabilizes CHE protein levels, and inhibiting its expression hinders CHE-binding to TBS. The text does not provide enough information to determine if HaRxL10 specifically binds to the TBS-binding site on CHE's surface. The structures between CHE and HaRxL10 are not discussed.

9. Figure 5m suggests that if che-2 cannot produce functional endogenous CHE, HaRxL10 may stabilize another component in the compromised CHE scenario. If the hypothesis that HaRxL10 compromises CHE in che-2 is correct, a specific part of CHE in che-2 may undergo a null mutation. Therefore, investigating a structural correlation between the HaRxL10 docking site on CHE is crucial for a comprehensive understanding.

10. Page 5, line 25-30, the text mentions that there is no change in localization with effector + CHE co-expression. However, there is a slight change where the nuclei ring becomes nucleolar. It is necessary to mention this change.

Minor

- On page 9, line 6, change 'suppl. Fig 3a' to 'Fig 3'.
- On page 9, line 1, the effector's name is incorrect.
- The full name of TCP needs to be described.
- In Fig 5a, it is difficult to distinguish between healthy and infected samples based on phenotype. A clearer image needs to be replaced.
- Page 17, line 27, mediaed > mediated.

Version 1:

Reviewer comments:

Reviewer #1

(Remarks to the Author)

I thank the authors for addressing all my comments and most of my concerns. I appreciate that they added new experimental information for this. The new data showing alterations on circadian rhythms in maltose levels in plants overexpressing HaRxL10, and its dependence on CHE, are quite clear.

I am still not very convinced of a significant GLOBAL effect of HaRxL10 on clock regulated gene expression. The authors are correct in mentioning that it is not simple to address global effects of clock gene expression under diurnal conditions, so they evaluated circadian expression for a core clock gene, CCA1, and an output gene, CAB2, under constant light conditions following circadian clock entrainment. Here the authors indicate that there is an effect on CCA1 mRNA circadian amplitude, but not period or phase, while for CAB2 they do observe differences in period and phase. Based on the data provided, which is associated with only the second day in constant light, the difference resulting in a period difference in CAB expression can only be seen in a single timepoint (the last one on their graph). Since it is a short timecourse, the reliability of a minor period effect is difficult to evaluate. The fact that no effect is observed for CCA1 period or phase in this same period of time and experimental condition, suggests that the effect of HaRxL10 on clock regulated gene expression is far from being strong and global. The authors should attenuate their claims associated with global effects of HaRxL10 on circadian oscillations of gene expression.

Reviewer #2

(Remarks to the Author)

Thank you for addressing my comments and providing extra experimental information. I think that the title "HaRxL10 hijacks the circadian clock component CHE to perturb both plant development and immunity" is appropriate, but that the suggestion that HaRxL10 "globally affects circadian controlled genes" or that its interaction with CHE affects "the period, phase and amplitude of circadian rhythmic genes globally" is not supported.

The period predictions based on 44h of gene expression data are not sufficiently powered to be reliable. Although leaf movement data are provided in the response to the reviewers, the methods for these are not stated that I could find, nor are they presented in the revised manuscript or supplementary material. I cannot see how the period differences in leaf movement were arrived at, and certainly by visual inspection they are not apparent. It is not clear why these were not included in the revised version. The maltose rhythms are reported, but are difficult to interpret as such a stark lack of rhythmicity in the HaRxL10x line would be expected to be reflected in other phenotypes too.

If this manuscript is to be accepted, the implications about the effects on the circadian clock or global rhythmicity should be removed as these are not supported by the data.

Reviewer #3

(Remarks to the Author)

In this revised ms, the authors did a nice job to address most of my comments. A number of new experiments were performed in order to strengthen the work. Here I have a few more comments:

a. ICS1 was previously shown to be a direct transcriptional target of CHE and ICS1 is a key gene in SA biosynthesis. In addition to some SA related genes, expression of ICS1 should be examined as shown in Fig. 5i-k. This reviewer agrees that Fig 5m shows HaRxL10-mediated ICS1 suppression was not seen in the che2 mutant. A further induction of ICS1 in che2 by HaRxL10 is quite interesting and unexpected. Does this induction also correlate with an increase in SA levels? How would this data explain expression suppression of some other SA related genes by HaRxL10 shown in Fig 5 and other places in this ms? The authors can also measure SA levels in the Col-0 and che2 plants infected Pst DC3000 or Pst DC3000 HaRxL10 and in HaRxL10 overexpression lines of Arabidopsis (in Col-0 and che2) with or without Pst DC3000 infection. Such data shall provide a better clarification on the role of HaRxL10 in SA-mediated defence regulation.

b. New data were included in Fig 6 to link HaRxL10 function to the regulation of key clock genes CCA1 and CAB2. However, the inclusion of only one cycle (24 hr) in LL would make the measurement of clock parameters, such as period, phase, and amplitude, less reliable. CHE is a known repressor of CCA1 (Pruneda-Paz et al, Science 2009). If HaRxL10 suppresses (and acts through) CHE function as stated in this ms, then would one expect to see an induction, rather than a suppression of CCA1, by HaRxL10? Perhaps the authors could do qRT-PCR with Col-0 and che2 plants infected Pst DC3000 or Pst DC3000 HaRxL10 to verify the data. In addition, HaRxL10 overexpression in Col-0 and in che2 with or without Pst DC3000 infection can also be used to verify the data.

c. In Fig. 7a-b, che2 does not seem to affect the rhythm of maltose daily levels, compared with WT (however, a side-by-side

comparison should be made to confirm this observation). Interestingly, HaRxL10 by itself almost abolishes this clock output in Col-0. If HaRxL10 acts through suppressing CHE function, one would expect that HaRxL10 ox/Col-0 and che2 have a similarly compromised clock phenotype, just like other phenotypes described in this ms (disease susceptibility, hypocotyl length, flowering time etc). Please explain the maltose phenotype.

d. Sppl Fig 17b: The FLC expression level correlates strongly with flowering time. The increased expression of FLC in HaRxL10 ox #21/Col-0 does not support the flowering time data. Sppl Fig 17c that shows MAF4 expression is similar between Col-0 and HaRxL10 ox #21/Col-0 also fails to support the early flowering phenotype of HaRxL10 ox #21/Col-0. Please double check the data and provide an explanation for the data.

e. Leaf movement data could provide a strong support to link HaRxL10 function to circadian regulation. Rather than including them in the Rebuttal document, the authors should include such data as a main figure (e.g. in Fig 7), using HaRxL10 overexpression in Col-0 and in che2.

Reviewer #4

(Remarks to the Author)

The revised version incorporates the previously omitted mechanism of HaRxL10's interaction with the circadian clock component CHE, which has significantly enhanced the manuscript. I have no significant objections to the comments that I have revised. However, the presentation of these new findings has given rise to several additional queries. In the supplementary figure 1, the authors demonstrated that several clock components, including CHE, interact with HaRxL10. CCA1 was identified as one of the interacting components. Furthermore, the authors asserted that HaRxL10 influenced the expression of central clock genes, including CCA1, through the observation of CCA1 gene expression patterns in HaRxL10-overexpressing samples. Nevertheless, it is challenging to ascertain whether the alterations in CCA1 gene expression are a direct consequence of HaRxL10-mediated transcription or an indirect effect of CCA1 protein interaction. In light of these considerations, the conclusion appears to be somewhat hasty.

Version 2:

Reviewer comments:

Reviewer #2

(Remarks to the Author)

Thank you for responding to my previous comments. I feel that the authors are very keen to hang onto the idea that HaRxL10 has an effect on the circadian clock, which is not justified by their data. The statement in the abstract "HaRxL10 triggered reprogramming of the transcriptome, particularly affecting the expression of central clock genes and circadian rhythmic genes." should be tempered to note those genes that are actually affected, e.g. SA-related defence genes. I do not find the leaf movement data compelling, nor the maltose data. If this manuscript is to be accepted, the implications about the effects on the circadian clock or global rhythmicity should be removed as these are not supported by the data.

Reviewer #3

(Remarks to the Author)

The authors addressed most of my comments. Because evidence of HaRxL10 in a direct regulation of the circadian clock, especially via a specific circadian clock component CHE, is limited, this reviewer would like to suggest the authors to carefully go through the manuscript and tone down some of the conclusions. CHE-dependent and -independent output caused by HaRxL10 should be clearly discussed in the Discussion section.

Reviewer #4

(Remarks to the Author)

I thank the authors for addressing all my comments. The revised version includes the previously omitted mechanism of HaRxL10's interaction with the with the circadian clock component CHE, which significantly improved the manuscript.

RESPONSE LETTER

Reviewer #1:

The manuscript by Mengyao Fu et al. is a nice contribution to our understanding of the interplay between immunity and the core clock machinery in plants. Several studies have previously revealed a complex interplay between the circadian network and immune responses, by showing that clock mutant plants display alterations in immunity as well as by revealing that pathogen infections alters expression of core clock genes. However, the precise mechanisms of these interactions were not known, particularly how pathogen effectors might modulate the circadian network to enhance pathogenicity.

By screening for potential interactions between *Hyaloperonospora arabidopsisdis* (Hpa) effectors and *Arabidopsis* core clock proteins, the authors found that the HaRxL10 effector directly interacts the CCA1 Hiking Expedition (CHE) protein. CHE is a TCP transcription factor that represses the expression of CIRCADIAN CLOCK ASSOCIATED 1 (CCA1) 1 gene. CCA encodes a MYB transcription factor that acts at the core of the plant circadian network, repressing the expression of other core clock genes and hundreds of clock outputs. Both CHE and CCA1 have previously been implicated in mediating/modulating immune responses in *Arabidopsis*.

After the initial Y2H screen, the authors confirmed that HaRxL10 acts as a secreted effector that enhances susceptibility to *Pseudomonas* infection. They also found that HaRxL10 overexpression enhances growth of biotrophic fungi such as *Phytophthora sojae*. They also showed that HaRxL10 inhibits CHE degradation by inhibiting its interaction with ZTL, an E3 Ubiquitin ligase, while at the same time binding of HaRxL10 to CHE reduces CHE ability to bind DNA and regulate gene expression. Therefore, exposure of plant cells to HaRxL10 appears to inhibit CHE activity. In agreement with this, the authors found that exposure of *Arabidopsis* plants to HaRxL10 through its expression as a secreted protein by *Pseudomonas syringae* pv tomato DC3000 (PSt), enhances bacterial virulence and this effects requires CHE, as it is not observed in the mutants.

A genome wide transcriptome analysis further confirmed that HaRxL10 triggers transcriptomic changes in Pst infected plants that are similar to those observed in the mutants. Indeed, they found that expression of some defence genes was altered upon HaRxL10 exposure in a CHE dependent way. The authors also found that an acute exposure to the HaRxL10 effector triggered alterations in expression of a few clock genes such as RVE4, TCP22 and LWD1, while constant exposure to HaRxL10 through overexpression from a constitutive promoter caused a reduction in CCA1 expression. These effects were associated with an apparent global change in periodicity and phase of clock regulated genes, as results from an evaluation of these parameters in a circadian RNA-seq expression time-course. Finally, the authors also found CHE dependent effects of HaRxL10 on flowering time regulation, supporting the idea that HaRxL10

affects immunity and development in *Arabidopsis* plants through its inhibitory effects on CHE activity.

The manuscript is well written. The results are very interesting, the experiments have been well conducted, and most of the conclusions are supported by their data. I have only a few concerns:

Response: We appreciate the reviewer's recognition of this work.

While it is clear that HaRxL10 can act as an effector that alters immunity through its interactions with CHE when secreted by *Pst* DC3000, whether the same effect occurs in its native system, i.e. *Hpa*, remains to be determined. I understand the reason for using this heterologous system is likely the result in difficulties in altering the function of HaRxL10 in *Hpa*. The authors should explicitly comment on this and should briefly explain the rationale of using *Pst* to test the effects of secreted HaRxL10 as an effector. I understand there have been reports describing the use of this approach before, but this needs to be mentioned and explained.

Response: This is an excellent suggestion. We have added explanation of the rationale using *Pst* DC3000 to study the function of an effector in the revised manuscript (**Page 4, lines 7-18; Page 15, lines 36-38**).

As this reviewer mentioned, *Hpa* is an obligate biotrophic pathogen, which extracts nutrients only from living plant tissue and cannot grow apart from its hosts (**Baxter et al., 2010**). Conventional genetic tools such as plasmid transformation and transposon insertion are usually not applicable to this type of pathogen. *Hpa* has close interactions with its host *Arabidopsis*, and such interactions may increase the difficulty of genetic manipulation, because any genetic manipulation may affect the interactions between the pathogens and their hosts, and even affect the life cycle and pathogenicity. Moreover, exogenous genes, such as antibiotic-resistant genes or fluorescent protein genes may be difficult to be maintained stably. Due to these possible reasons, there has been no report about successful genetic manipulation of *Hpa* to date. To surmount this technical difficulty, previous studies on *Hpa* effector often generated an effector overexpression line of *Arabidopsis* (**Yang et al., 2017**). In our study, we also utilised this strategy to study the function of HaRxL10 (**Fig. 7a-h, Supplementary Fig. 10c**).

However, the ectopic expression of a pathogen effector in plants may not accurately reflect the effector's natural function probably due to artificial effects generated by overexpression of the effector, whose abundance may not be within the physiological range during pathogen infection. Therefore, we also employed a natural effector delivery system, the bacterial type-three secretion system (TTSS) of *Pst* DC3000, to deliver HaRxL10 into the plant cells. *Pst* DC3000 is a virulent pathogen for *Arabidopsis*. Introduction of HaRxL10 into *Pst* DC3000 enabled us to study the effect of HaRxL10 by using *Pst* DC3000 as a control pathogen. Several other studies have also utilised *Pst*

DC3000 system to study the function of nonbacterial pathogen effectors (**Fabro et al., 2011; Rentel et al., 2008; Sohn et al., 2007**). For example, another *Hpa* RXLR-type effector, HaRxLL470, was found to target HY5 to attenuate the plant immunity where the effector HaRxLL470 was delivered into the host plant by TTSS of *Pst* DC3000 (**Chen et al., 2021**).

We added a section titled “Secretion of nonbacterial effector by *Pst* DC3000 type-three secretion system” in the Method of the revised manuscript (**Page 30, lines 21-34**) for technical details of construction of *Pst* DC3000-HaRxL10 and *Pst* DC3000-HaRxL10.

References:

- Baxter, L., Tripathy, S., Ishaque, N., Boot, N., Cabral, A., Kemen, E., Thines, M., Ah-Fong, A., Anderson, R., Badejoko, W., *et al.* (2010). Signatures of adaptation to obligate biotrophy in the *Hyaloperonospora arabidopsidis* genome. *Science* *330*, 1549-1551.
- Chen, S., Ma, T., Song, S., Li, X., Fu, P., Wu, W., Liu, J., Gao, Y., Ye, W., Dry, I.B., *et al.* (2021). Arabidopsis downy mildew effector HaRxLL470 suppresses plant immunity by attenuating the DNA-binding activity of bZIP transcription factor HY5. *New Phytol* *230*, 1562-1577.
- Fabro, G., Steinbrenner, J., Coates, M., Ishaque, N., Baxter, L., Studholme, D.J., Korner, E., Allen, R.L., Piquerez, S.J., Rougon-Cardoso, A., *et al.* (2011). Multiple candidate effectors from the oomycete pathogen *Hyaloperonospora arabidopsidis* suppress host plant immunity. *PLoS Pathog* *7*, e1002348.
- Rentel, M.C., Leonelli, L., Dahlbeck, D., Zhao, B., and Staskawicz, B.J. (2008). Recognition of the *Hyaloperonospora parasitica* effector ATR13 triggers resistance against oomycete, bacterial, and viral pathogens. *Proc Natl Acad Sci U S A* *105*, 1091-1096.
- Sohn, K.H., Lei, R., Nemri, A., and Jones, J.D. (2007). The downy mildew effector proteins ATR1 and ATR13 promote disease susceptibility in *Arabidopsis thaliana*. *Plant Cell* *19*, 4077-4090.
- Yang, L., Teixeira, P.J.P.L., Biswas, S., Finkel, O.M., He, Y.J., Salas-Gonzalez, I., English, M.E., Epple, P., Mieczkowski, P., and Dangl, J.L. (2017). *Pseudomonas syringae* Type III Effector HopBB1 Promotes Host Transcriptional Repressor Degradation to Regulate Phytohormone Responses and Virulence. *Cell Host & Microbe* *21*, 156-168.

My major concern though is related to the effect of HaRxL10 on the plant circadian clock. While the authors indicate that period and phase of clock regulated gene expression are altered, I could not find (maybe a missed it) an explanation on how these parameters were evaluated.

Response: To derive the oscillation parameters including period, phase, amplitude and average expression levels, weighted non-linear regression analysis was performed to fit the time-course expression levels to a cosine wave with an intercept using RStudio. The estimated period, phase and amplitude were retrieved from the fitted cosine wave and the intercept represented average expression levels. We have added the explanation in our revised manuscript (**Page 23, lines 23-27**).

Furthermore, looking at the individual clock gene oscillatory expression patterns

shown in supplementary figure 13, I do not see that the data supports a significant effect on rhythms of core clock gene expression. Most of the circadian patterns of expression of core clock genes are apparently unaffected in period and/or phase in that supplementary figure. The presumed effects of HaRxL10 on expression of clock genes is associated with genes that have only moderate (RVE4) or non (LWD1, TCP22) circadian regulated expression, and only overall mRNA levels of these genes are affected. The authors should clearly comment on this in the text.

Response: We agree with this reviewer that the period and phase of most of the central clock gene expression were unaffected in our experiment. Light is the strongest input signal of the plant circadian clock, which synchronizes the internal clock with the environmental cues (**Harmer, 2009**). Due to this reason, some known clock mutants display no or very mild clock dysfunction under diurnal conditions, but obvious clock period change phenotype under constant light conditions (**Green et al., 2002; Hicks et al., 1996**). Light significantly elevates plant defence against pathogen infection and constant light conditions reduce the effect of pathogen infection (**Ballaré, 2014**). Therefore, the light and dark cycle was used in our experiment to provide a favorable condition for the bacterial infection. This diurnal condition combines the effect of the internal clock and environmental light on gene expression. Therefore, the effect of HaRxL10 on the clock gene expression may be diminished under diurnal conditions. We have added a description and possible explanation of the results illustrated in **new Supplementary Fig. 14** (previous Supplementary Fig. 13) in the revised manuscript (**Page 19, lines 40-45; Page 20, lines 1-15**).

To better assess the effect of HaRxL10 on the clock function, we conducted leaf movement assay and measured maltose content and hypocotyl length under constant light conditions as suggested by this reviewer. Please refer to the next response for the results of these three experiments.

References:

- Ballaré, C.L. (2014). Light regulation of plant defense. *Annu Rev Plant Biol* 65, 335-363.
- Green, R.M., Tingay, S., Wang, Z.Y., and Tobin, E.M. (2002). Circadian rhythms confer a higher level of fitness to Arabidopsis plants. *Plant Physiol* 129, 576-584.
- Harmer, S.L. (2009). The circadian system in higher plants. *Annu Rev Plant Biol* 60, 357-377.
- Hicks, K.A., Millar, A.J., Carré, I.A., Somers, D.E., Straume, M., Meeks-Wagner, D.R., and Kay, S.A. (1996). Conditional circadian dysfunction of the Arabidopsis early-flowering 3 mutant. *Science* 274, 790-792.

They should also adjust their conclusions regarding effects of HaRxL10 on the clock to their results. A simple experiment to evaluate whether HaRxL10 affects circadian function is to analyze the rhythms of leaves movements or gene expression of a few clock genes in HaRxL10 overexpressing plants compared to those of WT plants. If period and phase of leaf movement rhythms or clock gene expression are not altered in those plants, the authors should adjust their conclusions. My impression is that all their

finding are interesting and well supported, but that the effect of HaRxL10, although dependent on CHE, are not mediated by changes in clock function but rather through more direct effects of CHE on immune and developmental responses as clock outputs.

Response: We analysed the expression of the central clock gene *CCA1* and a widely-used clock output gene *CAB2* in the wild-type and HaRxL10 overexpression plants. Our results showed that HaRxL10 overexpression plants displayed lower amplitude and average expression of *CCA1* compared to the wild-type plants (**Fig. R1a, new Fig. 6a**). Period and phase of *CCA1* did not show significant difference in the wild-type and HaRxL10 overexpression plants (**Fig. R1a, new Fig. 6a**). The period of *CAB2* was shortened in HaRxL10 overexpression plants compared to the wild-type plants (**Fig. R1b, new Fig. 6e**). HaRxL10 overexpression plants also displayed lower amplitude and average expression of *CAB2* (**Fig. R1b, new Fig. 6a**). Phase of *CAB2* did not show significant difference in the wild-type and HaRxL10 overexpression plants (**Fig. R1b, new Fig. 6a**).

To further assess the effect of HaRxL10 on the clock function, we conducted leaf movement assay and measured maltose content and hypocotyl length under constant light conditions. Our results showed that period of leaf movement rhythms was shorter in HaRxL10 overexpression plants than that in the wild-type plants (**Fig. R1c**). Phase of leaf movement rhythms was delayed in HaRxL10 overexpression plants (**Fig. R1c**). In addition to leaf movement, the circadian clock also regulates the amount of some metabolites, such as maltose. We found that maltose levels showed a strong circadian rhythm in the wild-type plants, but this rhythm was disrupted in HaRxL10 overexpression plants (**Fig. R1d, new Fig. 7a**). To study whether this HaRxL10-induced maltose oscillation perturbation was dependent on CHE, we measured maltose contents in the *che-2* and HaRxL10 overexpression in *che-2* plants. Both plants showed strong circadian rhythm of maltose (**Fig. R1e, new Fig. 7b**), suggesting that HaRxL10 requires CHE to perturb maltose oscillation. Hypocotyl growth is another well-characterized circadian-regulated physiological process. We found that overexpressing HaRxL10 in the wild-type plants lengthened the hypocotyl (**Fig. R1f, g; new Fig. 7c, d**). However, HaRxL10 overexpression plants in the *che-2* did not show hypocotyl length difference with the *che-2* plants (**Fig. R1f, g; new Fig. 7c, d**), suggesting that HaRxL10-regulated hypocotyl growth was dependent on CHE.

Taken together, our results demonstrated that HaRxL10 could affect several circadian-regulated physiological processes, corroborating its impact on plant clock function. We have revised our manuscript to incorporate these new findings accordingly (**Page 20, lines 15-22; Page 21, lines 4-10; Page 23, lines 33-45; Page 24, lines 1-2**).

Fig. R1 HaRxL10 affects circadian-regulated physiological processes. **a-b**, Relative expression levels of *CCA1* (**a**) and *CAB2* (**b**) in 3-week-old wild-type (Col-0) and HaRxL10 overexpression (HaRxL10_{ox} #18) *Arabidopsis* plants. Plants were grown under 12 h light/12 h dark conditions for 3 weeks and transferred to the constant light (LL) condition for 24 hours. Samples were collected at 4 hour-intervals under the LL condition and analysed by RT-qPCR with *UBQ5* as the internal control. White bar, subjective day. Gray bar, subjective night. The data are shown as mean \pm SEM (n = 3). Period, phase, amplitude and average expression were calculated by nonlinear regression. The *p* values were calculated by unpaired *t* test with Welch's correction. This experiment was repeated twice with similar results. **c**, Rhythm of leaf movement was measured in wild-type (Col-0) and HaRxL10 overexpression (HaRxL10_{ox} #18) *Arabidopsis* seedlings. Plants were grown under 12 h light/12 h dark conditions for 7 days and transferred to the constant light (LL) condition for 3 days. Pictures were then taken once an hour under the LL condition. Distance between two cotyledons was measured to represent the leaf movement. Gray bar, subjective night. The data are shown as mean \pm SEM (n = 15). Period and phase were calculated by nonlinear regression. The *p* values were calculated by unpaired *t*-test with Welch's correction. This experiment was repeated twice with similar results. **d-e**, Maltose contents in 3-

week-old wild-type (Col-0) and HaRxL10 overexpression (HaRxL10_{ox} #21) *Arabidopsis* plants (d) or *che-2* and HaRxL10 overexpression (HaRxL10_{ox} #4 in *che-2*) *Arabidopsis* plants (e). Plants were grown under 12 h light/12 h dark conditions for 3 weeks and transferred to the constant light (LL) condition for 24 hours. Samples were collected at 4 hour-intervals under the LL condition. White bar, subjective day. Gray bar, subjective night. The data are shown as mean \pm SEM (n = 6). Statistically significant oscillation of the data was determined by *F*-test comparing the goodness of fit derived from nonlinear regression using a cosine wave or a straight line ($p < 0.05$). **f-g**, Representative images (f) and statistical analysis of hypocotyl length (g) in the 7-day-old seedlings of wild-type (Col-0), *che-2* and HaRxL10 overexpression (HaRxL10_{ox}) in different backgrounds (Col-0 or *che-2*). Plants were grown under 12 h light/12 h dark conditions for 4 days and transferred to the constant light (LL) condition for 3 days. Photos were taken and hypocotyl length was measured. The *p* values were calculated by one-way ANOVA followed by Holm-Šídák's multiple comparisons test. The data are shown as mean \pm SEM (n = 33).

Reviewer #2:

This was a very lengthy and comprehensive manuscript describing the interaction of a putative oomycete effector protein, HaRxL10 with the CCA1 HIKING EXPEDITION (CHE) protein of *Arabidopsis*. The authors suggest that HaRxL10 targets CHE which leads to a suppression of plant immunity and early flowering. Unfortunately, the authors do not provide evidence that HaRxL10 is indeed expressed by *Hyaloperonospora arabidopsidis* (Hpa) during infection, nor that it plays a role in Hpa virulence as an effector.

Response: We apologise for the missing information about HaRxL10. In a previous study, the researchers utilised a high-throughput cDNA tag sequencing method to measure gene expression changes in both *Hpa* and *Arabidopsis* during infection. *HaRxL10* was found to be highly expressed after infecting *Arabidopsis* (Asai et al., 2014). We have added this information and relative reference in the revised manuscript (Page 3, lines 9-14).

We agree with this reviewer that the direct evidence to show the role of HaRxL10 in *Hpa* virulence was not provided. An ideal experiment is to construct a loss-of-function mutant of *HaRxL10* in *Hpa* and compare its infection of *Arabidopsis* with the wild-type *Hpa*. However, *Hpa* is an obligate biotrophic pathogen, which extracts nutrients only from living plant tissue and cannot grow apart from its hosts (Baxter et al., 2010). Conventional genetic tools such as plasmid transformation and transposon insertion are usually not applicable to this type of pathogen, as Reviewer #1 mentioned that there are "difficulties in altering the function of HaRxL10 in *Hpa*". To overcome this technical difficulty, we utilised two strategies to express HaRxL10 in *Arabidopsis* plants. One was the type-three secretion system (TTSS) of *Pst* DC3000-mediated effector delivery, which was also adopted by previous studies (Chen et al., 2021; Fabro et al., 2011;

Rentel et al., 2008; Sohn et al., 2007). The other one is the generation of an effector overexpression line of *Arabidopsis*, which was widely used in studying the function of a *Hpa* effector. Both results suggested that HaRxL10 enhanced the susceptibility of plants to *Pst* DC3000 infection (**Fig. 5a, b; Supplementary Fig. 10c**). Subsequent RNA-seq experiments revealed that HaRxL10 suppressed the expression of SA-related defence genes (**Fig. 5h-m**). A previous study showed that *Hpa* infection greatly suppressed the SA pathway marker gene, *PR1*, indicating an important role of SA in defending against *Hpa* (**Asai et al., 2014**). Collectively, these results implied that HaRxL10 promotes *Hpa* virulence by suppressing SA-mediated defence. We have added an explanation about our experiment designs in the revised manuscript (**Page 4, lines 7-18; Page 15, lines 36-38**).

References:

- Asai, S., Rallapalli, G., Piquerez, S.J., Caillaud, M.C., Furzer, O.J., Ishaque, N., Wirthmueller, L., Fabro, G., Shirasu, K., and Jones, J.D. (2014). Expression profiling during arabidopsis/downy mildew interaction reveals a highly-expressed effector that attenuates responses to salicylic acid. *PLoS Pathog* *10*, e1004443.
- Baxter, L., Tripathy, S., Ishaque, N., Boot, N., Cabral, A., Kemen, E., Thines, M., Ah-Fong, A., Anderson, R., Badejoko, W., *et al.* (2010). Signatures of adaptation to obligate biotrophy in the *Hyaloperonospora arabidopsidis* genome. *Science* *330*, 1549-1551.
- Chen, S., Ma, T., Song, S., Li, X., Fu, P., Wu, W., Liu, J., Gao, Y., Ye, W., Dry, I.B., *et al.* (2021). Arabidopsis downy mildew effector HaRxLL470 suppresses plant immunity by attenuating the DNA-binding activity of bZIP transcription factor HY5. *New Phytol* *230*, 1562-1577.
- Fabro, G., Steinbrenner, J., Coates, M., Ishaque, N., Baxter, L., Studholme, D.J., Korner, E., Allen, R.L., Piquerez, S.J., Rougon-Cardoso, A., *et al.* (2011). Multiple candidate effectors from the oomycete pathogen *Hyaloperonospora arabidopsidis* suppress host plant immunity. *PLoS Pathog* *7*, e1002348.
- Rentel, M.C., Leonelli, L., Dahlbeck, D., Zhao, B., and Staskawicz, B.J. (2008). Recognition of the *Hyaloperonospora parasitica* effector ATR13 triggers resistance against oomycete, bacterial, and viral pathogens. *Proc Natl Acad Sci U S A* *105*, 1091-1096.
- Sohn, K.H., Lei, R., Nemri, A., and Jones, J.D. (2007). The downy mildew effector proteins ATR1 and ATR13 promote disease susceptibility in *Arabidopsis thaliana*. *Plant Cell* *19*, 4077-4090.

It is not clear why HaRXxL10 was chosen to study, nor why CHE was chosen to focus on rather than any of the other 8 TCP transcription factors that were demonstrated to interact with HaRxL10.

Response: We apologise for unclear explanation of our rationale. The interplay between the circadian clock and plant immunity has been revealed by several previous studies. However, the molecular mechanisms of how pathogens affect the host circadian clock remain largely unknown. The availability of genome sequences of both *Hpa* and *Arabidopsis* has led to the widespread use of the *Hpa-Arabidopsis* pathosystem for studying the co-evolution of the host and parasite. The circadian-mediated plant defense response was first reported in the studies of the interaction between *Hpa* and

Arabidopsis. Effectors are molecular weapons of *Hpa* to suppress host immunity. We therefore hypothesized that *Hpa* may secrete specific effectors to target host clock component to interfere with plant immunity.

We started with performing a Y2H screen testing direct interactions between *Hpa* effectors and *Arabidopsis* clock components. We synthesized genes encoding 50 *Hpa* effectors including HaRxL10, which have been previously reported to be highly expressed during *Hpa* infection. We cloned 14 *Arabidopsis* clock genes from *Arabidopsis* cDNA, including *CCA1*, *LHY*, *CHE*, *TOC1*, *PRR3/5/7/9*, *ZTL*, *ELF3*, *ELF4*, *LUX*, *TCP20*, *TCP22*. From these 700 (50 × 14) sets of Y2H assays, we found that HaRxL10 interacted with several clock components (**Fig. R2, new Supplementary Fig. 1**). In this study, we focused on CHE because our results revealed that HaRxL10 suppressed plant defence in a CHE-dependent manner (**Fig. 5a, b**). Our on-going projects aim to investigate the biological meaning of HaRxL10 interacting with other clock components, such as CCA1.

This study focused on the interaction between a pathogen effector and clock components. CHE belongs to a large TCP family, which contains 24 members in *Arabidopsis*. CHE, along with TCP20 and TCP22, are three known clock components within the TCP family. HaRxL10 did not interact with TCP20 and TCP22 in yeast (**Supplementary Fig. 20**). That is the reason we chose to focus on CHE rather than other HaRxL10-interacting TCPs.

Fig. R2 Interactions between HaRxL10 and central clock members in yeast. Synthetic dropout medium without leucine and tryptophan (SD/-Leu-Trp) was used for positive yeast transformant selection. Synthetic dropout medium without leucine, tryptophan, histidine and adenine (SD/-Leu-Trp-His-Ade) was used for selection of protein interaction by the reporter gene *HIS3*. Photographs were taken 2 days after plating of yeast cells with $OD_{600\text{ nm}} = 1$.

The interactions of HaRxL10 with the other TCP factors is insufficiently discussed.

Response: We have added background information on TCP family proteins, provided results demonstrating HaRxL10's impact on *TCP* gene expression and discussed the biological meaning of the interaction between HaRxL10 and several TCP factors. Please refer to our revised manuscript for more details (**Page 26, lines 40-45; Page 27, lines 1-31**).

The work will be of interest to plant pathologists and those interested in plant pathogens.

Response: Thank you for the recognition of this work.

The results are generally interpreted in a very simplistic, direct way without due consideration of wider effects relating to consequences of overexpression, redundancy, or pleiotropy.

Response: We provided data showing the effect of overexpression of HaRxL10 on the expression of clock genes and other TCP genes (**new Fig. 6a; new Supplementary Fig. 16; new Supplementary Fig. 21**). In addition to gene expression, we observed developmental phenotypes caused by overexpression of HaRxL10 (**new Supplementary Fig. 19**). Moreover, considering that HaRxL10 could target multiple TCP proteins in addition to CHE and affect the expression of several *TCP* genes, HaRxL10 may also affect other physiological processes not assessed in our study in a CHE-independent manner. To provide a more comprehensive understanding of CHE-dependent and independent effects of HaRxL10, we performed additional assays and showed that HaRxL10 triggers a large-scale transcriptome reprogramming and identified CHE-dependent and independent HaRxL10 responsive genes (**Supplementary Fig. 22**). Please refer to our revised manuscript for details (**Page 20, lines 5-45; Page 21, lines 1-4; Page 25, lines 3-11; Page 27, lines 14-31**).

Aims and introduction

The manuscript is lacking in a clearly articulated hypothesis or research question. It is not clear why the authors chose the HaRxL10 to investigate or why they thought it may interact with clock components.

Response: Please refer to our response to the previous question which is similar to this one "It is not clear why HaRxL10 was chosen to study, nor why CHE was chosen to focus on rather than any of the other 8 TCP transcription factors that were demonstrated to interact with HaRxL10".

Although the authors state on p3 lines 7-10 that they "conducted a yeast-two hybrid (Y2H) screening between predicted Hpa RXLR effectors and Arabidopsis central clock components" these data are not shown. How many Hpa RXLR effectors were screened and where are the data for the other effectors and Arabidopsis central clock components?

There are Y2H data for interactions between HaRxL10 and TCP family members in shown in Supplementary Figure 15.

Response: We apologise for unclear explanation about our experimental setting. We conducted a yeast-two hybrid (Y2H) screen to test the interactions between 50 *Hpa* effectors which have been previously reported to be highly expressed during *Hpa* infection (Asai et al., 2014) and 11 *Arabidopsis* central clock components. We found that the effector HaRxL10 interacts with several clock components including CHE, CCA1 and LUX (Fig. R3, new Supplementary Fig. 1). We have revised the manuscript accordingly (Page 3, lines 8-14).

Fig. R3 Interactions between HaRxL10 and central clock members in yeast. Synthetic dropout medium without leucine and tryptophan (SD/-Leu-Trp) was used for positive yeast transformant selection. Synthetic dropout medium without leucine, tryptophan, histidine and adenine (SD/-Leu-Trp-His-Ade) was used for selection of protein interaction by the reporter gene *HIS3*. Photographs were taken 2 days after plating of yeast cells with $OD_{600\text{ nm}} = 1$.

In addition to HaRxL10, we found other *Hpa* effector could interact with *Arabidopsis* clock components in yeast. For example, we found HaRxL16 could also interact with CHE in Y2H assays (Fig. R4a). However, our bacterial infection assay showed that HaRxL16 could not enhance plant disease (Fig. R4b). These results are not the focus of this study and some findings have undergone further studies in our lab.

Fig. R4 Characterization of the effector HaRxL16. **a**, Interaction of HaRxL16 with CHE in Y2H assays. Synthetic dropout medium without leucine and tryptophan (SD/-Leu-Trp) was used for positive yeast transformant selection. Synthetic dropout medium without leucine, tryptophan, histidine and adenine (SD/-Leu-Trp-His-Ade) was used for selection of protein interaction by the reporter gene *HIS3*. Photographs were taken 2 days after plating of yeast cells with $OD_{600\text{ nm}} = 1$. This experiment was repeated three times with similar results. **b**, Bacterial growth in wild-type (Col-0) *Arabidopsis* leaves infiltrated with *Pst* DC3000 or *Pst* DC3000-HaRxL16 ($OD_{600\text{ nm}} = 0.002$) at 3 days post-inoculation (dpi). Data represent the mean \pm SEM ($n = 12$). The p value was calculated by two-sided unpaired Student's t -test. cfu, colony forming unit. Experiments were repeated three times with similar results.

References:

Asai, S., Rallapalli, G., Piquerez, S.J., Caillaud, M.C., Furzer, O.J., Ishaque, N., Wirthmueller, L., Fabro, G., Shirasu, K., and Jones, J.D. (2014). Expression profiling during arabidopsis/downy mildew interaction reveals a highly-expressed effector that attenuates responses to salicylic acid. *PLoS Pathog* 10, e1004443.

Results

Fig 1c. It is not clear what the leaves represent here – are they replicates from multiple plants, or leaves from a single representative plant? Either way, why have these been selected to be shown – they do not match the $n=16$, and do not appear to represent the same developmental leaf from different plants.

Response: We apologise for the unclear description in the figure legend. For this experiment, 16 individual plants were used for bacterial infiltration. For each plant, the 4th and 5th leaves were chosen for bacterial infiltration and two leaf discs respectively from these two leaves were collected in one tube and subjected to the bacterial growth counting steps. The pictures shown in Fig. 1c were the 4th and 5th leaves from two representative plants 3 days post bacterial infiltration. We have added above

explanation in the figure legend in our revised manuscript (**Page 5, lines 18-19**).

Similarly, for Fig 1g where 10 leaves are shown per line – does this match experimental n (not stated) or representative leaves, or leaves from one plant?

Response: We apologise for the unclear description in the figure legend. For this experiment, 5 individual plants were used for *Phytophthora sojae* infection assay. For each plant, the 5th and 6th leaves were chosen for inoculation. Ten infected leaves were photographed and collected for analysis of biomass 3 days after inoculation. The pictures shown in Fig. 1g were 10 infected leaves in one batch of experiment. We repeated this experiment twice with similar results. We have added above explanation in the figure legend in our revised manuscript (**Page 5, lines 29-30**).

It would be useful to provide some information about the phenotype of the Arabidopsis stable transgenic lines overexpressing HaRxL10 prior to infections, and include pictures of leaves that have not been challenged with *Phytophthora sojae* in Fig 1g for comparison.

Response: We found that the plants overexpressing HaRxL10 in the wild-type displayed larger true leaves in the 2-week-old seedlings (**Fig. R5a, b; new Supplementary Fig. 19a, b**). However, overexpressing HaRxL10 in the *che-2* did not show the leaf size difference compared to the *che-2* plants (**Fig. R5a, b; new Supplementary Fig. 19a, b**), suggesting that this HaRxL10-mediated leaf size phenotype is CHE-dependent. Different from young seedlings, the 3-week-old HaRxL10 overexpression plants in both wild-type and *che-2* backgrounds showed no obvious developmental phenotypes in aspects of leaf size and shape (**Fig. R5c, d; new Supplementary Fig. 19c, d**). We have provided above information in our revised manuscript (**Page 25, lines 3-11**).

We also provided the pictures of 5th and 6th leaves from 5 individual 5-week-old plants not challenged with *Phytophthora sojae* and observed no obvious phenotypes between HaRxL10 overexpression and the wild-type plants (**Fig. R5e, new Fig. 1g**).

Fig. R5 Leaf phenotypes of HaRxL10 overexpression plants at different developmental stages. **a**, Photographs of the representative 2-week-old *Arabidopsis* plants growing under 12 h light/12h dark. Scale bar, 1 cm. **b**, The areas of true leaves were measured from 2-week-old *Arabidopsis* plants growing under 12 h light/12h dark. The p values were calculated by one-way ANOVA followed by Holm-Šidák's multiple comparisons test. The data are shown as mean \pm SEM ($n = 20$). **c**, **d**, Photographs of 3-week-old *Arabidopsis* plants growing under 12 h light/12h dark (**c**) and their representative 4th and 5th leaves (**d**). Scale bar, 2 cm. **e**, Pictures of 5th and 6th leaves from 5-week-old *Arabidopsis* plants growing under 12 h light/12h dark without *Phytophthora sojae* infection. Scale bar, 2 cm.

The interaction of AD-CHEc with HaRxL10 in bimolecular interaction assay indicates that the interaction is not only in the nucleus (Fig2d), but this is not discussed. Please provide some comments on this finding and its implications.

Response: We agree with this reviewer that the BiFC results indicated that the interaction between CHEc and HaRxL10 could occur in both nucleus and cytosol, suggesting that the N-terminal and TCP domains may influence the subcellular localization of CHE. We have revised the manuscript accordingly (**Page 6, lines 23-27**).

At the top of page 9, HaRx21 appears – presumably a typo. Please correct.

Response: Thank you for pointing out this typo. We have corrected the name of the effector in the revised manuscript (**Page 10, line 3**).

The authors state that they wish to investigate the effect of HaRxL10 on the transcriptional regulation function of CHE, and chose to investigate CHE as a representative downstream gene. Why?

Response: For this purpose, we need to use a CHE direct target gene as a representative gene. The well-known CHE target is *CCA1*. We described how HaRxL10 affected *CCA1* transcription in our manuscript (**Page 20, lines 19-41**). In addition to this known CHE target gene, we wondered if HaRxL10 affected other CHE targets. A bioinformatic analysis of the *CHE* promoter sequence revealed the presence of a TCP-binding site (TBS), suggesting that CHE may regulate its own gene expression. Y1H experiment showed that CHE could indeed bind to its own promoter (**Fig. 4a**). Further ChIP-qPCR assays using *35S:YFP-CHE* transgenic plants demonstrated the recruitment of YFP-CHE to the TBS and the nearby region, providing evidence of CHE binding to its own promoter *in planta* (**Fig. 4b**). Additionally, dual-luciferase assays revealed that CHE protein activated the transcription of CHE gene (**Fig. 4c**). Collectively, these results indicate that CHE positively regulates its own gene expression by binding to the *CHE* promoter. Therefore, we showed that the transcription of *CHE* could be regulated by its protein for the first time, which fulfills the knowledge gap of how the transcription of *CHE* is regulated.

What about the other genes that are transcriptionally regulated by CHE, such as *CCA1*? This would be of interest when later discussing effects of HaRxL10 on the clock.

Response: We also wondered about the effect of HaRxL10 on the expression of *CCA1*. Analysis of *CCA1* expression in HaRxL10 overexpression plants by RT-qPCR revealed that HaRxL10 suppressed the transcription of *CCA1* (**Fig. R6a, b; new Fig. 6a, b**). Dual luciferase assays using *che-2* Arabidopsis protoplasts showed that HaRxL10 alone could not affect *CCA1* expression (**Fig. R6c, new Fig. 6c**). However, co-expression of HaRxL10 and CHE significantly suppressed the expression of *CCA1* in the wild-type *Arabidopsis* protoplasts, suggesting that HaRxL10-triggered *CCA1* inhibition requires CHE (**Fig. R6d, new Fig. 6d**). This result could not be explained by HaRxL10 interfering with CHE's binding to *CCA1* promoter, since CHE negatively regulates *CCA1* expression. Therefore, HaRxL10 may affect other components which regulate *CCA1* expression. One possibility is that HaRxL10 represses the expression of positive regulators of *CCA1*, *TCP22*, in a CHE-dependent way (**Supplementary Fig. 15a, d**). HaRxL10 overexpression plants showed higher *TOC1* expression, indicating that HaRxL10 could promote the expression of *CCA1* repressor gene, *TOC1* (**Fig. R6e, new Supplementary Fig. 16**). This HaRxL10-mediated *TOC1* induction may lead to the

suppression of *CCA1*. We have shown these results in our revised manuscript (Page 20, lines 19-45; Page 21, lines 1-4).

Fig. R6 HaRxL10 represses the expression of *CCA1* and promotes the expression of *TOC1*. **a**, Relative expression levels of *CCA1* in 3-week-old wild-type (Col-0) and HaRxL10 overexpression (HaRxL10_{ox} #18) *Arabidopsis* plants. Plants were grown under 12 h light/12 h dark conditions for 3 weeks and transferred to the constant light (LL) condition for 24 hours. Samples were collected at 4 hour-intervals under the LL condition and analysed by RT-qPCR with *UBQ5* as the internal control. White bar, subjective day. Gray bar, subjective night. The data are shown as mean \pm SEM (n = 3). Period, phase, amplitude and average expression were calculated by nonlinear regression. The *p* values were calculated by unpaired *t*-test with Welch's correction. This experiment was repeated twice with similar results. **b**, Expression levels of *CCA1* in the wild-type (Col-0) and two HaRxL10 overexpression (HaRxL10_{ox}) *Arabidopsis* plants. Samples were collected at ZT0 and analysed by RT-qPCR with *UBQ5* as internal control and normalised by the expression in Col-0. The data are shown as mean \pm SEM (n = 9, 3 independent experiments with 3 technical replicates). The *p* values were calculated by one-way ANOVA followed by Holm-Šidák's multiple comparisons test. **c**, Dual-luciferase assay performed using *che-2* *Arabidopsis* protoplasts transiently co-expressing HaRxL10-3xFlag or RFP-3xFlag (control) and the reporter driven by *CCA1* promoter. The ratio of firefly luciferase and renilla luciferase activities was calculated

and normalised to the control. The data are shown as mean \pm SEM (n = 9, 3 independent experiments with 3 technical replicates). The *p* value was calculated using two-sided unpaired Student's *t*-test. **d**, Dual-luciferase assay performed using *Arabidopsis* protoplasts transiently co-expressing different protein combinations and the reporter driven by *CCA1* promoter. The ratio of firefly luciferase and renilla luciferase activities was calculated and normalised to the control. The data are shown as mean \pm SEM (n = 9, 3 independent experiments with 3 technical replicates). The *p* value was calculated by two-sided unpaired Student's *t*-test. **e**, Relative expression levels of *TOC1* in 3-week-old wild-type (Col-0) and HaRxL10 overexpression (HaRxL10_{ox} #18) *Arabidopsis* plants. Plants were grown under 12 h light/12 h dark conditions for 3 weeks and transferred to the constant light (LL) condition for 24 hours. Samples were collected at 4 hour-intervals under the LL condition and analysed by RT-qPCR with *UBQ5* as the internal control. White bar, subjective day. Gray bar, subjective night. The data are shown as mean \pm SEM (n = 3). Period, phase, amplitude and average expression were calculated by nonlinear regression. The *p* values were calculated by unpaired *t*-test with Welch's correction. This experiment was repeated twice with similar results.

To further assess the effect of HaRxL10 on the clock function, we conducted leaf movement assay and measured maltose content and hypocotyl length under constant light conditions. Our results showed that period of leaf movement rhythms was shorter in HaRxL10 overexpression plants than that in the wild-type plants (**Fig. R7a**). Phase of leaf movement rhythms was delayed in HaRxL10 overexpression plants (**Fig. R7a**). In addition to leaf movement, the circadian clock also regulates contents of some metabolites, such as maltose. We found that maltose levels showed a strong circadian rhythm in the wild-type plants, but this rhythm was disrupted in HaRxL10 overexpression plants (**Fig. R7b, new Fig. 7a**). To study whether this HaRxL10-induced maltose oscillation perturbation was dependent on CHE, we measured maltose contents in *che-2* and HaRxL10 overexpression in *che-2* plants. Both plants showed strong circadian rhythm of maltose (**Fig. R7c, new Fig. 7b**), suggesting that HaRxL10 requires CHE to perturb maltose oscillation. Hypocotyl growth is another well-characterized circadian-regulated physiological process. We found that overexpressing HaRxL10 in the wild-type plants lengthened the hypocotyl (**Fig. R7d, e; new Fig. 7c, d**). However, HaRxL10 overexpression plants in the *che-2* did not show hypocotyl length difference compared with the *che-2* plants (**Fig. R7d, e; new Fig. 7c, d**), suggesting that HaRxL10-regulated hypocotyl growth was dependent on CHE.

Taken together, our results demonstrated that HaRxL10 could affect several circadian-regulated physiological processes, corroborating its impact on plant clock function. We have added these results and discussion in our revised manuscript (**Page 23, lines 33-45; Page 24, lines 1-2; Page 28, lines 27-29**).

Fig. R7 HaRxL10 affects various circadian-regulated physiological processes. **a**, Rhythm of leaf movement was measured in wild-type (Col-0) and HaRxL10 overexpression (HaRxL10_{ox} #18) *Arabidopsis* seedlings. Plants were grown under 12 h light/12 h dark conditions for 7 days and transferred to the constant light (LL) condition for 3 days. Pictures were then taken once an hour under the LL condition. Distance between two cotyledons was measured to represent the leaf movement. Gray bar, subjective night. The data are shown as mean \pm SEM ($n = 15$). Period and phase were calculated by nonlinear regression. The p values were calculated by unpaired t -test with Welch's correction. This experiment was repeated twice with similar results. **b-c**, Maltose content in 3-week-old wild-type (Col-0) and HaRxL10 overexpression (HaRxL10_{ox} #21) *Arabidopsis* plants (**b**) or *che-2* and HaRxL10 overexpression (HaRxL10_{ox} #4 in *che-2*) *Arabidopsis* plants (**c**). Plants were grown under 12 h light/12 h dark conditions for 3 weeks and transferred to the constant light (LL) condition for 24 hours. Samples were collected at 4 hour-intervals under the LL condition. White bar, subjective day. Gray bar, subjective night. The data are shown as mean \pm SEM ($n = 6$). Statistically significant oscillation of the data was determined by F -test comparing the goodness of fit derived from nonlinear regression using a cosine wave or a straight line ($p < 0.05$). **d-e**, Representative photos (**d**) and statistical analysis of hypocotyl length (**e**) in 7-day-old seedlings of wild-type (Col-0), *che-2* and HaRxL10 overexpression (HaRxL10_{ox}) in different backgrounds (Col-0 or *che-2*). Plants were grown under 12 h light/12 h dark conditions for 4 days and transferred to the constant light (LL) condition for 3 days. Photos were taken and hypocotyl length was measured. The p values were calculated by one-way ANOVA followed by Holm-Šidák's multiple comparisons test. The data are shown as mean \pm SEM ($n = 33$).

I find the data presented in Fig 4d difficult to extrapolate to the infection of a plant by *Hpa*. The level of HaRxL10 when overexpressed in plants does not necessarily relate to biologically relevant levels during an infection. This limitation should be acknowledged.

Response: *Hpa* is an obligate biotrophic pathogen, which extracts nutrients only from

living plant tissue and cannot grow apart from its hosts (**Baxter et al., 2010**). Conventional genetic tools such as plasmid transformation and transposon insertion are usually not applicable to this type of pathogen. *Hpa* has close interactions with its host *Arabidopsis*, and such interactions may increase the difficulty of genetic manipulation, because any genetic manipulation may affect the interactions between the pathogens and their hosts, and even affect the life cycle and pathogenicity. Moreover, exogenous genes, such as antibiotic-resistant genes or fluorescent protein genes may be difficult to be maintained stably. Due to these possible reasons, there has been no report about successful genetic manipulation of *Hpa* to date. To surmount this technical difficulty, previous studies on *Hpa* effector often generated an effector overexpression line of *Arabidopsis* (**Yang et al., 2017**). Therefore, we also utilised this strategy to study the function of HaRxL10 as shown in Fig. 4e (that is Fig. 4d of the previous manuscript).

As this reviewer mentioned, the level of HaRxL10 when overexpressed in plants does not necessarily relate to biologically relevant levels during an infection. Therefore, we employed a natural effector delivery system, the bacterial type-three secretion system (TTSS) of *Pst* DC3000, to deliver HaRxL10 into the plant cells. *Pst* DC3000 is a virulent pathogen for *Arabidopsis*. Introduction of HaRxL10 into *Pst* DC3000 enabled us to study the effect of HaRxL10 by using *Pst* DC3000 as a control pathogen. Several other studies have also utilised *Pst* DC3000 system to study the function of nonbacterial pathogen effectors (**Fabro et al., 2011; Rentel et al., 2008; Sohn et al., 2007**). For example, another *Hpa* RXLR-type effector, HaRxLL470, was found to target HY5 to attenuate the plant immunity where the effector HaRxLL470 was delivered into the host plant by TTSS of *Pst* DC3000 (**Chen et al., 2021**). Using this TTSS-mediated effector delivery system, we found that HaRxL10 also repressed the expression of *CHE* (**Supplementary Fig. 14d**).

Since these two methods of studying the effect of HaRxL10 on *CHE* expression showed similar results, HaRxL10 could likely repress the expression of *CHE*. As this reviewer suggested, we also revised the manuscript to explain the limitation of using overexpression lines (**Page 28, lines 35-45; Page 29, lines 1-6**).

References:

- Baxter, L., Tripathy, S., Ishaque, N., Boot, N., Cabral, A., Kemen, E., Thines, M., Ah-Fong, A., Anderson, R., Badejoko, W., *et al.* (2010). Signatures of adaptation to obligate biotrophy in the *Hyaloperonospora arabidopsidis* genome. *Science* *330*, 1549-1551.
- Chen, S., Ma, T., Song, S., Li, X., Fu, P., Wu, W., Liu, J., Gao, Y., Ye, W., Dry, I.B., *et al.* (2021). *Arabidopsis* downy mildew effector HaRxLL470 suppresses plant immunity by attenuating the DNA-binding activity of bZIP transcription factor HY5. *New Phytol* *230*, 1562-1577.
- Fabro, G., Steinbrenner, J., Coates, M., Ishaque, N., Baxter, L., Studholme, D.J., Korner, E., Allen, R.L., Piquerez, S.J., Rougon-Cardoso, A., *et al.* (2011). Multiple candidate effectors from the oomycete pathogen *Hyaloperonospora arabidopsidis* suppress host plant immunity. *PLoS Pathog* *7*, e1002348.
- Rentel, M.C., Leonelli, L., Dahlbeck, D., Zhao, B., and Staskawicz, B.J. (2008). Recognition of the

Hyaloperonospora parasitica effector ATR13 triggers resistance against oomycete, bacterial, and viral pathogens. Proc Natl Acad Sci U S A *105*, 1091-1096.

Sohn, K.H., Lei, R., Nemri, A., and Jones, J.D. (2007). The downy mildew effector proteins ATR1 and ATR13 promote disease susceptibility in Arabidopsis thaliana. Plant Cell *19*, 4077-4090.

Yang, L., Teixeira, P.J.P.L., Biswas, S., Finkel, O.M., He, Y.J., Salas-Gonzalez, I., English, M.E., Epple, P., Mieczkowski, P., and Dangl, J.L. (2017). Pseudomonas syringae Type III Effector HopBB1 Promotes Host Transcriptional Repressor Degradation to Regulate Phytohormone Responses and Virulence. Cell Host & Microbe *21*, 156-168.

Furthermore, what are the effects of overexpression of HaRxL10 on the expression of other clock genes and TCP genes?

Response:

We analysed the expression of two key central clock genes, *CCA1* and *TOC1*, in the wild-type and HaRxL10 overexpression plants. Our results showed that HaRxL10 overexpression plants displayed lower amplitude and average expression of *CCA1* compared to the wild-type plants (**Fig. R8a, new Fig. 6a**). Period and phase of *CCA1* did not show significant difference in the wild-type and HaRxL10 overexpression plants (**Fig. R8a, new Fig. 6a**). HaRxL10 overexpression plants displayed higher amplitude and average expression of *TOC1* compared to the wild-type plants (**Fig. R8b, new Supplementary Fig. 16**). Period and phase of *TOC1* did not show significant difference in the wild-type and HaRxL10 overexpression plants (**Fig. R8b, new Supplementary Fig. 16**). Moreover, we analysed the expression of a widely-used clock output gene, *CAB2*. Our results showed that the period of *CAB2* was shorter in HaRxL10 overexpression plants compared to the wild-type plants (**Fig. R8c, new Fig. 6e**). HaRxL10 overexpression plants also displayed lower amplitude and average expression of *CAB2* (**Fig. R8c, new Fig. 6e**). Phase of *CAB2* did not show significant difference in the wild-type and HaRxL10 overexpression plants (**Fig. R8c, new Fig. 6e**).

Fig. R8 Overexpression of HaRxL10 affects the expression of several clock genes. Relative expression levels of *CCA1* (a), *TOC1* (b) and *CAB2* (c) in 3-week-old wild-type (Col-0) and HaRxL10 overexpression (HaRxL10_{ox} #18) *Arabidopsis* plants. Plants were grown under 12 h light/12 h dark conditions for 3 weeks and transferred to the constant light (LL) condition for 24 hours. Samples were collected at 4 hour-intervals under the LL condition and analysed by RT-qPCR with *UBQ5* as the internal control. White bar, subjective day. Gray bar, subjective night. The data are shown as mean \pm SEM (n = 3). Period, phase, amplitude and average expression were calculated by nonlinear regression. The *p* values were calculated by unpaired *t*-test with Welch's correction. This experiment was repeated twice with similar results.

TCP family TFs contain 24 members in *Arabidopsis*, which are grouped into Class I and Class II subfamilies. Since HaRxL10 interacted with CHE (also known as TCP21) and regulated its transcription, it is possible that HaRxL10 also affects other TCP gene expressions. We selected several representative TCP genes to explore this possibility. Different TCP genes show diverse daily expression patterns. Therefore, we performed a 24-hour time-course RT-qPCR experiment to analyse the effects of overexpressing HaRxL10 on expression of TCPs. TCP7 belongs to Class I TCPs and shares the highest the sequence similarity with CHE. Moreover, TCP7 interacts with HaRxL10 in yeast (**Supplementary Fig. 20**). Our results showed that overexpression of HaRxL10 significantly affected the expression pattern of *TCP7* (**Fig. R9a**; **Supplementary Fig. 21a**). During the subjective day, HaRxL10 repressed *TCP7*. While during the subjective night, HaRxL10 promoted *TCP7*. TCP20, a Class I TCP, is a clock component. Overexpression of HaRxL10 did not significantly affect the expression of *TCP20* (**Fig.**

R9b; Supplementary Fig. 21b). TCP3 and TCP12 are both classified as Class II TCPs. Specifically, TCP3 is a member of CIN clade and TCP12 is a member of CYC/TB1 clade. Our time-course RT-qPCR results showed that HaRxL10 significantly repressed the expression of *TCP3* and *TCP12* (Fig. R9c, d; Supplementary Fig. 21c, d). Collectively, our results suggest that overexpression of HaRxL10 could affect the expression of several Class I and Class II TCP genes.

Fig. R9 Overexpression of HaRxL10 affects the expression of several TCP genes. a-d, Relative expression levels of *TCP7* (a), *TCP20* (b), *TCP3* (c), and *TCP12* (d) in 3-week-old wild-type (Col-0) and HaRxL10 overexpression (HaRxL10_{ox} #18) *Arabidopsis* plants. Plants were grown under 12 h light/12 h dark conditions for 3 weeks and transferred to the constant light (LL) condition for 24 hours. Samples were collected at 4 hour-intervals under the LL condition and analysed by RT-qPCR with *UBQ5* as the internal control. White bar, subjective day. Gray bar, subjective night. The data are shown as mean \pm SEM (n = 3). The p values were calculated by two-way ANOVA. This experiment was repeated twice with similar results.

The same comments about leaves shown in Fig 1 apply to those in Fig 5 a.

Response: We apologise for the unclear description in the figure legend. The pictures shown in Fig. 5a were the 4th and 5th leaves from three representative plants 3 days post bacterial infiltration. We have added above explanation in the figure legend in our revised manuscript (Page 18, lines 7-8).

What was the time of day of inoculation of leaves with in Pst DC3000 with and without HaRxL10: CHE is expressed with a peak at mid-day – is that important in the assay?

Response: We infiltrated *Pst* DC3000 with or without HaRxL10 at ZT0. Since it would take some time for HaRxL10 protein to be synthesized and delivered into the plant cells, HaRxL10 is likely to be delivered into the host cells around ZT8, which is the peak expression time of CHE. Moreover, a previous study has revealed that *Arabidopsis* plants display a temporal difference in defence against *Pst* DC3000 and respond more effectively to pathogen challenges in the daytime (**Bhardwaj et al., 2011**). Therefore, infiltration of pathogen in the morning may amplify the effects of HaRxL10 on the plants.

Reference:

Bhardwaj, V., Meier, S., Petersen, L.N., Ingle, R.A., and Roden, L.C. (2011). Defence responses of *Arabidopsis thaliana* to infection by *Pseudomonas syringae* are regulated by the circadian clock. *PLoS One* 6, e26968.

Can the authors comment on the fold-difference in cfu per leaf disc in *che-2* vs Col-0 in *Pst* DC3000 without HaRxL10 (Fig 5.b and c)?

Response: We noticed that *che-2* showed enhanced disease susceptibility to *Pst* DC3000 without HaRxL10. A previous study found that CHE positively regulates the expression of SA synthesis gene, *ICS1* (**Zheng et al., 2015**); the co-first author of this study is the corresponding author of this manuscript. Our results further showed that CHE-induced genes are enriched with defence response genes (**Supplementary Fig. 13**), indicating that CHE positively regulates innate immunity. We have commented on this result in our revised manuscript (**Page 17, lines 24-27**).

Reference:

Zheng, X.Y., **Zhou, M.**, Yoo, H., Pruneda-Paz, J.L., Spivey, N.W., Kay, S.A., and Dong, X. (2015). Spatial and temporal regulation of biosynthesis of the plant immune signal salicylic acid. *Proc Natl Acad Sci U S A* 112, 9166-9173.

The authors state that HaRxL10 affects the expression of central clock genes. I am not convinced of the ‘centrality’ of the genes are stated to be affected. More importantly, the effect on TCP22 is not seen until 36 hpi, while the effects are detectable for LWD1 earlier, at 16-20 hpi, but appear to be similar at all other time points, but these subtleties are not commented on.

Response: *CHE*, *TCP22*, *RVE4* and *LWD1* were four clock genes whose expressions were affected by *Pst* DC3000-HaRxL10 in the time-course RNA-seq experiment (**Supplementary Fig. 14**). However, the effect of HaRxL10 on the clock gene expression may be diminished under diurnal conditions used in the RNA-seq experiment. To further study the effect of HaRxL10 on the clock gene expression, we analysed the circadian expression patterns of two key central clock genes, *CCA1* and *TOC1*, in the wild-type and HaRxL10 overexpression plants under the constant light condition. Our results showed that HaRxL10 repressed the expression of *CCA1* and

promoted the expression of *TOC1* (**new Fig. 6a, new Supplementary Fig. 16**). These effects were much greater than what we observed in the expression changes of *TCP22* and *LWD1* triggered by HaRxL10.

The plant circadian clock consists of multiple transcriptional-translational feedback loops. Because of this complicated self-regulatory network, clock genes usually respond to environmental input in a non-linear way. For example, the gene expressions of *CCA1* and *TOC1* were affected by exogenous SA treatment at different times (**Zhou et al., 2015**). As this reviewer pointed out, HaRxL10 affected *TCP22* expression at later time points and affected *LWD1* expression at earlier time points, which is common for clock genes. We have revised our manuscript accordingly to comment on these results (**Page 19, lines 40-45; Page 20, lines 1-3**).

Reference:

Zhou, M., Wang, W., Karapetyan, S., Mwimba, M., Marques, J., Buchler, N.E., and Dong, X. (2015). Redox rhythm reinforces the circadian clock to gate immune response. *Nature* 523, 472-476.

What are the consequences for the clock and temporal regulation of defences and metabolism, if any?

Response: We analysed the expression of the central clock gene *CCA1* and a widely-used clock output gene *CAB2* in the wild-type and HaRxL10 overexpression plants. Our results showed that HaRxL10 overexpression plants displayed lower amplitude and average expression of *CCA1* compared to the wild-type plants (**Fig. R10a, new Fig. 6a**). Period and phase of *CCA1* did not show significant difference in the wild-type and HaRxL10 overexpression plants (**Fig. R10a, new Fig. 6a**). The period of *CAB2* was shorter in HaRxL10 overexpression plants compared to the wild-type plants (**Fig. R10b, new Fig. 6e**). HaRxL10 overexpression plants also displayed lower amplitude and average expression of *CAB2* (**Fig. R10b, new Fig. 6e**). Phase of *CAB2* did not show significant difference in the wild-type and HaRxL10 overexpression plants (**Fig. R10b, new Fig. 6e**).

To further assess the effect of HaRxL10 on the clock function, we conducted leaf movement assay and measured maltose content and hypocotyl length under the constant light condition. Our results showed that period of leaf movement rhythms was shorter in HaRxL10 overexpression plants than the wild-type plants (**Fig. R10c**). Phase of leaf movement rhythms was delayed in HaRxL10 overexpression plants (**Fig. R10c**). In addition to leaf movement, the circadian clock also regulates contents of some metabolites, such as maltose. We found that maltose levels showed a strong circadian rhythm in the wild-type plants, but this rhythm was disrupted in HaRxL10 overexpression plants (**Fig. R10d, new Fig. 7a**). To study whether this HaRxL10-induced maltose oscillation perturbation was dependent on CHE, we measured maltose contents in *che-2* and HaRxL10 overexpression in *che-2* plants. Both plants showed strong circadian rhythm of maltose (**Fig. R10e, new Fig. 7b**), suggesting that HaRxL10

requires CHE to perturb maltose oscillation. Hypocotyl growth is another well-characterized circadian-regulated physiological process. We found that overexpressing HaRxL10 in the wild-type plants lengthened the hypocotyl (Fig. R10f, g; new Fig. 7c, d). However, HaRxL10 overexpression plants in the *che-2* did not show hypocotyl length difference with the *che-2* plants (Fig. R10f, g; new Fig. 7c, d), suggesting that HaRxL10-regulated hypocotyl growth was dependent on CHE.

Taken together, our results demonstrated that HaRxL10 could affect several circadian-regulated physiological processes, corroborating its impact on plant clock function. We have revised our manuscript accordingly (Pages 19 to 24).

Fig. R10 HaRxL10 affects circadian-regulated physiological processes. a-b, Relative expression levels of *CCA1* (a) and *CAB2* (b) in 3-week-old wild-type (Col-0) and HaRxL10 overexpression (HaRxL10_{ox} #18) *Arabidopsis* plants. Plants were grown under 12 h light/12 h dark conditions for 3 weeks and transferred to the constant light (LL) condition for 24 hours. Samples were collected at 4 hour-intervals under the LL condition and analysed by RT-qPCR with *UBQ5* as an internal control. White bar, subjective day. Gray bar, subjective night. The data are shown as mean \pm SEM (n = 3).

Period, phase, amplitude and average expression were calculated by nonlinear regression. The p values were calculated by unpaired t -test with Welch's correction. This experiment was repeated twice with similar results. **c**, Rhythm of leaf movement was measured in wild-type (Col-0) and HaRxL10 overexpression (HaRxL10_{ox} #18) *Arabidopsis* seedlings. Plants were grown under 12 h light/12 h dark conditions for 7 days and transferred to the constant light (LL) condition for 3 days. Pictures were then taken once an hour under the LL condition. Distance between two cotyledons was measured to represent the leaf movement. Gray bar, subjective night. The data are shown as mean \pm SEM ($n = 15$). Period and phase were calculated by nonlinear regression. The p values were calculated by unpaired t -test with Welch's correction. This experiment was repeated twice with similar results. **d-e**, Maltose contents in 3-week-old wild-type (Col-0) and HaRxL10 overexpression (HaRxL10_{ox} #21) *Arabidopsis* plants (**d**) or *che-2* and HaRxL10 overexpression (HaRxL10_{ox} #4 in *che-2*) *Arabidopsis* plants (**e**). Plants were grown under 12 h light/12 h dark conditions for 3 weeks and transferred to the constant light (LL) condition for 24 hours. Samples were collected at 4 hour-intervals under the LL condition. White bar, subjective day. Gray bar, subjective night. The data are shown as mean \pm SEM ($n = 6$). Statistically significant oscillation of the data was determined by F -test comparing the goodness of fit derived from nonlinear regression using a cosine wave or a straight line ($p < 0.05$). **f-g**, Representative images (**f**) and statistical analysis of hypocotyl length (**g**) in the 7-day-old seedlings of wild-type (Col-0), *che-2* and HaRxL10 overexpression (HaRxL10_{ox}) in different backgrounds (Col-0 or *che-2*). Plants were grown under 12 h light/12 h dark conditions for 4 days and transferred to the constant light (LL) condition for 3 days. Photos were taken and hypocotyl length was measured. The p values were calculated by one-way ANOVA followed by Holm-Šidák's multiple comparisons test. The data are shown as mean \pm SEM ($n = 33$).

I am not totally convinced by the data that the statement that “HaRxL10-triggered repression of RVE4 and LWD1 is partially dependent on CHE” is correct, partly because of the measurement at 24 hpi without a reference to the ZT of infection.

Responses: We infected wild-type and *che-2* *Arabidopsis* plants with *Pst* DC3000 or *Pst* DC3000-HaRxL10 at ZT0. We collected infected leaves at 24 hpi, that is ZT24, for gene expression analysis. We have added this information in our revised manuscript (**SI, figure legend of Supplementary Fig. 15d-f**).

Our RNA-seq results showed that *Pst* DC3000-HaRxL10 could not trigger significant gene expression change of *TCP22* compared to the control *Pst* DC3000 infiltration in *che-2* plants ($p > 0.05$, *che-2* *Pst* DC3000 vs. *che-2* *Pst* DC3000-HaRxL10), suggesting that HaRxL10-triggered repression of *TCP22* required CHE (**Supplementary Fig. 15a-c**). However, *Pst* DC3000-HaRxL10 could still trigger significant gene expression changes of *RVE4* and *LWD1* compared to the control *Pst* DC3000 infiltration in *che-2* plants (**Supplementary Fig. 15a-c**). The subsequent single time-point RT-qPCR result supported that HaRxL10-triggered repression of *TCP22* is dependent on CHE

(**Supplementary Fig. 15d**). RT-qPCR results showed that although the reduction in gene expressions of *RVE4* and *LWD1* were less pronounced in *che-2* mutants compared to wild-type plants, *Pst* DC3000-HaRxL10 could still significantly repress the expression of *RVE4* and *LWD1* (**Supplementary Fig. 15e, f**). Considering the probably marginal effects of CHE on HaRxL10-mediated expression changes of *RVE4* and *LWD1*, we revised our conclusion to emphasise the effect of CHE on *TCP22* but not *RVE4* and *LWD1* (**Page 19, lines 40 to 45; Page 20, lines 1 to 3**).

There are a number of typos on p17: line15 found instead of fine, line 25 regulating instead of regulate, line 30 expressions instead of expression

Response: We apologise for these mistakes. We have revised the manuscript accordingly (**Page 20, line 26; Page 20, line 38; Page 21, line 13**).

It is not clear what time of day samples were collected for the data shown in Fig 6 g.

Response: We apologise for unclear description. These samples were collected at ZT0. We have revised the manuscript accordingly (**Page 23, line 5**).

While overexpression of HaRxL10 affects the levels of *CCA1* expression, it seems overly speculative to extrapolate this finding to the role of HaRxL10 to a natural infection. This limitation must be acknowledged.

Response: We agree with this reviewer that the ectopic overexpression of a pathogen effector in plants may not accurately reflect the effector's natural function probably due to its higher abundance in the overexpression plants. To further study the effect of HaRxL10 on *CCA1* expression, we used *Pst* DC3000 TTSS-mediated natural delivery of HaRxL10. Introduction of HaRxL10 into *Pst* DC3000 enabled us to study the effect of HaRxL10 by using *Pst* DC3000 as a control pathogen. Several other studies have also utilised *Pst* DC3000 system to study the function of nonbacterial pathogen effectors (**Fabro et al., 2011; Rentel et al., 2008; Sohn et al., 2007**). We found that infiltration of *Pst* DC3000-HaRxL10 could repress *CCA1* expression compared to *Pst* DC3000 at 24 hpi (**Fig. R11**). Since these two methods generated similar results, it is likely that HaRxL10 affects *CCA1* expression in a natural infection. We provided more discussion of the limitation of using overexpression lines as the reviewer suggested (**Page 28, lines 35-45; Page 29, lines 1-6**).

Fig. R11 HaRxL10 represses the expression of CCA1. Relative gene expression level of *CCA1* in the wild-type (Col-0) *Arabidopsis* plants at 24 hours after *Pst* DC3000 or *Pst* DC3000-HaRxL10 infection analysed by RT-qPCR with *UBQ5* as an internal control. The pathogen infection assay was conducted at ZT0. The data are shown as mean \pm SEM ($n = 9$, 3 independent experiments with 3 technical replicates). The p value was calculated by a two-sided unpaired Student's t -test.

References:

- Fabro, G., Steinbrenner, J., Coates, M., Ishaque, N., Baxter, L., Studholme, D.J., Korner, E., Allen, R.L., Piquerez, S.J., Rougon-Cardoso, A., *et al.* (2011). Multiple candidate effectors from the oomycete pathogen *Hyaloperonospora arabidopsidis* suppress host plant immunity. *PLoS Pathog* *7*, e1002348.
- Rentel, M.C., Leonelli, L., Dahlbeck, D., Zhao, B., and Staskawicz, B.J. (2008). Recognition of the *Hyaloperonospora parasitica* effector ATR13 triggers resistance against oomycete, bacterial, and viral pathogens. *Proc Natl Acad Sci U S A* *105*, 1091-1096.
- Sohn, K.H., Lei, R., Nemri, A., and Jones, J.D. (2007). The downy mildew effector proteins ATR1 and ATR13 promote disease susceptibility in *Arabidopsis thaliana*. *Plant Cell* *19*, 4077-4090.

I do not think that the authors can conclude that HaRxL10 globally affects circadian-controlled genes as there is no Col-0 mock infection expression analyses for comparison.

Response: We compared the effects of *Pst* DC3000 and *Pst* DC3000-HaRxL10 on the circadian-controlled gene expressions. Introduction of HaRxL10 into *Pst* DC3000 enabled us to study the virulence of HaRxL10 by using *Pst* DC3000 as a control pathogen. In this case, mock infection was not a good control because *Pst* DC3000 infection alone could trigger transcriptome change.

Where all experiments carried out in LD, and if so how were samples for RNA analyses collected during the dark periods?

Response: Plants for RNA-seq analysis were individually grown in individual pots in a

growth chamber. Pots for different treatments were put on different layers of the growth chamber so that each pot could be quickly located and taken out from the growth chamber for sample collection. The plant samples were then snap-frozen in liquid nitrogen and stored at -80°C. During dark periods, the lights of both the growth chamber and the lab were shut down. The very dimmed light from outside of the lab enabled the visualisation of the sample collection process. These very dimmed lights may affect the plants inside the growth chamber when opening the door of the growth chamber to take out of the pot. Since the duration time of opening the growth chamber was very short (usually took only a few seconds) and the sample collection time was also very short (within 1 minute) each time, these very dimmed light effects would be neglectable.

Why were the times ZT20 in long days and ZT0 in SD chosen for measuring transcript levels? Was this why FT was not detected in Short days? Why were expression levels measured only in 10 day old plants?

Response: *FT* is a key floral signal integrator gene, which plays a key role in photoperiod pathway-promoted flowering. For *Arabidopsis* Col-0, long-day (LD) condition promotes *FT* expression and flowering. Therefore, *FT* is always used as a marker gene to represent flowering time of *Arabidopsis* growing under the LD but not under the short-day (SD) condition. Previous study showed that *FT* displays a daily rhythmic expression with high expression during night (Cheng et al., 2020). Therefore, we collected samples growing under LD at ZT20 for *FT* expression analysis.

Floral transition in *Arabidopsis* growing under the LD condition occurs 9 to 13 days after germination and *FT* gene has been highly expressed during this developmental stage (Shen et al., 2011). The *FT* gene expression level at this development stage correlates well with flowering time. Due to this reason, previous studies usually use 9 to 11-day-old seedlings to analyse *FT* gene expression (Cheng et al., 2020; Shen et al., 2011). In this study, we used 10-day-old seedlings and successfully detected *FT* expression under the LD condition (Supplementary Fig. 17a).

Under the SD condition, *FT* does not play a critical role. Therefore, we chose to analyse expressions of the main flowering-suppression gene, *FLC*, along with its homologues *MAF4* and *MAF5* (Li et al., 2018). These flowering repressor genes function in autonomous pathway and could be detected in young seedlings growing under the SD condition. These genes do not display strong rhythmic expression. Therefore, we collected samples at the normal working time in the morning (ZT0). Therefore, we collected samples at night (ZT20) specifically for *FT* analysis. We revised our manuscript to indicate the sample collection time difference for gene specific reasons (SI, figure legend in Supplementary Fig. 17).

Reference:

Cheng, Z., Zhang, X., Huang, P., Huang, G., Zhu, J., Chen, F., Miao, Y., Liu, L., Fu, Y.F., and Wang, X. (2020). Nup96 and HOS1 Are Mutually Stabilized and Gate CONSTANS Protein Level, Conferring

Long-Day Photoperiodic Flowering Regulation in Arabidopsis. *Plant Cell* 32, 374-391.

Li, Z., Ou, Y., Zhang, Z., Li, J., and He, Y. (2018). Brassinosteroid Signaling Recruits Histone 3 Lysine-27 Demethylation Activity to FLOWERING LOCUS C Chromatin to Inhibit the Floral Transition in Arabidopsis. *Mol Plant* 11, 1135-1146.

Shen, L., Kang, Y.G., Liu, L., and Yu, H. (2011). The J-domain protein J3 mediates the integration of flowering signals in Arabidopsis. *Plant Cell* 23, 499-514.

Discussion

Some of the data have been interpreted without proper consideration of the limitations of the experimental design. Many of the data were collected from experiments with heterologously overexpressed genes in bacteria or plants. This may not reflect the activity of the putative effector in an *Hpa* infection.

Response: We have added limitation of this study in the Discussion part of our revised manuscript (**Page 28, lines 35-45; Page 29, lines 1-6**). *Hpa* is an obligate biotrophic pathogen, which extracts nutrients only from living plant tissue and cannot grow apart from its hosts. Conventional genetic tools such as plasmid transformation and transposon insertion are usually not applicable to this type of pathogen, as Reviewer #1 mentioned that there are “difficulties in altering the function of HaRxL10 in *Hpa*”. To partially overcome this technical difficulty, we used TTSS of *Pst* DC3000 to deliver HaRxL10 into plant cells. This natural effector delivery system may mimic *Hpa* infection to some extent and has been widely used in the field.

The effects of the HaRxL10 on the other TCP proteins it was demonstrated to interact with have not been considered. The roles of other TCPs may underpin some of the effects seen.

Response: Our Y2H results showed that HaRxL10 could interact with 10 *Arabidopsis* TCPs in yeast, including 7 Class I TCPs (TCP6, 7, 9, 11, 14, 23 and CHE) and 3 Class II TCPs (TCP3, 12 and 17). TCP12 is a CYC/TB1 TCP. TCP3 and TCP17 belong to CIN TCPs. These results indicated that HaRxL10 could target multiple TCP proteins in addition to CHE to generate some unexplored effects. We found that HaRxL10 triggered a large-scale transcriptome reprogramming, among which 79.7% of HaRxL10-induced and 84.7% of HaRxL10-repressed genes are not dependent on CHE (**Supplementary Fig. 22**). These effects may due to HaRxL10's interaction with other TCPs but need further investigation. We have revised our manuscript accordingly to discuss the potential effect of HaRxL10 interacting with other TCP proteins. Moreover, we added data showing the effects of HaRxL10 on the expression of some representative TCP genes (**Page 27, lines 14-22**).

The conclusion that CHE may be a candidate for genetic modification to improve disease resistance may be premature.

Response: CHE regulates both flowering and plant immunity but could be targeted by the effector during pathogen infection. CHE may use different domains for protein interaction and DNA-binding function respectively. Our Y2H and BiFC results showed that CHE_C interacted with HaRxL10 while CHE_{ΔC} (deletion of C-terminal) did not interact with HaRxL10 (**Fig. 2c, d**), suggesting that C-terminal of CHE mediates protein interaction with HaRxL10. Previous studies showed that TCP domain functions in DNA binding (**Cubas et al., 1999**). Therefore, it is possible that CHE_{ΔC}, which contains TCP domain, maintains DNA-binding ability and cannot be recognized by HaRxL10. This CHE mutant plant may escape from pathogen attack on the host circadian clock and maintain plant regular growth and defence. We agree that this is the future direction and revised our manuscript to tune down the description (**Page 2, line 12-13; Page 29, lines 9-17**).

Reference:

Cubas, P., Lauter, N., Doebley, J., and Coen, E. (1999). The TCP domain: a motif found in proteins regulating plant growth and development. *Plant J* 18, 215-222.

Methods

Bacterial assay and *Phytophthora sojae* assays should specify which leaf of the plants were chosen for inoculation. Matching developmental stage for infections can minimise variation.

Response: We have added the leaf information in related figure legends (**Page 5, lines 18-19; Page 5, 29-30; Page 18, lines 7-8**). For the same experiment, we used the same developmental plants and leaves for infection.

Bacteria are not in solution, but in suspension – this needs correction throughout.

Response: Thank you for pointing out this mistake. We have revised our manuscript accordingly (**Page 32, line 24; Page 34, lines 20 and 30**)

Y2H assay is missing information for all the TCP genes used.

Response: We added the TCP genes information in the “plasmid constructs” and “Y2H assay” parts under the Methods section in our revised manuscript (**Page 29, lines 24-31; Page 34, lines 4-6**).

For the BiFC assay, were *Agrobacterium tumefaciens* cells really washed in ddH₂O prior to resuspension in tobacco buffer?

Response: *Agrobacterium tumefaciens* cells were washed in tobacco buffer not in ddH₂O. We have revised our manuscript accordingly (**Page 34, lines 16-17**)

Type on p27 line 40 Agrobacteria should be Agrobacterium or simply A. tumefaciens

Response: Thank you for pointing out this mistake. We have revised our manuscript accordingly (**Page 34, line 25**)

Reviewer #3:

Recent studies have established the crosstalk between the circadian clock and plant innate immunity. This manuscript reports a new mechanism underlying such a crosstalk that an oomycete effector HaRxL10 directly targets Arabidopsis central clock component CCA1 HIKING EXPEDITION (CHE) in suppressing CHE's function in circadian clock, defense, as well as flowering.

Response: Thank you for recognition of this study.

Interestingly, HaRxL10 stabilizes CHE at the protein level but inhibit its transcription and transcriptional function. How to resolve such a paradox is unclear.

Response: Our results suggested that the binding of CHE to its target gene promoter was significantly repressed by HaRxL10 (**Fig. 4g, new Fig. 4i**). Although HaRxL10 stabilises CHE, HaRxL10 also interferes with the DNA binding ability of CHE thus suppressing the transcriptional regulation function of CHE. One possibility is that the binding of HaRxL10 with CHE blocks the DNA binding site of CHE since TCP domain is important for both DNA binding and protein interactions.

A strong link of HaRxL10 to clock and defense regulation via CHE remains to be fully established.

Response: We performed more experiments to further assess the effect of HaRxL10 on the clock function and the dependency of CHE. The circadian clock regulates leaf movement, maltose content, and hypocotyl growth. We conducted leaf movement assay and measured maltose content and hypocotyl length in the wild-type and HaRxL10 overexpression plants (**Fig. R12, new Fig. 7a-d**). The results collectively suggested that HaRxL10 affects the clock function in a CHE-dependent way. Please refer to our revised manuscript for details (**Page 23, lines 33-45; Page 24 lines 1-2**).

We performed pathogen infection assays and RT-qPCR experiments to show that HaRxL10 represses the expression of SA-related defence genes and enhances disease susceptibility in a CHE-dependent manner (**Fig. 5**).

Fig. R12 HaRxL10 affects various circadian-regulated physiological processes. **a**, Rhythm of leaf movement was measured in wild-type (Col-0) and HaRxL10 overexpression (HaRxL10_{ox} #18) *Arabidopsis* seedlings. Plants were grown under 12 h light/12 h dark conditions for 7 days and transferred to the constant light (LL) condition for 3 days. Pictures were then taken once an hour under the LL condition. Distance between two cotyledons was measured to represent the leaf movement. Gray bar, subjective night. The data are shown as mean \pm SEM ($n = 15$). Period and phase were calculated by nonlinear regression. The p values were calculated by unpaired t -test with Welch's correction. This experiment was repeated twice with similar results. Maltose contents in 3-week-old wild-type (Col-0) and HaRxL10 overexpression (HaRxL10_{ox} #21) *Arabidopsis* plants (**b**) or *che-2* and HaRxL10 overexpression (HaRxL10_{ox} #4 in *che-2*) *Arabidopsis* plants (**c**). Plants were grown under 12 h light/12 h dark conditions for 3 weeks and transferred to the constant light (LL) condition for 24 hours. Samples were collected at 4 hour-intervals under the LL condition. White bar, subjective day. Gray bar, subjective night. The data are shown as mean \pm SEM ($n = 6$). Statistically significant oscillation of the data was determined by F -test comparing the goodness of fit derived from nonlinear regression using a cosine wave or a straight line ($p < 0.05$). Representative images (**d**) and statistical analysis of hypocotyl length (**e**) in the 7-day-old seedlings of wild-type (Col-0), *che-2* and HaRxL10 overexpression (HaRxL10_{ox}) in different backgrounds (Col-0 or *che-2*). Plants were grown under 12 h light/12 h dark conditions for 4 days and transferred to the constant light (LL) condition for 3 days. Photos were taken and hypocotyl length was measured. The p values were calculated by one-way ANOVA followed by Holm-Šidák's multiple comparisons test. The data are shown as mean \pm SEM ($n = 33$).

A large body of data were presented in this ms. The authors also made many useful tools to study the interaction between HaRxL10 and CHE. One set of such tools is stable transgenic *Arabidopsis* seedlings expressing CHEp:CHE-3 \times HA in *che-2* (CHE-HA) with or without 35S:YFP-HaRxL10 in CHE-HA (two independent homozygous T3 lines) as presented in Figure 2e. These lines were used to show that HaRxL10 stabilizes CHE at the protein level. In addition to using different expression system, this same

system could be further used to show whether HaRxL10 indeed suppresses CHE transcriptional function, including suppressing CHE target genes for their RNA expression (including CHE own transcription) and binding of CHE to target gene promoters.

Response: As this reviewer suggested, we analysed the expressions of CHE target genes, *CHE* and *ICS1* in CHE-HA and two independent 35S:YFP-HaRxL10 CHE-HA lines. Our results showed that the expression of endogenous *CHE* was decreased when overexpressing HaRxL10 in CHE-HA (Fig. R13a). Furthermore, we performed a ChIP-qPCR experiment using CHE-HA and 35S:YFP-HaRxL10 CHE-HA lines and showed that the binding of CHE to its promoter region was inhibited by overexpression of HaRxL10 (Fig. R13b, c; new Fig. 4i). These data indicated that HaRxL10 suppressed the transcriptional function of CHE.

Fig. R13 HaRxL10 suppresses the transcriptional function of CHE. **a**, The expression of *CHE* in 7-day-old transgenic *Arabidopsis* seedlings expressing *CHEp::CHE-3×HA* in *che-2* (CHE-HA) and *35S:YFP-HaRxL10* in CHE-HA (two independent homozygous T3 lines) was analysed by RT-qPCR with *UBQ5* as an internal control and normalised by the expression in CHE-HA seedlings. Samples were collected at ZT8. The data are shown as mean ± SEM (n = 9, 3 independent experiments with 3 technical replicates). The *p* values were calculated by one-way ANOVA followed by Holm-Šídák's multiple comparisons test. **b**, Illustration of primers used for the ChIP-qPCR experiment. **c**, HaRxL10 inhibited the binding of CHE to the *CHE* promoter revealed by ChIP-qPCR. Seven-day-old transgenic *Arabidopsis* seedlings expressing *CHEp::CHE-3×HA* in *che-2* (CHE-HA) or *35S:YFP-HaRxL10* in CHE-HA (line #14) were used. Anti-HA antibody was used. Data represent the mean ± SEM (n = 3). **, *p* < 0.01; ns, not significant (Holm-Šídák multiple comparisons test). This experiment was repeated three times with similar results.

additional specific comments:

1. Fig. 3: Fig. 3e showed an almost complete abolishment of CHE and ZTL interaction in the presence of HaRxL10. However, Fig. 3d did not clearly show this point. In absence of 3AT, HaRxL10 did not affect the interaction between CHE and ZTL,

compared with the control YFP. In the presence of 3AT, the interaction of CHE and ZTL appeared to be reduced in the presence of HaRxL10 or YFP.

Response: We agree with this reviewer that in the absence of 3-AT, the effect of HaRxL10 on the interaction between CHE and ZTL was not obvious when the yeast plating concentration was high ($OD_{600\text{ nm}} = 1$). To amplify this effect, we showed the pictures with lower yeast plating concentrations ($OD_{600\text{ nm}} = 0.1$ and 0.01 respectively), which clearly showed that the interaction between CHE and ZTL was significantly weaker in the presence of HaRxL10 compared to the control YFP protein (**Fig. R14; new Fig. 3d**). We have updated this figure and related description of this result in our revised manuscript (**Page 10, lines 12-16**).

As this reviewer pointed out, the interaction of CHE and ZTL appeared to be reduced in the presence of HaRxL10 or YFP when 3-AT was applied. It was probably due to the general effect of 3-AT on yeast growth. In this case, the interaction of CHE and ZTL was completely blocked by HaRxL10 while the control YFP protein did not significantly affect the CHE-ZTL interaction, further indicating that HaRxL10 specifically suppresses the CHE-ZTL interaction.

Fig. R14 Y3H assays illustrating that HaRxL10 significantly inhibits the interaction between CHE and ZTL. Synthetic dropout medium without leucine, tryptophan and methionine (SD/-Leu-Trp-Met) was used for positive yeast transformant selection. Synthetic dropout medium without leucine, tryptophan, methionine and histidine (SD/-Leu-Trp-Met-His) was used for selection of protein interaction by the reporter gene *HIS3*. 3-AT was used to inhibit the self-activation in yeast. Photographs were taken 3 days after plating of yeast cells with $OD_{600\text{ nm}} = 1$, 0.1 or 0.01 respectively. This experiment was repeated three times with similar results.

In light of the result shown in Fig 3e, perhaps His-HaRxL10 should be included in Fig. 3a to clearly demonstrate HaRxL10 disruption of CHE and ZTL interaction, a likely cause for CHE stabilization in the presence of HaRxL 10.

Response: As this reviewer suggested, we studied the effect of His-HaRxL10 on the interaction between MBP-His-CHE and GST-His-ZTL by a pull-down assay. Our results showed that the interaction between MBP-His-CHE and GST-His-ZTL was inhibited by His-HaRxL10 in a dose-dependent manner (**Fig. R15, new Fig. 3e**), suggesting that HaRxL10 disrupts the CHE-ZTL interaction. We have added this result in our revised manuscript (**Page 10, lines 16-21**).

Fig. R15 His-HaRxL10 inhibits the interaction between MBP-His-CHE and GST-His-ZTL. Protein combinations were pulled down by MBP beads followed by Western blot with α -MBP, α -GST and α -His antibodies respectively. The molecular weight of MBP-His-CHE is 70.1 kD. The molecular weight of GST-His-ZTL is 99 kD. The molecular weight of His-HaRxL10 is 28.5 kD. This experiment was repeated three times with similar results.

2. YFP-HaRxL10 was overexpressed in Col-0 and *che-2* mutant. Two YFP-HaRxL10/Col-0 lines, #18 and #21 were used in this report. While #18 showed much higher expression of YFP-HaRxL10 than #21, both lines were reported to have reduced CHE transcripts and early flowering time (Fig 4d and Fig 7), compared with Col-0. However, line #21 showed wt-like bacterial growth (Fig. s9c) and even higher than Col-0 expression of the flowering repressor gene, *FLC* (Fig. 7). Such discrepancies might be due to other reasons than just expression difference of HaRxL10 between the two lines. Perhaps some other lines could be examined to clarify the point.

Response: Based on our current results, it seems that the disease symptoms have a stronger correlation with the dosage of HaRxL10. HaRxL10ox #18 which showed higher expression of HaRxL10 displayed more severe disease phenotypes of both *Phytophthora sojae* (**Fig. 1g, h**) and *Pseudomonas syringae* (**Supplementary Fig. 10c**) infections. The effects of overexpressing HaRxL10 on flowering time are not correlated to the expression levels of *HaRxL10*. Both high and low overexpression lines of HaRxL10 displayed a similar early flowering phenotype. The fact that HaRxL10ox #21 showing higher *FLC* expression than the wild-type plants may be due to the specific location of the T-DNA insertion in this line. However, HaRxL10ox #21 showed lower expression of the *FLC* homologue gene, *MAF5*, which may explain the early flowering

time under the SD condition observed in this line.

3. *ICS1* was previously shown as a direct transcriptional target of CHE and its expression was suppressed in the *che-2* mutant in the absence of pathogen challenge. Such a basal level expression pattern was unclear in Fig 5m. Fig 5m showed that *ICS1* expression was induced by *Pst* DC3000 and such an induction was suppressed by HaRxL10. Interestingly, in the presence of HaRxL10 and *Pst* DC3000, *che-2* expressed much higher *ICS1* than just *Pst* DC3000-infected *che-2*. Does this mean that such HaRxL10-controlled *ICS1* expression is actually CHE independent?

Response: Since the basal expression of *ICS1* was low, the difference between the *ICS1* expression in the wild-type and the *che-2* was not obvious in our previous Y-axis scale. To better display our results, we chose two-segments Y-axis, where the suppression of basal *ICS1* expression in *che-2* mutant plants compared to the wild-type plants was obvious (Fig. R16, new Fig. 5m). Based on our data, the basal *ICS1* expression was reduced about 40% compared to the wild-type plants, which was consistent with previous study (Zheng et al., 2015).

Fig. R16 HaRxL10 interferes with CHE-mediated activation of *ICS1*. Relative expression of *ICS1* in the 3-week-old *Arabidopsis* of wild-type (Col-0), *che-2* and HaRxL10 overexpression (HaRxL10_{ox}) in different backgrounds (Col-0 or *che-2*) analysed by RT-qPCR with *UBQ5* as internal control at 0 or 1 dpi (day post-inoculation) after *Pst* DC3000 (OD_{600 nm} = 0.002) infection. The expression levels were normalised to the relative expression level in the wild-type plants. The data are shown as mean ± SEM (n = 6, 2 independent experiments with 3 technical replicates). The p values were calculated by three-way ANOVA analysis.

For Fig. 5m, we investigated the effects of presence or absence of three factors—CHE, HaRxL10 and *Pst* DC3000 on the *ICS1* expression. Our data showed that *Pst* DC3000 could induce *ICS1* expression (compare the left two bars), but this *Pst* DC3000-triggered *ICS1* induction was partially suppressed by HaRxL10 because the *Pst* DC3000-triggered *ICS1* induction was lower in HaRxL10_{ox} than the wild-type (compare the left four bars). To test if CHE plays a role in HaRxL10-mediated suppression of *Pst* DC3000-triggered *ICS1* induction, we analysed *ICS1* expression in

the *che-2* and HaRxL10ox *che-2* plants upon *Pst* DC3000 infiltration (the right four bars). If CHE did not play a role in HaRxL10-mediated suppression of *Pst* DC3000-triggered *ICS1* induction, the *ICS1* induction should be lower in HaRxL10ox *che-2* than *che-2*. However, the *Pst* DC3000-triggered *ICS1* induction was even higher in HaRxL10ox *che-2* compared to the *che-2* plants (compare the right four bars), suggesting that HaRxL10-mediated *ICS1* suppression was partially dependent on CHE as illustrated by **three-way ANOVA** analysis. Moreover, our data showed that the *Pst* DC3000-triggered *ICS1* induction in the HaRxL10ox *che-2* plants was not the same as in the *che-2* plants, indicating that additional components other than CHE also regulate HaRxL10-mediated *ICS1* suppression. Our RNA-seq analysis showed that HaRxL10 triggered a large-scale of transcriptome reprogramming and majority of HaRxL10 responsive genes are not regulated by CHE (**Supplementary Fig. 22**). Therefore, our data collectively suggest that CHE plays a role in HaRxL10-mediated *ICS1* suppression and other unknown proteins are also involved in this regulatory process.

Reference:

Zheng, X.Y., Zhou, M., Yoo, H., Pruneda-Paz, J.L., Spivey, N.W., Kay, S.A., and Dong, X. (2015). Spatial and temporal regulation of biosynthesis of the plant immune signal salicylic acid. *Proc Natl Acad Sci U S A* *112*, 9166-9173.

4. Fig 6h and i: in 6h, are there endogenous CHE proteins in the protoplasts used? If this is the case, the data shown in these two panels should be relatively similar for HaRxL10 suppression of *CCA1* expression, which is not the case when the two panels are compared.

Response: We apologise for the unclear description of Fig. 6h. We used *che-2 Arabidopsis* protoplasts for Fig. 6h, where *CHE* transcripts were significantly lower than the wild-type plants. In this case, we found that HaRxL10 did not affect the expression of *CCA1*. We repeated this experiment with 3 biological replicates and replaced previous data with 2 biological replicates with this new data (**new Fig. 6c**). However, HaRxL10 could significantly suppress the expression of *CCA1* in the wild-type *Arabidopsis* protoplast with overexpression of CHE by dual-luciferase assays (**Fig. 6d**). In addition, overexpression of HaRxL10 in *Arabidopsis* where endogenous CHE was functional also decreased the expression of *CCA1* (**Fig. 6a, b**). Therefore, we concluded that HaRxL10 suppresses *CCA1* expression in a CHE-dependent way.

In addition, the RNAseq data did not support this point that HaRxL10 suppresses *CCA1* expression (See comment 5).

5. The data from RNAseq and qRT-PCR appear to be inconsistent in several places. For example, *CCA1* is a known CHE target. RNAseq data showed almost no effect of HaRxL10 on *CCA1* expression (Fig.S13s) yet Fig 6g showed a clear suppression by HaRxL10.

Response: In addition to different techniques, different plants were used in these two figures. For **Supplementary Fig. 13s (new Supplementary Fig. 14s)**, the wild-type plants upon *Pst* DC3000-HaRxL10 were used. For **Fig. 6g (new Fig. 6b)**, two independent HaRxL10 overexpression lines were used. This data discrepancy may be due to the higher HaRxL10 protein levels in HaRxL10 overexpression plants than the wild-type plants upon *Pst* DC3000 TTSS-mediated HaRxL10 delivery. Higher amount of HaRxL10 may trigger more obvious *CCA1* suppression. We have added the explanation in our revised manuscript (**Page 20, lines 27-31**).

Such a discrepancy can also be found for *TCP22*, *LWD1*, *CKX5*, *ANAC055*, and *SIF* (Fig 6 and S11) when the 24 hpi time point is compared for data from RNAseq and qRT-PCR.

Response: We used RNA-seq to analyse the global transcriptome change triggered by HaRxL10. However, RNA-seq has less sensitivity compared to RT-qPCR for individual gene expression. For *TCP22* and *LWD1*, the time point **24 hpi** was chosen to re-analyse the gene expression using RT-qPCR since these two genes displayed high expression levels at 24 hpi based on RNA-seq. For a better comparison, we re-plotted data of the gene expressions at 24 hpi from RNA-seq. Results from both RNA-seq and RT-qPCR showed that HaRxL10 repressed *TCP22* in a CHE-dependent manner with more obvious change revealed by qRT-PCR (**Fig. R17a, b**). For *LWD1* expression analysis, the data from RNA-seq did not show significant difference between different genotypes and treatments (**Fig. R17c**). However, the *LWD1* expression at 24 hpi analysed by RT-qPCR showed mild difference between the wild-type and the *che-2* plants (**Fig. R17d**). The data discrepancy may be due to the detection sensitivity of two techniques. Therefore, we concluded that HaRxL10-triggered repression of *LWD1* **might** be partially dependent on CHE. Since *LWD1* and *REV4* are not the key clock components affected by HaRxL10, we have revised our model to remove these two components and emphasise our main findings (**new Fig. 7i; Page 27, lines 34-43**).

For *CKX5*, *ANAC055* and *SIF1*, the time point **12 hpi** was chosen to re-analyse the gene expression using RT-qPCR since HaRxL10-triggered gene induction could be observed at 12 hpi based on RNA-seq data. For a better comparison, we re-plotted data of the gene expressions at 12 hpi from RNA-seq. Results from both RNA-seq and qRT-PCR showed that HaRxL10 induced the expression of *CKX5*, *ANAC055* and *SIF1* in a CHE-dependent manner with more striking changes revealed by RT-qPCR (**Fig. R17e-j**).

Fig. R17 Comparisons of gene expressions by RNA-seq and RT-qPCR. a, c, e, g, i, Transcript abundance of *TCP22* (a), *LWD1* (c), *CKX5* (e), *ANAC055* (g) and *SIF1* (i) in the wild-type (Col-0) and *che-2* *Arabidopsis* leaves after infiltrated with *Pst* DC3000 or *Pst* DC3000-HaRxL10 represented by \log_2 CPM (count-per-million) from time-course RNA-seq. The data are shown as mean \pm SEM (n = 3 biological replicates). hpi, hour post-infiltration. b, d, f, h, j, Relative gene expression levels of *TCP22* (b), *LWD1*

(d), *CKX5* (f), *ANAC055* (h) and *SIF1* (j) in the wild-type (Col-0) and *che-2* *Arabidopsis* plants after *Pst* DC3000 or *Pst* DC3000-HaRxL10 infection analysed by RT-qPCR with *UBQ5* as an internal control. The pathogen infection assay was conducted at ZT0. The data are shown as mean \pm SEM (n = 9, 3 independent experiments with 3 technical replicates).

Instead of hpi, perhaps time of day should be used in RNAseq and qRT-PCR to allow a better comparison.

Response: We have revised our manuscript and shown both hpi and time of the day used in RNA-seq and RT-qPCR (**Fig. 6, Supplementary Fig. 12, Supplementary Fig. 15**).

Related to the RNAseq data, please clarify how to calculate the average expression of genes in a time course (Page 17 line 3).

Response: The average expression levels of genes contributed by different treatment conditions were calculated through linear regression followed by empirical Bayesian analysis.

In addition, it would be great if data from *che-2* could be included in Fig. S13. This is particularly important for the analysis of *CCA1* expression influenced by *CHE*.

Response: Except for *TCP22* and *CHE*, expressions of the other 18 clock genes did not display significant difference between the wild-type and the *che-2* plants. Moreover, we focused on the effect of HaRxL10 on the clock gene expression. We found that HaRxL10 repressed the expressions of *CHE*, *TCP22*, *RVE4* and *LWD1* based on RNA-seq. Subsequent qRT-PCR results showed that HaRxL10-triggered *TCP22* suppression requires *CHE* and HaRxL10-triggered *RVE4* and *LWD1* suppression may partially require *CHE*. Therefore, we chose to show the data in the wild-type plants to avoid overlap of 4 lines for better visualisation.

For this reviewer's request, we provided the RNA-seq of *CCA1* expression in both the wild-type and *che-2* plants (**Fig. R18a**). Previous study showed that *CHE* represses the expression of *CCA1*. The trend of *CCA1* transcription levels appears higher in the *che-2* plants (the green and purple lines) than in the wild-type plants (the blue and red lines). However, the statistical analysis using linear models followed by empirical Bayesian analysis showed no significance between the two genotypes (p value = 0.233) probably due to the lower sensitivity of RNA-seq. We analysed the expressions of *CCA1* in the wild-type and the *che-2* plants by qRT-PCR. Our results showed that the *CCA1* transcription level was higher in the *che-2* than the wild-type (**Fig. R18b**), which was consistent with the previous finding.

Fig. R18 Expression of *CCA1* in the wild-type and *che-2* *Arabidopsis* plants. **a, Transcript abundance of *CCA1* in the wild-type (Col-0) *Arabidopsis* leaves after infiltrated with *Pst* DC3000 or *Pst* DC3000-HaRxL10 represented by log₂CPM (count-per-million) from time-course RNA-seq. The data are shown as mean \pm SEM (n = 3 biological replicates). **b**, Expression of *CCA1* in the wild-type and the *che-2* *Arabidopsis* plants. Relative gene expression levels of *CCA1* in the 3-week-old wild-type (Col-0) and *che-2* *Arabidopsis* plants at ZT0 analysed by RT-qPCR with *UBQ5* as internal control. The data are shown as mean \pm SEM (n = 9, 3 independent experiments with 3 technical replicates). The *p* value was calculated using two-sided unpaired Student's *t*-test.**

6. Page 17 line 34: It is unclear what these circadian-controlled genes with robust rhythmic expression (circadian oscillation correlation > 0.7) are. Expression of most main clock genes did not change after infection with *Pst* DC3000 or *Pst* DC3000-HaRxL10 (Fig S13). Please use a table to show such genes.

Response: We showed the expressions of the circadian-controlled genes in **Supplementary Data 2 under the sheet “FFT-NLLS analysis results”**. There were 1584 genes in total, which showed a circadian oscillation correlation of more than 0.7 under at least one of the four conditions.

With only a 44 hr-time course in the RNAseq experiment, it might be difficult to accurately assess clock parameters, such as period, amplitude, and phase.

Response: The oscillation parameters were estimated by weighted non-linear regression analysis. The accuracy of the estimation mainly depends on the degree of freedom of the dataset. For oscillatory datasets, the temporal span covered by the datasets also contributes to the estimation accuracy of period. For our RNA-seq experiment, the 44-hr time course covers 2 cycles considering a 24-hr cycle, which is sufficient for the accurate estimation of the period. With a sampling frequency of 4 hours spanning 2 days with 3 biological replicates, the degree of freedom of the fitted cosine wave with an intercept is 32, which is sufficient for the accurate estimation of oscillation parameters. As a comparison, a standard Student's *t*-test on a triplicated dataset yields a degree of freedom of only 4.

7. Page 23 paragraph 3: the data should be presented in the result section.

Response: We have revised our manuscript as this reviewer suggested (**Page 27, lines 8 to 22**).

Reviewer #4:

This article demonstrates that the Hpa effector HaRxL10 stabilizes the Arabidopsis central circadian protein CHE and inhibits ZTL binding to CHE. The accumulation of CHE inhibits its binding to downstream gene promoters, leading to reduced expression of circadian rhythmic genes, SA-responsive genes, and flowering-related genes, and enhanced pathogenesis. The paper presents new findings on the relationship between the circadian cycle and plant immunity.

Response: We appreciate the reviewer's concise summary and recognition of this work.

The quality of experiments and logic are reliable, but further demonstration is needed for some points. Additionally, the writing needs improvement in terms of English language usage.

Response: We have revised our manuscript, added more experiments (new Fig. 3e; new Fig. 4i; new Fig. 6a, e; new Fig. 7a-d; new Supplementary Fig. 1; new Supplementary Fig. 16; new Supplementary Fig. 19; new Supplementary Fig. 21) and related explanations as well as checked the grammar to improve the quality of this manuscript.

Major

1. The sentences should be more concise, and the Material and Method section should be explained more thoroughly if necessary. There are several incomplete sentences in the text, such as “YFP, 33.2 kD. YFP-HaRxL10, 52.3 kD. HA-CHE, 28.3 kD”.

Response: We have revised these phrases to complete sentences (**Page 9, lines 6-8, 24-26, 43; Page 10, lines 7-8; SI Pages 4, 7, 11**). We also added more information in the Material and Method section.

2. The last paragraph in the introduction seems incomplete. It would be better to add a couple of sentences for closing remarks.

Response: We have revised our manuscript according to this reviewer's suggestion (**Page 3, lines 9-35**).

3. More description is needed to understand why RxL10 was chosen among the 130 effectors of Hpa. Providing detailed procedures for Y2H screening is helpful in the Material and Method section.

Response: We have added more description of the rationale for choosing HaRxL10 as our focus (**Page 3, lines 8-14**). We also provided the procedures of Y2H screening under the “Y2H assay” part in the Methods section (**Page 33, lines 35-45**).

4. The paragraph and experimental results focused on the CHE and CHE-mediated circadian cycle regulation. However, there is no explanation about CHE or its related mechanism. To make the importance of RxL10 function clear, it is recommended to provide a brief description of CHE-mediated regulation in any section.

Response: We added the information about CHE and its functions in the regulation of the circadian clock and plant immunity in the “Introduction” section (**Page 3, lines 14-25**).

5. The authors mention the TCP domain of CHE, but do not provide an explanation about it. Additionally, an explanation of the CHE gene or protein structure may be helpful in understanding its interaction with the effector.

Response: We have added the information about the TCP domain in our revised manuscript (**Page 6, lines 12-14**). We also provided the explanation of different domains of CHE as this reviewer suggested (**Page 6, lines 14-17**).

6. Please provide a description of TCP. Additionally, provide the importance of RxL10 function in terms of inhibiting CHE binding and interfering with the interaction between CHE and ZTL. The discussion section did not mention the novel aspect of RxL10 functioning as a virulence effector by manipulating the circadian clock.

Response: The TCP family proteins shared a common TCP domain for DNA-binding function. Based on these TCP domain sequence variations, TCP family members were further grouped into two major classes, named class I and class II TCPs, respectively. Class I is formed by a group of relatively closely related proteins, whereas class II can be further subdivided into two clades also based on differences within the TCP domain. The CIN clade exemplified by *CINCINNATA (CIN)* of *Antirrhinum*, contains genes involved in lateral organ development and the CYC/TB1 clade includes genes mainly involved in the development of axillary meristems giving rise to either flowers or lateral shoots. We added the description of TCP in our revised manuscript (**Page 26, lines 40-45; Page 27 lines 1-6**). In addition, we discussed the function of HaRxL10 as a virulence effector to interfere with CHE’s function and manipulate the circadian clock in our revised manuscript (**Page 28, lines 17-33**).

7. It is unclear whether the SP sequences of effectors used in this paper were present or removed. It appears that RxLR was used exclusively, but this should be accurately described and indicated in the figures.

Response: The signal peptide of HaRxL10 (HaRxL10^{SP}) was used for the signal peptide

secretion assay (**Fig. 1a, b**). The signal peptide was deleted for all the functional studies of HaRxL10, which is Δ SP-HaRxL10. We referred to Δ SP-HaRxL10 as HaRxL10 in short for convenience (all the other figures about HaRxL10). We have provided related explanation in the main text (**Page 4, lines 13 to 14**), the figure legend (**Page 5, line 5**) and the method part (**Page 29, lines 33-38**) in our revised manuscript.

8. HaRxL10 stabilizes CHE protein levels, and inhibiting its expression hinders CHE-binding to TBS. The text does not provide enough information to determine if HaRxL10 specifically binds to the TBS-binding site on CHE's surface. The structures between CHE and HaRxL10 are not discussed.

Response: Our Y2H results showed that HaRxL10 could not bind to the TCP domain of CHE, which usually mediates the DNA binding and probably contains a TBS-binding site (**Fig. 2c**). Therefore, HaRxL10 probably could not bind to the TBS-binding site on CHE's surface. The binding of HaRxL10 to the C-terminal of CHE (**Fig. 2c**) may trigger a conformational change of CHE, inhibiting its DNA-binding ability. We added the discussion in our revised manuscript (**Page 13, lines 13 to 18**).

9. Figure 5m suggests that if *che-2* cannot produce functional endogenous CHE, HaRxL10 may stabilize another component in the compromised CHE scenario. If the hypothesis that HaRxL10 compromises CHE in *che-2* is correct, a specific part of CHE in *che-2* may undergo a null mutation. Therefore, investigating a structural correlation between the HaRxL10 docking site on CHE is crucial for a comprehensive understanding.

Response: The *che-2* mutant is a knock-down line, where the T-DNA was inserted in the promoter region. Therefore, the *che-2* plants could still produce functional low-level endogenous CHE. The data presented in Fig. 5m indicated that HaRxL10-mediated *ICS1* repression was partially dependent on CHE since HaRxL10 could not repress *ICS1* expression in the *che-2* background. Instead, HaRxL10 even enhanced *ICS1* expression in the *che-2* background. One possibility is that HaRxL10 may activate or stabilise other *ICS1* activators in the low-level endogenous CHE scenario.

10. Page 5, line 25-30, the text mentions that there is no change in localization with effector + CHE co-expression. However, there is a slight change where the nuclei ring becomes nucleolar. It is necessary to mention this change.

Response: The nuclear localisation is important for the transcription factor CHE to fulfil its transcriptional regulation function. In this study, we wondered whether HaRxL10 could affect the nuclear localisation of CHE. As this reviewer mentioned, the picture shown in Supplementary Fig. 14 showed a slight change of CHE localisation within the nuclei. However, we may not conclude that HaRxL10 could change the localisation of CHE from nuclei ring to nucleolar without using specific markers of these structures. Since our focus is whether HaRxL10 perturbs the nuclear localisation of CHE, we

concluded in the manuscript that “Co-expression with HaRxL10-CFP did not perturb the nuclear localisation of RFP-CHE”.

Monor

- On page 9, line 6, change 'suppl. Fig 3a' to 'Fig 3'.

Response: We apologise for this mistake. We have revised the manuscript accordingly (**Page 10, line 7**).

- On page 9, line 1, the effector's name is incorrect.

Response: Thank you for pointing out this mistake. We have corrected the name of the effector in the revised manuscript (**Page 10, line 3**).

- The full name of TCP needs to be described.

Response: We have added the full name of TCP in the revised manuscript (**Page 3, line 15**).

- In Fig 5a, it is difficult to distinguish between healthy and infected samples based on phenotype. A clearer image needs to be replaced.

Response: As this reviewer suggested, we have replaced previous images with new pictures. The new photographs set against a white background enhanced the visual contrast to make the differences between leaves infected with *Pst* DC3000 or *Pst* DC3000-HaRxL10 in wild-type plants more distinguished (**Fig. R19a, new Fig. 5a**).

We apologise for the unclear description and images related to Fig. 5a. For each genotype, we infected 3-week-old *Arabidopsis* leaves with either *Pst* DC3000 or *Pst* DC3000-HaRxL10. *Pst* DC3000 is a virulent pathogen for *Arabidopsis* and causes obvious disease symptoms. Consequently, leaves injected with *Pst* DC3000 exhibited a yellower and unhealthy appearance compared to uninfected plants (**Fig. R19b**). The introduction of effector HaRxL10 into *Pst* DC3000 exacerbated the severity of the disease symptoms.

Fig. R19 HaRxL10 enhances disease symptoms in leaves triggered by *Pst* DC3000 in a CHE-dependent manner. a, Disease symptoms in 3-week-old wild-type (Col-0), *che-2* and *CHE* complementation line (CHE_{CE}) *Arabidopsis* leaves infiltrated with *Pst* DC3000 or *Pst* DC3000-HaRxL10 ($OD_{600\text{ nm}} = 0.002$) at 3 days post-inoculation. The 4th and 5th infiltrated leaves from three representative plants were shown. Scale bar, 2 cm. **b,** Photos of the 4th and 5th leaves from the representative 3-week-old wild-type (Col-0), *che-2* and *CHE* complementation line (CHE_{CE}) *Arabidopsis* plants. Scale bar, 2 cm.

- Page 17, line 27, mediaed > mediated.

Response: We apologise for this mistake. We have revised the manuscript accordingly (Page 20, line 40).

RESPONSE LETTER

Reviewer #1:

I thank the authors for addressing all my comments and most of my concerns. I appreciate that they added new experimental information for this. The new data showing alterations on circadian rhythms in maltose levels in plants overexpressing HaRxL10, and its dependence on CHE, are quite clear.

I am still not very convinced of a significant GLOBAL effect of HaRxL10 on clock regulated gene expression. The authors are correct in mentioning that it is not simple to address global effects of clock gene expression under diurnal conditions, so they evaluated circadian expression for a core clock gene, CCA1, and an output gene, CAB2, under constant light conditions following circadian clock entrainment. Here the authors indicate that there is an effect on CCA1 mRNA circadian amplitude, but not period or phase, while for CAB2 they do observe differences in period and phase. Based on the data provided, which is associated with only the second day in constant light, the difference resulting in a period difference in CAB expression can only be seen in a single timepoint (the last one on their graph). Since it is a short timecourse, the reliability of a minor period effect is difficult to evaluate. The fact that no effect is observed for CCA1 period or phase in this same period of time and experimental condition, suggests that the effect of HaRxL10 on clock regulated gene expression is far from being strong and global. The authors should attenuate their claims associated with global effects of HaRxL10 on circadian oscillations of gene expression.

Response: We appreciate this critical point raised by this reviewer. We agree that it is hard to assess the effect of HaRxL10 on the circadian-regulated oscillatory gene under the diurnal condition which we used in our RNA-seq experiment. Although we found distributions of periods, phases and amplitudes of oscillatory genes were affected by *Pst* DC3000-HaRxL10 infection through statistical analysis, HaRxL10-triggered effect on the expression of different oscillatory genes may be different and mild. We did show that HaRxL10 could affect the expression of several circadian-regulated oscillatory genes, including *CAB2* (Fig. 6e), *CKX5* (Supplementary Fig. 13a, d, h), and *SIF1* (Supplementary Fig. 13c, f, i). To tune down the conclusion and avoid misleading, we removed the previous description on the global impact of HaRxL10 on clock-regulated genes and opted to conclude that HaRxL10 affects the expression of several circadian-regulated oscillatory genes in our revised manuscript as this reviewer suggested. Please refer to the main text for details (Page 2, lines 7-9; Page 21, lines 23-32; Page 28, lines 7-13, lines 31-40).

Reviewer #2:

Thank you for addressing my comments and providing extra experimental information. I think that the title "HaRxL10 hijacks the circadian clock component CHE to perturb both plant development and immunity" is appropriate, but that the suggestion that

HaRXL10 "globally affects circadian controlled genes" or that its interaction with CHE affects "the period, phase and amplitude of circadian rhythmic genes globally" is not supported.

The period predictions based on 44h of gene expression data are not sufficiently powered to be reliable. Although leaf movement data are provided in the response to the reviewers, the methods for these are not stated that I could find, nor are they presented in the revised manuscript or supplementary material. I cannot see how the period differences in leaf movement were arrived at, and certainly by visual inspection they are not apparent. It is not clear why these were not included in the revised version. The maltose rhythms are reported, but are difficult to interpret as such a stark lack of rhythmicity in the HaRxL10ox line would be expected to be reflected in other phenotypes too.

If this manuscript is to be accepted, the implications about the effects on the circadian clock or global rhythmicity should be removed as these are not supported by the data.

Response: Thank you for this critical point. We performed a 44-h time-course RNA-seq experiment to assess the effect of HaRxL10 on the circadian-regulated oscillatory gene expression. This 44-hour time span was chosen based on several plant circadian clock studies including the widely-used Diurnal database for providing the oscillatory information of specific genes (Mockler et al., 2007). Although we found distributions of periods, phases and amplitudes of oscillatory genes were affected by *Pst* DC3000-HaRxL10 infection through statistical analysis, HaRxL10-triggered effect on the expression of certain oscillatory gene may be different and mild. We agree that the claim that HaRxL10 globally affects the circadian clock gene expression and rhythmicity lacks strong evidence. To tune down this conclusion and avoid misleading, we removed previous description on the global impact of HaRxL10 on clock-regulated genes and opted to conclude that HaRxL10 affects the expression of several circadian-regulated oscillatory genes in our revised manuscript. Please refer to the main text for details (Page 2, lines 7-9; Page 21, lines 23-32; Page 28, lines 7-13, lines 31-40).

Leaf movement data was previously included in the "Response to reviewer" only. Therefore, the detailed method of this experiment and related statistical analysis were not included in the previous version of the revised manuscript. We apologise for making this confusion. In this revised manuscript, we included the leaf movement data in the main figure (Fig. R1; new Fig. 7a-c and new Supplementary Fig. 18) and provided the experiment details in the Method part (Page 39, lines 10-18). Our results showed that overexpression of HaRxL10 in the wild-type plant delayed the phase of leaf movement while overexpression of HaRxL10 in *che-2* displayed a similar phase as *che-2*, suggesting that HaRxL10-triggered leaf movement phase delay is dependent on CHE. However, HaRxL10 shortened the period and reduced the amplitude of leaf movement in both wild-type and *che-2* backgrounds, indicating that HaRxL10 requires other unknown components to affect the period and amplitude of leaf movement.

Maltose is the main form of a plant-available carbon source during the night, which is

derived from starch breakdown. Previous study showed that maltose content displays a circadian rhythm with a peak during subjective night under the constant light condition (Lu et al., 2005). Our results of maltose contents in the wild-type plants are consistent with the previous finding with regard to the peak time of maltose content. Compared to leaf movement, HaRxL10 had a greater impact on the rhythmicity of maltose contents, resulting in arrhythmicity in HaRxL10 overexpression plants in the wild-type background. One possible explanation is that maltose metabolism is regulated by multiple factors including synthesis and degradation rate of starch, cellular redox status, enzyme activity and gene expression (Glaring et al., 2012; Lu and Sharkey, 2006). Therefore, maltose level might be more sensitive to HaRxL10-triggered alternation of cellular status and gene expression compared to leaf movement.

Although the plant circadian clock regulates multiple physiological processes, different physiological processes may be regulated by different clock components. Moreover, one specific clock component may contribute differently to circadian output pathways. For example, *cca1* mutant displays a short period of leaf movement (Yakir et al., 2009) and early flowering (Niwa et al., 2007) but no obvious hypocotyl growth phenotype (Mizoguchi et al., 2002), indicating that CCA1 plays different roles in regulating different circadian-controlled physiological processes. Several physiological processes, such as hypocotyl growth, leaf movement and flowering time are widely accepted as representative circadian-controlled output pathways. However, our understanding of how these physiological processes are regulated by specific clock components remains limited. Although we showed that HaRxL10-induced rhythmic changes of leaf movement and maltose contents are dependent on CHE, CHE may contribute differently, which partially accounts for different effects of HaRxL10 on these circadian-regulated physiological pathways.

Reference:

- Glaring, M.A., Skryhan, K., Kötting, O., Zeeman, S.C., and Blennow, A. (2012). Comprehensive survey of redox sensitive starch metabolising enzymes in *Arabidopsis thaliana*. *Plant Physiol Biochem* 58, 89-97.
- Lu, Y., Gehan, J.P., and Sharkey, T.D. (2005). Daylength and circadian effects on starch degradation and maltose metabolism. *Plant Physiol* 138, 2280-2291.
- Lu, Y., and Sharkey, T.D. (2006). The importance of maltose in transitory starch breakdown. *Plant Cell Environ* 29, 353-366.
- Mizoguchi, T., Wheatley, K., Hanzawa, Y., Wright, L., Mizoguchi, M., Song, H.R., Carré, I.A., and Coupland, G. (2002). LHY and CCA1 are partially redundant genes required to maintain circadian rhythms in *Arabidopsis*. *Dev Cell* 2, 629-641.
- Mockler, T.C., Michael, T.P., Priest, H.D., Shen, R., Sullivan, C.M., Givan, S.A., McEntee, C., Kay, S.A., and Chory, J. (2007). The DIURNAL project: DIURNAL and circadian expression profiling, model-based pattern matching, and promoter analysis. *Cold Spring Harb Symp Quant Biol* 72, 353-363.
- Niwa, Y., Ito, S., Nakamichi, N., Mizoguchi, T., Niinuma, K., Yamashino, T., and Mizuno, T. (2007). Genetic linkages of the circadian clock-associated genes, TOC1, CCA1 and LHY, in the photoperiodic control of flowering time in *Arabidopsis thaliana*. *Plant Cell Physiol* 48, 925-937.

Yakir, E., Hilman, D., Kron, I., Hassidim, M., Melamed-Book, N., and Green, R.M. (2009). Posttranslational regulation of CIRCADIAN CLOCK ASSOCIATED1 in the circadian oscillator of *Arabidopsis*. *Plant Physiol* 150, 844-857.

Fig. R1 HaRxL10 affects leaf movement of *Arabidopsis*. **a-b**, Rhythm of leaf movement was measured in wild-type (Col-0) and HaRxL10 overexpression (HaRxL10_{ox} #21 in Col-0) (**a**) as well as *che-2* and HaRxL10 overexpression (HaRxL10_{ox} #4 in *che-2*) (**b**) *Arabidopsis* seedlings. Plants were grown under 12 h light/12 h dark conditions for 7 days and transferred to the constant light (LL) condition for 1 day. Pictures were then taken once an hour under the LL condition. Distance between two cotyledons was measured to represent the leaf movement. Gray bar, subjective night. The data are shown as mean \pm SEM (n = 12 seedlings). **c-e**, Phase (**c**), period (**d**) and amplitude (**e**) were calculated by nonlinear regression using a cosine wave. The *p* values were calculated by unpaired *t*-test with Welch's correction.

Reviewer #3:

In this revised ms, the authors did a nice job to address most of my comments. A number of new experiments were performed in order to strengthen the work. Here I have a few more comments:

a. ICS1 was previously shown to be a direct transcriptional target of CHE and ICS1 is a key gene in SA biosynthesis. In addition to some SA related genes, expression of ICS1 should be examined as shown in Fig. 5i-k. This reviewer agrees that Fig 5m shows HaRxL10-mediated ICS1 suppression was not seen in the *che2* mutant. A further induction of ICS1 in *che2* by HaRxL10 is quite interesting and unexpected. Does this induction also correlate with an increase in SA levels? How would this data explain expression suppression of some other SA related genes by HaRxL10 shown in Fig 5 and other places in this ms? The authors can also measure SA levels in the Col-0 and

che2 plants infected Pst DC3000 or Pst DC3000 HaRxL10 and in HaRxL10 overexpression lines of Arabidopsis (in Col-0 and *che2*) with or without Pst DC3000 infection. Such data shall provide a better clarification on the role of HaRxL10 in SA-mediated defence regulation.

Response: As this reviewer suggested, we analysed *ICS1* expression as shown in Fig. 5i-k (**Fig. R2a; new Supplementary Fig. 12**). Under this condition, HaRxL10 induced *ICS1* expression in both wild-type and *che-2* plants, indicating that other components contribute to HaRxL10-mediated *ICS1* expression. We performed dual-luciferase assays in tobacco leaves and observed significant repression of *ICS1* transcription in the presence of CHE and HaRxL10 (**Fig. R2b; new Supplementary Fig. 12**). These data indicate that although HaRxL10 represses CHE-mediated *ICS1* induction, HaRxL10 may promote expression of other *ICS1* positive regulators, such as *SARD1* and *CBP60g*. To test this hypothesis, we used wild-type, *che-2* and stable HaRxL10 overexpression transgenic plants infiltrated with *Pst* DC3000 using the same dosage as the disease phenotype analysis. Our results showed that the expression of *SARD1*, *CBP60g* and *ICS1* were induced by HaRxL10 (**Fig. R2c-e; new Supplementary Fig. 12**). Consistent with these results, overexpression of HaRxL10 promoted free SA contents (**Fig. R2f; new Supplementary Fig. 12**). HaRxL10-triggered increase of SA seems contradictory to the function of HaRxL10 as the virulence effector to repress plant immunity. To address this puzzle, we further analysed the expression of *PR2*, which is a downstream defense gene identified by our RNA-seq experiment and also a more downstream proxy of SA-mediated defence level. HaRxL10 overexpression and *che-2* plants showed decreased *PR2* expression (**Fig. R2g; new Supplementary Fig. 12**), which were consistent with enhanced disease susceptibility of these plants. In summary, HaRxL10 affects SA-mediated defence regulation in a complicated way through multiple plants targets (**Fig. R2h; new Supplementary Fig. 12**). HaRxL10 represses CHE-mediated *ICS1* induction but promotes *SARD1*/*CBP60g*-mediated *ICS1* induction, resulting in higher *ICS1* expression and SA level. HaRxL10 also represses *PR2* expression to dampen defence response and cause disease susceptibility.

Thank you for the experiment suggestions which offered us an opportunity to make our conclusion clearer. The effects of HaRxL10 on SA-related defence is complicated because HaRxL10 affects multiple SA synthesis-related and defence gene expression. We have modified our description and related figures in our revised manuscript (**Page 16, lines 33-45; Page 17, lines 1-16**).

Fig. R2 Effects on HaRxL10 on SA-related gene expression and SA levels. **a**, Relative gene expression levels of *ICS1* in the wild-type (Col-0) and *che-2* *Arabidopsis* plants at 24 hours after *Pst* DC3000 or *Pst* DC3000-HaRxL10 ($OD_{600\text{ nm}} = 0.002$) infection analysed by RT-qPCR with *UBQ5* as an internal control. The data are shown as mean \pm SEM ($n = 6$, 2 independent experiments with 3 technical replicates). The p values were calculated by two-way ANOVA followed by Šídák's multiple comparisons test. **b**, Dual-luciferase assay performed using *N. benthamiana* leave transiently co-expressing different protein combinations and the reporter driven by *ICS1* promoter. The ratio of firefly luciferase and renilla luciferase activities was calculated and normalised to the control. The data are shown as mean \pm SEM ($n = 9$, 3 independent experiments with 3 technical replicates). The p value was calculated by two-sided unpaired Student's *t*-test. **c-g**, SA-related gene expression and endogenous SA levels were analysed in 3-week-old *Arabidopsis* leaves infiltrated with *Pst* DC3000 ($OD_{600\text{ nm}} = 0.00005$) at 1 day post-inoculation. Relative expression of *ICS1* (**c**), *SARD1* (**d**), *CBP60g* (**e**) and *PR2* (**g**) analysed by RT-qPCR with *UBQ5* as an internal control. The expression levels were normalised to the relative expression level in the wild-type plants. Data represent the mean \pm SEM ($n = 9$, 3 independent experiments with 3 technical replicates). The p values were calculated by two-way ANOVA followed by

Šídák's multiple comparisons test. Endogenous free SA levels were measured (f). Data represent the mean \pm SEM (n = 3 independent experiments). The *p* values were calculated by two-way ANOVA followed by Šídák's multiple comparisons test. **h**, Illustration of effects of HaRxL10 on SA-related gene expression and SA levels. HaRxL10 has multiple targets and exerts different effects on different targets. HaRxL10 represses CHE-mediated *ICS1* induction but promotes SARD1/CBP60g-mediated *ICS1* induction, resulting in higher *ICS1* expression and SA levels. HaRxL10 also represses *PR2* expression to dampen defence response and cause disease susceptibility.

b. New data were included in Fig 6 to link HaRxL10 function to the regulation of key clock genes *CCA1* and *CAB2*. However, the inclusion of only one cycle (24 hr) in LL would make the measurement of clock parameters, such as period, phase, and amplitude, less reliable. CHE is a known repressor of *CCA1* (Pruneda-Paz et al, Science 2009). If HaRxL10 suppresses (and acts through) CHE function as stated in this ms, then would one expect to see an induction, rather than a suppression of *CCA1*, by HaRxL10? Perhaps the authors could do qRT-PCR with Col-0 and *che2* plants infected Pst DC3000 or Pst DC3000 HaRxL10 to verify the data. In addition, HaRxL10 overexpression in Col-0 and in *che2* with or without Pst DC3000 infection can also be used to verify the data.

Response: As this reviewer suggested, we performed RT-qPCR to analyse *CCA1* expression in the wild-type and *che-2 Arabidopsis* plants infiltrated with *Pst* DC3000 or *Pst* DC3000-HaRxL10 (**Fig. R3a; new Fig. 6d**). HaRxL10 significantly repressed the expression of *CCA1* in the wild-type plants. The *che-2* plants displayed higher *CCA1* expression level than wild-type plants in the absence of HaRxL10, which is consistent with the previous finding that CHE negatively regulates *CCA1* expression (**Pruneda-Paz et al., 2009**). HaRxL10 could still repress *CCA1* expression in the *che-2* plants, suggesting that other components may regulate HaRxL10-mediated *CCA1* suppression in this scenario. Furthermore, overexpression of HaRxL10 in both wild-type and *che-2* could also suppress the expression of *CCA1* (**Fig. R3b**). These results could not be explained by HaRxL10 interfering with CHE's binding to *CCA1* promoter, since CHE negatively regulates *CCA1* expression. Therefore, HaRxL10 may affect other components which regulate *CCA1* expression.

One possibility is that HaRxL10 represses the expression of positive regulators of *CCA1*³⁹, *TCP22*, in the *che-2* background (**Fig. R3c; Supplementary Fig. 16d**), leading to the HaRxL10-mediated suppression of *CCA1*. HaRxL10 overexpression plants displayed higher amplitude and average expression of *TOC1* compared to the wild-type plants, indicating that HaRxL10 could promote the expression of *TOC1*, which is a *CCA1* repressor (**Fig. R3d; Supplementary Fig. 17**). This HaRxL10-mediated *TOC1* induction may also account for HaRxL10-mediated suppression of *CCA1*. In addition to transcriptional regulation, HaRxL10 may indirectly affect the expression of *CCA1* through protein interaction. We found that HaRxL10 interacted with *CCA1* in yeast (**Supplementary Fig. 1**). A previous study showed that

overexpression of *CCA1* suppresses endogenous *CCA1* expression, suggesting that *CCA1* protein could regulate its gene expression (Wang and Tobin, 1998). Therefore, this gene regulatory function of *CCA1* protein may be altered upon interacting with HaRxL10, resulting in altered gene expression of *CCA1*.

We have added discussion about HaRxL10-mediated *CCA1* suppression in our revised manuscript (Page 20, lines 30-45; Page 21, lines 1-14).

Reference:

Pruneda-Paz, J.L., Breton, G., Para, A., and Kay, S.A. (2009). A Functional Genomics Approach Reveals CHE as a Component of the Arabidopsis Circadian Clock. *Science* 323, 1481-1485.

Wang, Z.Y., and Tobin, E.M. (1998). Constitutive expression of the CIRCADIAN CLOCK ASSOCIATED 1 (*CCA1*) gene disrupts circadian rhythms and suppresses its own expression. *Cell* 93, 1207-1217.

Fig. R3 HaRxL10 suppresses the expression of *CCA1*. **a**, Relative gene expression levels of *CCA1* in the wild-type (Col-0) and *che-2* *Arabidopsis* plants at 24 hours after *Pst* DC3000 or *Pst* DC3000-HaRxL10 (OD_{600 nm} = 0.002) infection analysed by RT-qPCR with *UBQ5* as an internal control. The data are shown as mean ± SEM (n = 6, 2 independent experiments with 3 technical replicates). The *p* values were calculated by two-way ANOVA followed by Šidák's multiple comparisons test. **b**, Relative expression of *CCA1* at CT0 were analysed in *Arabidopsis* leaves by RT-qPCR with *UBQ5* as an internal control. The expression levels were normalised to the relative expression level in the wild-type plants. Data represent the mean ± SEM (n = 6, 2 independent experiments with 3 technical replicates). The *p* values were calculated by

two-way ANOVA followed by Šídák's multiple comparisons test. **c**, Relative gene expression levels of TCP22 in the wild-type (Col-0) and *che-2 Arabidopsis* plants at 24 hours after *Pst* DC3000 or *Pst* DC3000-HaRxL10 ($OD_{600\text{ nm}} = 0.002$) infection analysed by RT-qPCR with *UBQ5* as an internal control. The data are shown as mean \pm SEM ($n = 6$, 2 independent experiments with 3 technical replicates). The p values were calculated by two-way ANOVA followed by Šídák's multiple comparisons test. **d**, Relative expression levels of *TOC1* in 3-week-old wild-type (Col-0) and HaRxL10 overexpression (HaRxL10_{ox} #18) *Arabidopsis* plants. Plants were grown under the 12 h light/12 h dark condition for 3 weeks and transferred to the constant light (LL) condition for 24 hours. Samples were collected at 4 hour-intervals under the LL condition and analysed by RT-qPCR with *UBQ5* as the internal control. White bar, subjective day. Gray bar, subjective night. The data are shown as mean \pm SEM ($n = 3$). Amplitude and average expression were calculated by nonlinear regression using a cosine wave. The p values were calculated by unpaired t -test with Welch's correction. This experiment was repeated twice with similar results.

c. In Fig. 7a-b, *che2* does not seem to affect the rhythm of maltose daily levels, compared with WT (however, a side-by-side comparison should be made to confirm this observation). Interestingly, HaRxL10 by itself almost abolishes this clock output in Col-0. If HaRxL10 acts through suppressing CHE function, one would expect that HaRxL10 ox/Col-0 and *che2* have a similarly compromised clock phenotype, just like other phenotypes described in this ms (disease susceptibility, hypocotyl length, flowering time etc). Please explain the maltose phenotype.

Response: Maltose is the main form of a plant-available carbon source during the night, which is derived from starch breakdown. Previous study showed that maltose content displays a circadian rhythm with a peak during subjective night under the constant light condition (Lu et al., 2005). Our results of maltose contents in the wild-type plants are consistent with the previous finding regarding the peak time of maltose content. Compared to leaf movement, HaRxL10 had a greater impact on the rhythmicity of maltose contents, resulting in arrhythmicity in HaRxL10 overexpression plants in the wild-type background. One possible explanation is that maltose metabolism is regulated by multiple factors including synthesis and degradation rate of starch, cellular redox status, enzyme activity and gene expression (Glaring et al., 2012; Lu and Sharkey, 2006). Therefore, maltose level might be more sensitive to HaRxL10-triggered alternation of cellular status and gene expression compared to leaf movement.

Although the plant circadian clock regulates multiple physiological processes, different physiological processes may be regulated by different clock components. Moreover, one specific clock component may contribute differently to circadian output pathways. For example, *cca1* mutant displays a short period of leaf movement (Yakir et al., 2009) and early flowering (Niwa et al., 2007) but no obvious hypocotyl growth phenotype (Mizoguchi et al., 2002), indicating that CCA1 plays different roles in regulating different circadian-controlled physiological processes. Several physiological processes,

such as hypocotyl growth, leaf movement and flowering time are widely accepted as representative circadian-controlled output pathways. However, our understanding of how these physiological processes are regulated by specific clock components remains limited. Although we showed HaRxL10-induced rhythmic changes of leaf movement and maltose contents are dependent on CHE, CHE may contribute differently, which partially accounts for different effects of HaRxL10 on these circadian-regulated physiological pathways.

Reference:

- Glaring, M.A., Skryhan, K., Kötting, O., Zeeman, S.C., and Blennow, A. (2012). Comprehensive survey of redox sensitive starch metabolising enzymes in *Arabidopsis thaliana*. *Plant Physiol Biochem* *58*, 89-97.
- Lu, Y., Gehan, J.P., and Sharkey, T.D. (2005). Daylength and circadian effects on starch degradation and maltose metabolism. *Plant Physiol* *138*, 2280-2291.
- Lu, Y., and Sharkey, T.D. (2006). The importance of maltose in transitory starch breakdown. *Plant Cell Environ* *29*, 353-366.
- Mizoguchi, T., Wheatley, K., Hanzawa, Y., Wright, L., Mizoguchi, M., Song, H.R., Carré, I.A., and Coupland, G. (2002). LHY and CCA1 are partially redundant genes required to maintain circadian rhythms in *Arabidopsis*. *Dev Cell* *2*, 629-641.
- Niwa, Y., Ito, S., Nakamichi, N., Mizoguchi, T., Niinuma, K., Yamashino, T., and Mizuno, T. (2007). Genetic linkages of the circadian clock-associated genes, TOC1, CCA1 and LHY, in the photoperiodic control of flowering time in *Arabidopsis thaliana*. *Plant Cell Physiol* *48*, 925-937.
- Yakir, E., Hilman, D., Kron, I., Hassidim, M., Melamed-Book, N., and Green, R.M. (2009). Posttranslational regulation of CIRCADIAN CLOCK ASSOCIATED1 in the circadian oscillator of *Arabidopsis*. *Plant Physiol* *150*, 844-857.

d. Spp1 Fig 17b: The FLC expression level correlates strongly with flowering time. The increased expression of FLC in HaRxL10 ox #21/Col-0 does not support the flowering time data. Spp1 Fig 17c that shows MAF4 expression is similar between Col-0 and HaRxL10 ox #21/Col-0 also fails to support the early flowering phenotype of HaRxL10 ox #21/Col-0. Please double check the data and provide an explanation for the data.

Response: We previously used 10-day-old seedlings to analyse the expression of *FLC* and *MAF4* under the short day (SD, 8 h light/16 h dark) condition. As this reviewer mentioned, the expression levels of *FLC* and *MAF4* in HaRxL10 overexpression line #21 did not correlate well with the early flowering phenotype. Wild-type (Col-0) *Arabidopsis* plants usually start to bolting after 6 weeks under the SD condition. Ten-day is probably not a good developmental stage to examine flowering gene expression under SD condition since floral transition does not occur at this time.

In the revised manuscript, we used 5-week-old plants to re-analyse the expression of *FLC* and *MAF4* under the SD condition (Fig. R4). HaRxL10 overexpression line #21 displayed lower expression level of *FLC* compared to the wild-type plants, which correlated well with the early flowering phenotype of this line. *MAF4* expression level

of HaRxL10 overexpression line #21 was similar to the wild-type plants. Since *FLC* is the well-known major repressor of flowering (He, 2012), the lower expression of *FLC* alone may account for early flowering of HaRxL10 overexpression line #21. We updated *FLC* expression data, related figure legend and main text in our revised manuscript (new Supplementary Fig. 19b; Page 24, lines 33-39).

Reference:

He, Y. (2012). Chromatin regulation of flowering. *Trends Plant Sci* 17, 556-562.

Fig. R4 HaRxL10 affects *FLC* expression in a CHE-dependent manner. Relative expression of *FLC* (a) and *MAF4* (b) in the 5-week-old seedlings of wild-type (Col-0), *che-2* and HaRxL10 overexpression (HaRxL10_{ox}) in different backgrounds (Col-0 or *che-2*) analysed by RT-qPCR with *UBQ5* as an internal control. The expression levels were normalised to the relative expression level in the wild-type plants. Short day, 8 h light/16 h dark. Samples were collected at ZT8. The data are shown as mean \pm SEM (n = 9, 3 independent experiments with 3 technical replicates). The *p* values were calculated by one-way ANOVA followed by Holm-Šidák's multiple comparisons test.

e. Leaf movement data could provide a strong support to link HaRxL10 function to circadian regulation. Rather than including them in the Rebuttal document, the authors should include such data as a main figure (e.g. in Fig 7), using HaRxL10 overexpression in Col-0 and in *che2*.

Response: Thank you for your suggestion. In this revised manuscript, we included the leaf movement data in the main figure (Fig. R5; new Fig. 7a-c and new Supplementary Fig. 18) and provided the experiment details in the Method part (Page 39, lines 10-18). Our results showed that overexpression of HaRxL10 in the wild-type plant delayed the phase of leaf movement while overexpression of HaRxL10 in *che-2* displayed a similar phase as *che-2*, suggesting that HaRxL10-triggered leaf movement phase delay is dependent on CHE. However, HaRxL10 shortened the period and reduced the amplitude of leaf movement in both wild-type and *che-2* backgrounds, indicating that HaRxL10 requires other unknown components to affect the period and amplitude of leaf movement.

Fig. R5 HaRxL10 affects leaf movement of *Arabidopsis*. a-b, Rhythm of leaf movement was measured in wild-type (Col-0) and HaRxL10 overexpression (HaRxL10_{ox} #21 in Col-0) (a) as well as *che-2* and HaRxL10 overexpression (HaRxL10_{ox} #4 in *che-2*) (b) *Arabidopsis* seedlings. Plants were grown under 12 h light/12 h dark conditions for 7 days and transferred to the constant light (LL) condition for 1 day. Pictures were then taken once an hour under the LL condition. Distance between two cotyledons was measured to represent the leaf movement. Gray bar, subjective night. The data are shown as mean \pm SEM (n = 12 seedlings). c-e, Phase (c), period (d) and amplitude (e) were calculated by nonlinear regression using a cosine wave. The p values were calculated by unpaired t -test with Welch's correction.

Reviewer #4:

The revised version incorporates the previously omitted mechanism of HaRxL10's interaction with the circadian clock component CHE, which has significantly enhanced the manuscript. I have no significant objections to the comments that I have revised. However, the presentation of these new findings has given rise to several additional queries.

In the supplementary figure 1, the authors demonstrated that several clock components, including CHE, interact with HaRxL10. CCA1 was identified as one of the interacting components. Furthermore, the authors asserted that HaRxL10 influenced the expression of central clock genes, including CCA1, through the observation of CCA1 gene expression patterns in HaRxL10-overexpressing samples. Nevertheless, it is challenging to ascertain whether the alterations in CCA1 gene expression are a direct consequence of HaRxL10-mediated transcription or an indirect effect of CCA1 protein interaction. In light of these considerations, the conclusion appears to be somewhat

hasty.

Response: We agree with this reviewer that HaRxL10 may affect CCA1 at both transcription level and protein level. Our results showed that HaRxL10 affects transcripts abundance of *CCA1* (**Fig. 6a-d**) and interacts with CCA1 protein (**Supplementary Fig. 1**). A previous study showed that overexpression of CCA1 suppresses endogenous *CCA1* expression, suggesting that CCA1 protein could regulate its gene expression (**Wang and Tobin, 1998**). Therefore, this gene regulatory function of CCA1 protein may be altered upon interacting with HaRxL10, resulting in altered gene expression of *CCA1*. We have added related discussion in our revised manuscript (**Page 21, lines 5-13**).

Reference:

Wang, Z.Y., and Tobin, E.M. (1998). Constitutive expression of the CIRCADIAN CLOCK ASSOCIATED 1 (CCA1) gene disrupts circadian rhythms and suppresses its own expression. *Cell* *93*, 1207-1217.

Reviewer #2:

Thank you for responding to my previous comments. I feel that the authors are very keen to hang onto the idea that HaRxL10 has an effect on the circadian clock, which is not justified by their data. The statement in the abstract "HaRxL10 triggered reprogramming of the transcriptome, particularly affecting the expression of central clock genes and circadian rhythmic genes." should be tempered to note those genes that are actually affected, e.g. SA-related defence genes. I do not find the leaf movement data compelling, nor the maltose data. If this manuscript is to be accepted, the implications about the effects on the circadian clock or global rhythmicity should be removed as these are not supported by the data.

Response: As this reviewer suggested, we have removed all direct claims that HaRxL10 impacts the circadian clock in general or circadian regulated processes in our revised manuscript.

Reviewer #3:

The authors addressed most of my comments. Because evidence of HaRxL10 in a direct regulation of the circadian clock, especially via a specific circadian clock component CHE, is limited, this reviewer would like to suggest the authors to carefully go through the manuscript and tone down some of the conclusions. CHE-dependent and -independent output caused by HaRxL10 should be clearly discussed in the Discussion section.

Response: As this reviewer suggested, we have removed all direct claims that HaRxL10 impacts the circadian clock in general or circadian regulated processes in our revised manuscript. In the Discussion section, we proposed that although HaRxL10 may have other plant targets, CHE is one of linkers between pathogens and the plant circadian clock. Moreover, we discussed CHE-dependent and -independent output caused by HaRxL10 in the Discussion section.

Reviewer #4:

I thank the authors for addressing all my comments. The revised version includes the previously omitted mechanism of HaRxL10's interaction with the with the circadian clock component CHE, which significantly improved the manuscript.

Response: We appreciate the reviewer's recognition of our work.